# How Few-Shot Examples Add Up: A Causal Decomposition of Function Vectors in In-Context Learning

**Entang Wang**[1] **Yiwei Wang**[1] **Aleksandra Bakalova**[1] **Michael Hahn**[1]

## Abstract

In-context learning (ICL) excels at new tasks from minimal examples, yet we still lack a mechanistic explanation of how few-shot prompts shape a model's function vector (FV)–a causal activation direction that drives task behavior on the ICL query. Across tasks and models, an $n$-shot FV is well-approximated by a linear combination of example-level sub-FVs, suggesting additive and composable contributions from individual demonstrations. Beyond additivity, we show that models contextualize individual examples' representations based on prior examples to adaptively reweight which demonstrations dominate the FV: attention shifts toward examples that are more informative and less ambiguous under the context. Finally, a causal decomposition separates Query–Key routing from Value updates, finding that contextualization's most consistent contributions to FV quality arise from Query–Key alignment–particularly in ambiguous settings–while Value-mediated effects are more heterogeneous. Together, these results unify additive superposition with context-dependent attention reweighting into a mechanistic, testable account of how few-shot prompts implement tasks.

## 1. Introduction

As one of the most remarkable capabilities of Large Language Models (LLMs), in-context learning (ICL) (Brown et al., 2020; Wei et al., 2022) enables models to adapt to and perform new tasks at inference time using only a few demonstrations: given a small number of demonstrations, an LLM can generalize to novel test inputs, a capability that has been leveraged in diverse applications such as reasoning (Zhou et al., 2023), translation (Agrawal et al., 2023), and self-correction (Pourreza & Rafiei, 2023). This paradigm, which obviates the need for parameter updates or task-specific fine-tuning, offers exceptional flexibility and efficiency. Consequently, understanding the underlying mechanisms of ICL and enhancing its effectiveness has become a central focus of research in LLM.

Recent work has established the *function vector* (FV) (Todd et al., 2024; Hendel et al., 2023) as a causal mechanism underlying ICL: when performing ICL, models carry an explicit, compact representation of the input-output mapping as a direction in activation space representing the task. FVs are operationalized as the mean activations of a set of **Function Vector heads**, localized by causal mediation and patching (Hendel et al., 2023; Todd et al., 2024; Yin & Steinhardt, 2025; Davidson et al., 2025; Jiang et al., 2025). When activations from these heads are erased, in-context learning collapses; when the aggregation of these activations is injected into a model's residual stream, it can cause a next-token prediction in the task's output space. Importantly, the FV remains effective on new contexts and novel test inputs, showing that it constitutes a causal, task-specific representation.

Recent circuit-level work (Cho et al., 2025; Kharlapenko et al., 2025; Bakalova et al., 2025) has begun to unpack how a model forms the task representation from the individual few-shot examples in a prompt. In particular, Bakalova et al. (2025) shows that the process can be viewed as a two-stage strategy: Lower layers enrich the representations of individual few-shot examples with information from preceding examples (**contextualization**). Middle layers then aggregate the representations of individual examples into a single representation (**aggregation**) driving the prediction for the query. However, the details of the aggregation process, and the function of contextualization, remain unclear. Overall, despite a lot of recent interest, it is still unknown how the FV is composed from the individual few-shot examples.

In this work, we present a prompt-level, causal account of how few-shot examples form and refine an FV during in-context learning. Concretely, our main contributions are:

[1]Saarland Informatics Campus, Saarland Univesity, Saarbrücken, Germany. Correspondence to: Entang Wang <ewang@lst.uni-saarland.de>, Michael Hahn <mhahn@lst.uni-saarland.de>.

*Proceedings of the 43rd International Conference on Machine Learning*, Seoul, South Korea. PMLR 306, 2026. Copyright 2026 by the author(s).

(a) Linear Superposition

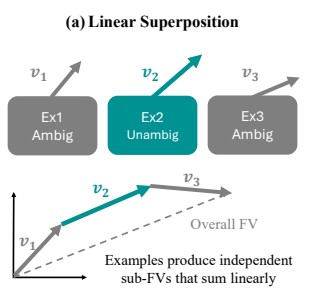

(b) Attention Reweighting Mechanism

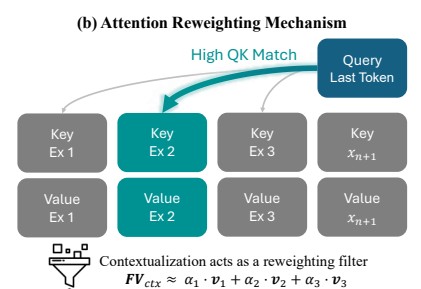

(c) Geometric Vector Refinement

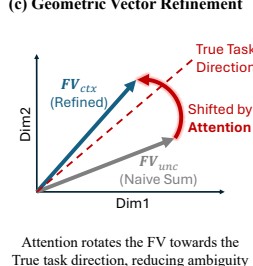

*Figure 1.* **Overview of FV formation in in-context learning.** (a) **Linear superposition:** the $n$-shot function vector (FV) is approximated as a weighted sum of example-level sub-FVs. (b) **Attention reweighting:** contextualization modulates attention over demonstrations, changing the weights assigned to different sub-FVs. (c) **Geometric refinement:** contextualization further improves FV quality by altering the FV direction via pathway-specific updates, most consistently through Query–Key routing.

1. An $n$-shot FV can be approximated, both representationally and causally, as a linear superposition of per-example sub-FVs mediated by attention of the FV heads. This establishes a prompt-level account of how few-shot examples combine to form a task representation. (Section 3)

2. FV heads allocate more attention to examples that unambiguously reveal task identity. This attention pattern is further amplified by *contextualization*, that is, few-shot examples obtain information about prior examples via attention. (Section 4)

3. We causally decompose the effects of both attention mechanism pathways QK&V and verify that this enhancement of attention is causal to improving FV quality (i.e., injection accuracy). (Section 5)

4. In ambiguous contexts, FV quality is primarily driven by information from the ICL examples, which improves the alignment between the Query of the last token and the Keys of unambiguous examples and thereby increases selective attention toward unambiguous examples. (Section 6)

## 2. Background and Preliminaries

### 2.1. In-Context Learning

We study a family of tasks $\mathcal{T}$. Each task $t \in \mathcal{T}$ specifies a mapping $g_t : \mathcal{X} \to \mathcal{Y}$, where $\mathcal{X}, \mathcal{Y} \subseteq \mathcal{V}$ and $\mathcal{V}$ is the model vocabulary. An $n$-shot in-context prompt for task $t$ concatenates $n$ demonstrations followed by an *ICL query*:

$$p_n^{(t)} = \big((x_1, y_1), \ldots, (x_n, y_n), x_{n+1}\big)$$

where $y_i = g_t(x_i)$. Following standard practice, we format prompts with separators; specifically, as in Bakalova et al. (2025), we use \t between $x_i$ and $y_i$ and \n between examples:

$$\underbrace{x_1 \;\backslash\text{t}\; y_1 \;\backslash\text{n}}_{Ex_1} \; \cdots \; \underbrace{x_n \;\backslash\text{t}\; y_n \;\backslash\text{n}}_{Ex_n} \; \underbrace{x_{n+1}}_{Q} \; \underbrace{\backslash\text{t}}_{t_{n+1}}$$

where $\{Ex_1, \ldots, Ex_n, Q, t_{n+1}\}$ are the *components* of the prompt. We use *0-shot* to denote prompts containing only the query and the separator \t, i.e., $p_0 = (x_1 \backslash\text{t})$. Let $f_\theta$ be a pretrained language model that maps a prompt to a next-token distribution. The predicted answer is

$$\hat{y}_{n+1} = \arg\max_{y \in \mathcal{V}} f_\theta(y \mid p_n^{(t)}).$$

with correct answer $y_{n+1} = g_t(x_{n+1})$.

### 2.2. Function Vectors and FV Heads

For each model and task, we localize a set $\mathcal{A}_{\text{FV}}$ of *function-vector (FV) heads* using causal mediation closely following prior work (Todd et al., 2024; Hendel et al., 2023; Yin & Steinhardt, 2025) (Appendix E.2). Unlike prior work that defines the FV by averaging across many prompts, we consider **per-prompt** function vectors $v_{\text{FV}}(p)$. Given a prompt $p_n^{(t)}$, we define its FV as the sum of FV-head activations at the last token:

$$v_{\text{FV}}(p_n^{(t)}) = \sum_{a \in \mathcal{A}_{\text{FV}}} a\big(p_n^{(t)}, t_{n+1}\big),$$

where $a\big(p_n^{(t)}, t_{n+1}\big)$ denotes the output of head $a$ at the final separator position $t_{n+1}$ following $x_{n+1}$ on the prompt $p_n^{(t)}$.

**FV injection and evaluation** To test whether $v_{\text{FV}}(p_n^{(t)})$ reinstates the task on a 0-shot prompt $\tilde{p}_0^{(t)}$, we inject it into the residual stream at FV-head output positions:

$$h_{n+1}^{(\ell)} \leftarrow h_{n+1}^{(\ell)} + \alpha \, v_{\text{FV}}(p_n^{(t)}),$$

where $\ell$ is an injection layer and $\alpha$ is a scaling factor. We evaluate the injected model $f_\theta^+$ by accuracy on a 0-shot prompt set $\tilde{P}_t$, with

$$\hat{y}_{n+1,i}^+ = \arg\max_{y \in \mathcal{V}} f_\theta^+\big(y \mid \tilde{p}_{0,i}^{(t)}\big),$$

and

$$\text{Acc}^{(\ell,\alpha)}(v_{\text{FV}}) = \frac{1}{|\tilde{P}_t|} \sum_i \mathbf{1}\left[\hat{y}_{n+1,i}^+ = \tilde{y}_{n+1,i}\right].$$

We quantify the quality of the FV using a single scalar quality score obtained via a layer/scale sweep:

$$\text{Acc}_{\max}(v_{\text{FV}}) = \max_{\ell \leq L', \, \alpha \in \mathcal{A}} \text{Acc}^{(\ell,\alpha)}(v_{\text{FV}}),$$

where $L'$ and $\mathcal{A}$ follow a model-specific sweep protocol (Appendix E.2).

### 2.3. Contextualized and Uncontextualized Model

Bakalova et al. (2025) show that two kinds of information flow are causally important in ICL: between examples (*contextualization*) and to the final position $t_{n+1}$ (*aggregation*). In particular, the FV heads, moving information from the prompt to $t_{n+1}$, are involved in the *aggregation* step. However, the function of the *contextualization* step, and how it helps FV composition, remains underspecified. In order to cleanly isolate the role of contextualization, we introduce an *uncontextualized* ablation via attention *edge ablation* (Appendix E.3).

In the **uncontextualized** setting, we (i) keep attention within prompt components, and (ii) preserve attention into the last token, but (iii) zero out attention between tokens belonging to different prompt components. An intuitive masking construction process is provided in Appendix E.3 Fig. 10. Correspondingly, the intact model with full cross-component interactions is referred to as **contextualized**. This intervention provides a counterfactual baseline that isolates the causal contribution of cross-example interactions: it eliminates cross-example interaction, so differences between contextualized and uncontextualized runs can be attributed to contextualization rather than to changes in per-example encoding or the aggregation step. Intuitively, *uncontextualized* forces prompt components to be informationally isolated, while *contextualized* allows examples to incorporate information from earlier examples through attention.

### 2.4. Experiment Setup

We evaluate across a series of **tasks** (Appendix C) and **n-shot** (3/5/10) settings on different **model families** (Appendix D). Unless otherwise specified, all results presented in the main text are based on the `gemma-2-2b` model, accompanied by pointers to Appendix sections providing corresponding results for other models.

## 3. An n-shot FV can be approximated as a linear superposition of sub-FVs of individual examples

Our first finding is that the FV of an $n$-shot prompt decomposes approximately into a linear superposition of per-example contributions, *with weights independent of the prompt*. To extract a per-example sub-FV, we mask attention so that $t_{n+1}$ can only attend to the $i$-th example and to the ICL query $x_{n+1}$. The FV extracted under this restricted attention graph is denoted $v_i$. We then fit the full prompt FV $v_{\text{FV}}$ using **ordinary least squares (OLS)**:

$$v_{\text{FV}} \approx \sum_{i=1}^{n} w_i v_i + \varepsilon,$$

Crucially, $w_i$ is fitted globally **across a batch of prompts**. It thus describes the aggregate contribution of each position across prompts. As shown in Figure 2, the OLS-predicted FVs match the observed FVs closely in both contextualized (**ctx**) and uncontextualized (**unc**) settings (mean cosine $\geq 0.925$, mean $R^2 \geq 0.875$). This finding demonstrates that each example contributes an independent task representation, combining linearly into the global task direction. We confirm that this additive structure is non-trivial via a null hypothesis test in Appendix F.

We further test whether the same linear reconstruction preserves the causal task effect of the full FV. Specifically, we inject the OLS-reconstructed FV $\hat{v}_{\text{FV}} = \sum_i w_i v_i$ into 0-shot prompts and compare its injection accuracy with that of the true full-prompt FV. Figure 3 reports the ratio $\text{Acc}_{\max}(\hat{v}_{\text{FV}})/\text{Acc}_{\max}(v_{\text{FV}})$ across tasks. The reconstructed FV recovers most of the full FV's causal effect in the main contextualized setting, and achieves substantial recovery in the uncontextualized setting as well. Thus, the linear approximation is not only geometrically accurate, but also captures much of the causal task-relevant effect of the full FV. To probe robustness on many-shot prompts, we conduct a preliminary **20-shot** check on two representative tasks (**CC** and **PP-A**) using `gemma` models, finding that the linear properties remain clearly evident both at the representational and causal levels (Appendix F Table 4).

Importantly, while the additive form remains stable, we observe that the fitted mixture weights $\{w_i\}$ (Appendix Tab. 5) can shift substantially between the uncontextualized and contextualized settings, indicating that contextualization modulates *how much* each example contributes. Since FV formation at the last token is implemented through attention-based aggregation at FV heads, a natural hypothesis is that these weight shifts are reflected in–and potentially driven by–changes in FV-head attention allocation over examples. In the next section, we therefore study how FV-head attention is distributed across examples under uncontextualized

versus contextualized computation, and what mechanisms control this reweighting.

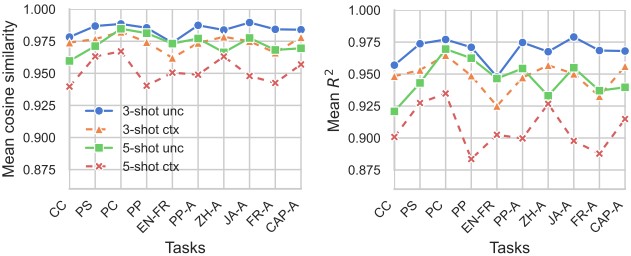

*Figure 2.* **Left:** mean cosine similarity between the observed FV and the OLS reconstruction with/without contextualization (**ctx/unc**). **Right:** mean $R^2$ of the same fit. See Appendix F Fig. 11 for the complete results across all evaluated models.

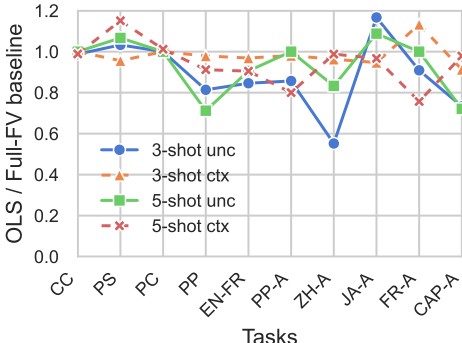

*Figure 3.* **Causal reconstruction of linear superposition.** We inject OLS-reconstructed FVs, $\hat{v}_{\text{FV}} = \sum_i w_i v_i$, into 0-shot prompts and report their maximum injection accuracy relative to that of the true full-prompt FV. Values close to 1 indicate that the linear reconstruction recovers most of the full FV's causal effect. Results are shown across tasks for 3/5-shot prompts under contextualized (**ctx**) and uncontextualized (**unc**) settings.

# 4. Contextualization amplifies Query–Key alignment toward unambiguous examples

Since FV formation is implemented through attention-based aggregation at FV heads, we now ask how attention is allocated across examples, and whether contextualization changes this allocation through the Query–Key (QK) interface.

Attention on FV heads is often dominated by stable structural effects such as recency and positional bias (Fig. 5a), making content-dependent variation difficult to observe.

**Ambiguous Prompt Datasets**    To expose whether *semantic* factors can systematically reshape attention beyond recency, we introduce **ambiguous prompts** that contain both *unambiguous* examples–which uniquely identify the underlying input–output mapping–and *ambiguous* examples which are compatible with multiple candidate mappings

(Appendix C Tab. 1). This setting serves as a stress test, probing whether FV heads can resolve information conflict and selectively attend to the most informative demonstrations when forming the task representation, thereby amplifying attention differences that are otherwise subtle in clean prompts. To mitigate positional bias, for each ambiguous dataset at each $n$-shot setting we apply two positional settings by permuting the placement of ambiguous and unambiguous examples. Within ambiguous prompts, we fix the ambiguous: unambiguous example ratio at $2 : 1$. This ratio places the task in an identifiable but non-trivial regime: there are enough ambiguous demonstrations to induce genuine information conflict (avoiding a near-clean prompt), while retaining a moderate number of unambiguous examples so that FV injection accuracy is neither saturated nor collapsed (reducing ceiling/floor effects). Empirically, this also makes selective attention shifts toward the unambiguous examples more pronounced and hence easier to measure.

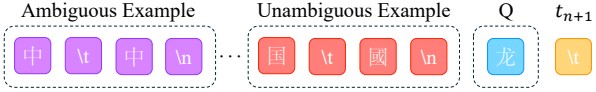

*Figure 4.* A demonstration of an ambiguous prompt. The ambiguous example is compatible with both character transformation and identity mapping.

## 4.1. Unambiguous examples are intrinsically more attractive

Attention to examples is governed by Query–Key similarity: for a fixed last-token Query, higher attention to examples implies stronger QK alignment with their Keys. To isolate the role of per-example Keys, we first study the *uncontextualized* ablation (Sec 2.3), where cross-example attention is removed so that examples cannot contextualize one another and retain prompt-independent representations.

On normal datasets (i.e., datasets without ambiguous examples), FV-head attention is largely explained by recency bias: examples are treated as approximately exchangeable, and later examples receive higher weight (Fig. 5a). In contrast, on ambiguous datasets, the attention distribution becomes content-sensitive: FV heads systematically assign higher attention to *unambiguous* examples than to ambiguous ones, suggesting that unambiguous examples are intrinsically more attractive at the level of their Keys (Fig. 5b). We confirm this via a symmetric *Key patching* intervention (Appendix Fig. 12).

**Q/K/V Patching**    We use Q/K/V patching to isolate how attention **Query**, **Key**, and **Value** channels affect FV formation. We first run a *source* (possibly contextualization ablated) dataset to record $(\mathbf{q}_{\text{src}}^{\mathcal{H}}, \mathbf{k}_{\text{src}}^{\mathcal{H}}, \mathbf{v}_{\text{src}}^{\mathcal{H}})$ of target token positions on *all attention heads*, then run the *target* prompt

and replace some of $\mathbf{q}$, $\mathbf{k}$, and $\mathbf{v}$ with the recorded values:

$$f_\theta\left(p_n^{(t)} \mid \mathbf{q}^{\mathcal{H}}, \mathbf{k}^{\mathcal{H}}, \mathbf{v}^{\mathcal{H}} := \mathbf{q}_{\text{src}}^{\mathcal{H}}, \mathbf{k}_{\text{src}}^{\mathcal{H}}, \mathbf{v}_{\text{src}}^{\mathcal{H}}\right) \rightarrow v_{\text{FV}}.$$

Unless stated otherwise, Q patching replaces only the last token's query vector, while K/V patching replaces only the keys/values used in the last token's attention update; full operational details are deferred to Appendix E.3.

In this subsection, we instantiate this framework in a Keys-only form to test whether the preference for unambiguous examples is already encoded in the example-side Keys themselves. We keep the prompt and in particular the last token's Query fixed, while modifying only example-side Keys on all attention heads. We run two directional variants: (i) replace ambiguous examples' Keys with Keys taken from unambiguous examples drawn from other prompts of the same task; and (ii) replace unambiguous examples' Keys with ambiguous Keys. Replacing ambiguous Keys removes the unambiguous preference and restores a largely position-driven allocation; conversely, overwriting unambiguous Keys yields an analogous effect (Fig. 6).

### 4.2. Contextualization sharpens QK alignment and amplifies attention concentration

Next, we study how *contextualization* reshapes Query–Key alignment by comparing the *contextualized* model against the *uncontextualized* model.

On normal datasets, we find that contextualization primarily acts as a mechanism for positional re-balancing that scales with the context length. While the inherent recency bias remains dominant in low-shot regimes (e.g., 3-shot) where we often observe negligible backward FV attention mass center shifts, the forward-shifting effect consistently emerges and intensifies as the number of demonstrations increases. By the 10-shot setting, a robust negative shift in the FV attention mass center ($\Delta\mathcal{C} \approx -0.4$ to $-0.6$) is observed across nearly all normal tasks (Appendix Tab. 6), indicating a systematic reallocation of attention mass toward earlier examples. To quantify the smoothness of attention across $n$-shot examples, we define the **normalized attention entropy** as $\hat{H} = \frac{-\sum p_i \log p_i}{\log n} \in [0, 1]$, where $p_i$ is the normalized FV attention mass on the $i$-th example. Crucially, this redistribution occurs without sacrificing distributional smoothness: the normalized entropy remains high ($\hat{H} > 0.95$, $\Delta\hat{H} \approx 0$), confirming that contextualization mitigates recency bias by flattening the attention profile rather than inducing sharp, example-specific peaks (Fig. 7a).

In contrast to the smoothing observed in normal datasets, ambiguous tasks exhibit a distinct qualitative divergence in attention dynamics. We find that contextualization in these settings induces a consistent and significant entropy reduction (averaging $-0.08$ to $-0.15$ from 3-shot to 10-shot, Appendix Tab. 6), suggesting it **sharpens** the attention allocation by concentrating attention on unambiguous examples while down-weighting ambiguous ones (Fig. 7b). This pattern effectively amplifies the model's selection of the most informative demonstrations under conflict, turning contextualization into a *selection* mechanism rather than merely redistributing mass across positions. The Shared Query–Key PCA on the Top-1 FV head gives an intuitive visualization of the underlying shift, showing a clear separation between ambiguous and unambiguous Keys after contextualization (Appendix H Fig. 38).

## 5. How does contextualization improve FV quality? A causal decomposition into Query–Key and Value pathways

Prior work shows that contextualization can improve ICL performance, especially under an ambiguous context (Bakalova et al., 2025), but left open *how* contextualization benefits ICL. In the former sections, we established that contextualization sharpens FV-head attention, concentrating mass on unambiguous demonstrations. This suggests an intuitive causal hypothesis: contextualization improves FV quality by improving QK-mediated routing (i.e., attention allocation) toward informative examples. However, contextualization could also improve FV quality by changing the representational content aggregated at FV heads by modifying the per-example Values. To disentangle these two direct causal conduits on FV heads, we design a controlled two-factor intervention that independently toggles contextualization in the QK and V pathways (Appendix Fig. 39).

$2 \times 2$ **factorial intervention over QK and V** We treat maximum FV injection accuracy as a function $F(QK, V)$, where $QK \in \{\text{unc}, \text{ctx}\}$ denotes whether the Query and Key activations are taken from the uncontextualized ablation model or the contextualized intact model, and $V \in \{\text{unc}, \text{ctx}\}$ does the same for Value activations. Writing $\text{unc} = 0$ and $\text{ctx} = 1$, we evaluate four configurations:

1. **Uncontextualized** ($QK_{\text{unc}} + V_{\text{unc}}$): extract FVs from the uncontextualized model, yielding $F(0, 0)$.

2. **Uncontextualized QK + Contextualized V** ($QK_{\text{unc}} + V_{\text{ctx}}$): keep QK from uncontextualized run but patch in contextualized Values, yielding $F(0, 1)$.

3. **Contextualized QK + Uncontextualized V** ($QK_{\text{ctx}} + V_{\text{unc}}$): keep QK from contextualized run but patch in uncontextualized Values, yielding $F(1, 0)$.

4. **Contextualized** ($QK_{\text{ctx}} + V_{\text{ctx}}$): extract FVs from the intact model, yielding $F(1, 1)$.

The overall contextualization gain is $G = F(1, 1) - F(0, 0)$.

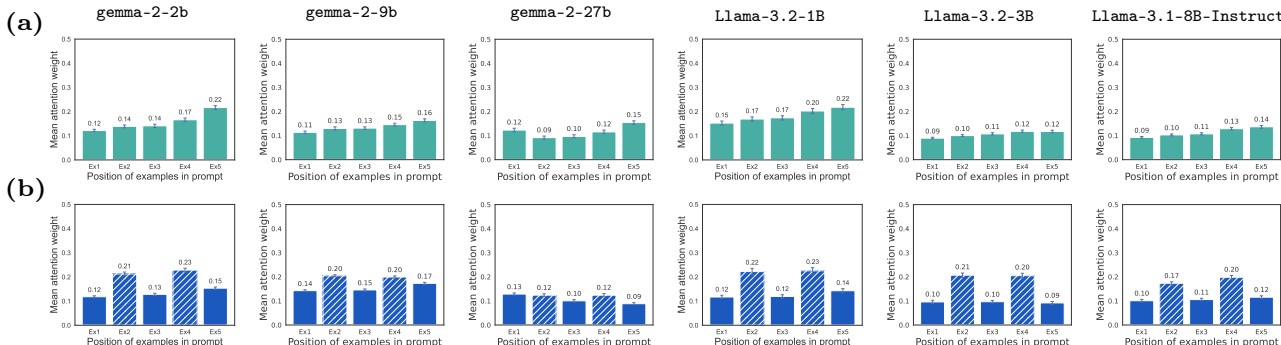

*Figure 5.* **(a)** Mean attention weight of FV heads on a normal task (COUNTRY–CAPITAL), attention is largely explained by recency bias. **(b)** Mean attention weight of FV heads on an ambiguous task (CHINESE AMBIGUOUS), FV heads consistently upweight the *unambiguous* demonstrations (here fixed at **Ex2** and **Ex4**, marked by **diagonal hatching**) relative to ambiguous ones. Each experiment setting is averaged over 100 5-shot prompts. Error bars: 95% CIs from 1000 bootstrap resamples. Results are in the *uncontextualized* setup to isolate intrinsic effects of examples. Attention spikes on unambiguous examples are even larger in the full, *contextualized* setup (Appendix Fig. 32). See Appendix Fig. 13-24 for the complete results.

The contrasts $F(1,0) - F(0,0)$ and $F(1,1) - F(0,1)$ capture the effect of contextualizing QK under two V settings, while $F(0,1) - F(0,0)$ and $F(1,1) - F(1,0)$ analogously capture the effect of contextualizing V under two QK settings (Appendix I).

**Shapley value decomposition** To attribute $G$ to QK and V on equal footing (avoiding an arbitrary choice of intervention order), we use a symmetric two-factor Shapley-style decomposition (Rozemberczki et al., 2022):

$$\phi_{QK} = \tfrac{1}{2}\big(F(1,0) - F(0,0)\big) + \tfrac{1}{2}\big(F(1,1) - F(0,1)\big),$$
$$\phi_{V} = \tfrac{1}{2}\big(F(0,1) - F(0,0)\big) + \tfrac{1}{2}\big(F(1,1) - F(1,0)\big),$$

so that $\phi_{QK} + \phi_{V} = G$ when effects are additive. Intuitively, $\phi_{QK}$ (resp. $\phi_{V}$) is the average marginal improvement from

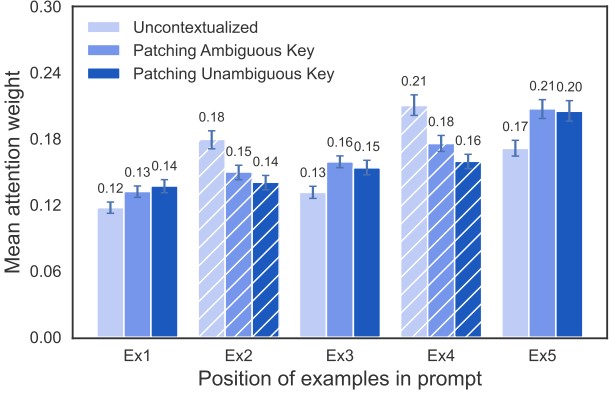

*Figure 6.* Mean attention weight of FV heads under symmetric *Key patching* on all attention heads on PRESENT–PAST AMBIGUOUS dataset. Unambiguous examples are fixed at **Ex2** and **Ex4**, marked by **diagonal hatching**. Each experiment setting is averaged over 100 5-shot prompts. Error bars: 95% CIs from 1000 bootstrap resamples. See Appendix G Fig. 25 for all results.

contextualizing QK (resp. V) across both possible orders in which the two factors could be applied.

Fig. 8 summarizes Shapley contributions averaged across experimental settings (3/5/10-shot; and two positional controls for ambiguous datasets), showing that $\phi_{QK}$ is typically larger and more consistently positive than $\phi_{V}$. We further investigate the behavior across configurations in Appendix I. Detailed breakdowns by model, dataset, and setting are reported in Appendix I Tab. 7. Overall, these results provide causal evidence that contextualization improves FV quality primarily by reshaping QK-mediated attention routing toward informative examples, with Value contextualization acting as a selective, sometimes complementary, modulator.

## 6. Dissecting the origins of Query–Key alignment within ICL prompt components

Having established in Sec. 5 that contextualization gains are primarily mediated by the Query–Key (QK) pathway, we next ask *where* this QK alignment signal comes from within the ICL prompt: the examples, or the ICL query $x_{n+1}$? Some prior work suggests the identity of the ICL query $x_{n+1}$ is important in shaping attention in ICL (Yu & Ananiadou, 2024), whereas others suggest it is not the main driver (Bakalova et al., 2025). To disentangle these, we perform *Q patching* on the *contextualized* model with token-replacement corruption, constructing two controlled variants for each task: (i) *examples-only* corruption, where all demonstration segments are replaced by demonstrations from an unrelated task while keeping $x_{n+1}$ fixed; and (ii) $x_{n+1}$-*only* corruption, where $x_{n+1}$ is replaced while keeping the demonstrations fixed. In both cases, we preserve the attention graph and quantify the effect on FV-head routing (attention statistics) and downstream FV injection accuracy.

On normal datasets, we do not observe a single universal

(a)

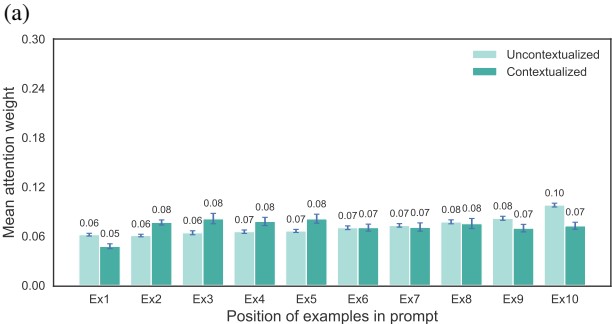

(b)

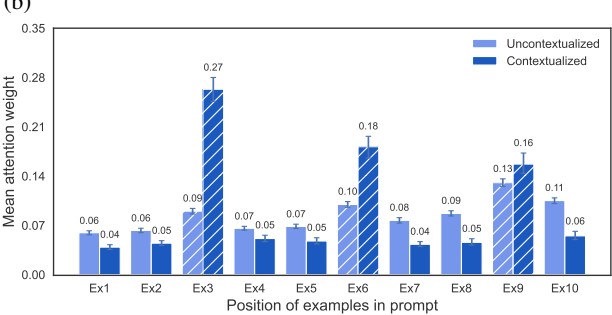

*Figure 7.* Mean attention weight of FV heads in datasets under uncontextualized and contextualized settings. **(a)** On a normal task (PARK–COUNTRY), contextualization primarily mitigates recency bias: attention becomes less back-heavy and the centroid shifts toward earlier demonstrations, without inducing strong example-specific peaks.**(b)** On an ambiguous task (PRESENT–PAST AMBIGUOUS), contextualization instead *sharpens* the allocation, increasing attention concentration on the unambiguous demonstrations (here fixed at **Ex3**, **Ex6**, and **Ex9**, marked by **diagonal hatching**) After contextualization, unambiguous examples share attention increases from 32% (unc) to 61% (ctx). Both experiment settings are averaged over 100 10-shot prompts. Error bars: 95% CIs from 1000 bootstrap resamples. See Appendix H Fig. 26-37 for the complete results across all evaluated models.

alignment driver. Depending on the configuration, we observe situations where corrupting either the examples or $x_{n+1}$ does not induce a clear drop in FV injection accuracy or overall attention on examples, suggesting that either component can provide sufficient information to sustain QK alignment – but also situations where the model asymmetrically relies on one component (Appendix J Fig. 58-63).

On ambiguous datasets, as Fig. 9a, 9b shows, corrupting $x_{n+1}$ and corrupting the examples have qualitatively different effects on FV-head routing. $x_{n+1}$-only corruption primarily affects the *total* attention mass ($T$) allocated to examples: it typically reduces how much attention the model pays to examples overall. In contrast, examples-only corruption primarily disrupts *which* examples are preferred: the attention proportion shifts away from unambiguous examples toward ambiguous ones, even when the total mass changes less dramatically. On most ambiguous tasks, FV injection accuracy is much more sensitive to examples-only

corruption than to $x_{n+1}$-only corruption. The main exception is the CAPITALIZATION–AMBIGUOUS family, where the roles flip: $x_{n+1}$-only corruption more strongly distorts the unambiguous/ambiguous proportion and induces larger FV-accuracy losses (Fig. 9c, 9d), indicating that in this family the query itself carries a decisive disambiguating signal for selective routing. Meanwhile, this exception refuses this natural concern: examples-only corruption modifies many more tokens than $x_{n+1}$-only corruption, so that it might have larger effects by introducing more noise.

In summary, this ablation reveals a sharp contrast: Query–Key alignment in normal tasks is largely task-specific, whereas in most ambiguous settings (excluding capitalization), it is predominantly **example-led**. This suggests that robust in-context learning under ambiguity is driven not merely by query-based retrieval, but by an example-driven reweighting of attention. In Appendix K, we provide an exploratory observation and preliminary discussion on the effects of example content.

## 7. Discussion

**Implications for Theory of ICL** A recent literature has developed theoretical models of transformers performing in-context learning, especially on linear regression tasks, with by now very well developed theoretical understanding (e.g. Mahankali et al., 2024; Von Oswald et al., 2023; Vladymyrov et al., 2024; Akyürek et al., 2023; Zhang et al., 2025, inter alia). Here, the task is defined in terms of a latent vector $w \in \mathbb{R}^d$, and the task maps $x \in \mathbb{R}^d$ to the output $y = w^T x \in \mathbb{R}$. Transformers have been shown to be able to learn to perform ICL for this task family in both theory and experiment. A particularly popular idea is to model linear attention, where a single layer and single head can – in the simplest case – implement the prediction

$$\hat{y}_{n+1} = \eta \sum_{i=1}^{n} y_i x_i^T x_{n+1} \qquad (1)$$

where the (linear) attention head has, for a few-shot example $(x_i, y_i)$ the *key* $x_i$ and the *value* $y_i$; and, at the ICL query $x_{n+1}$, the *query* $x_{n+1}$ This rule can be interpreted as a single step of gradient descent, and is indeed optimal for a single layer of linear attention performing this task (e.g. Mahankali et al., 2024). The rule (1) can also be interpreted in terms of *similarity-based retrieval* of outputs $y_i$ weighted on the basis of the similarity (quantified by the inner product) between inputs $x_i$ and the ICL query $x_{n+1}$. Relatedly, Dragutinović et al. (2025) study *classification tasks* with softmax attention, and find a similar similarity-based (kernel-like) retrieval based on similarity between $x_i$ and the ICL query $x_{n+1}$.

Perhaps surprisingly, contact between this rich theoretical literature considering idealized domains, and mechanistic

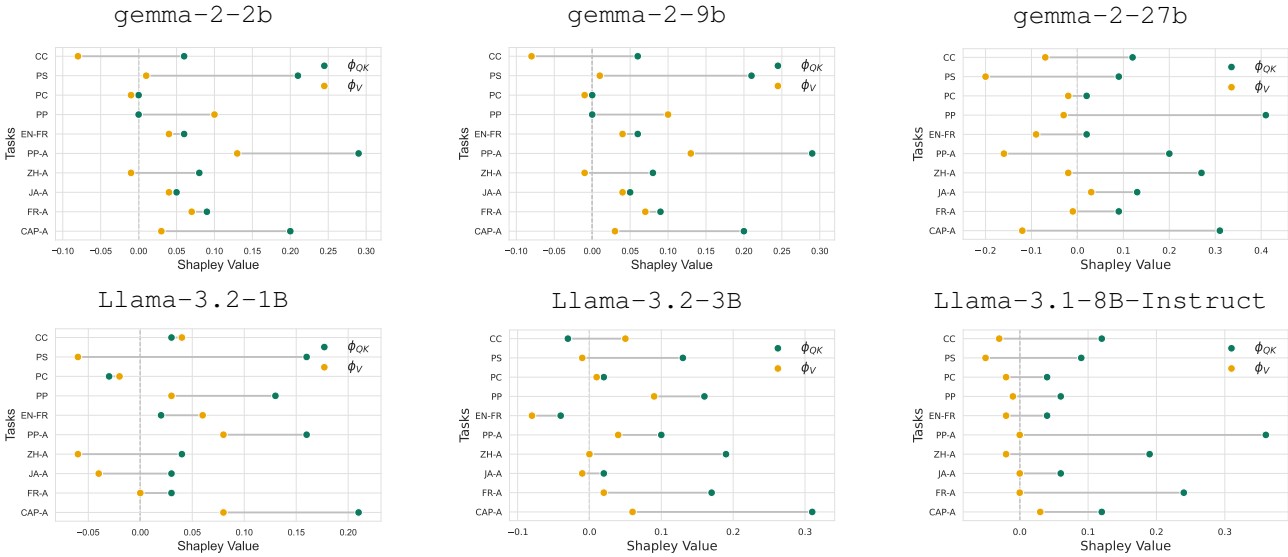

*Figure 8.* Shapley value decomposition ($\phi$) averaged over experiment configurations ($n$-shots, positional controls) on `gemma-2` and `Llama-3` model families. Contextualizing QK usually has a stronger positive effect on FV quality, compared to contextualizing V. See Appendix I Fig. 40-51 for details.

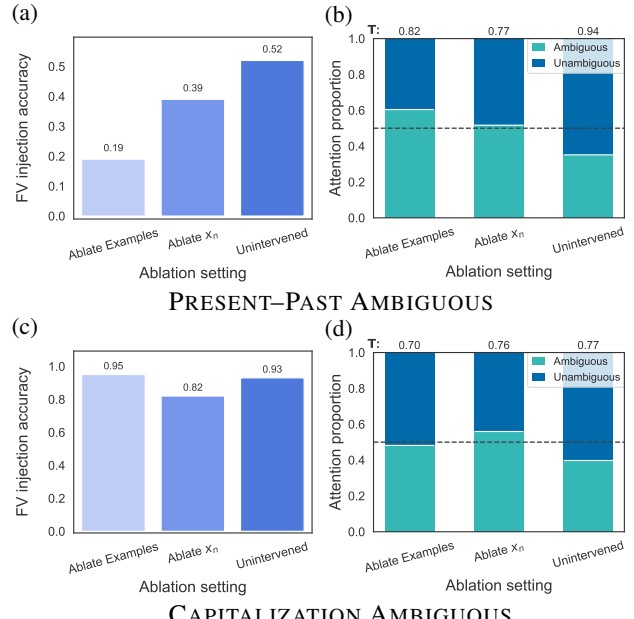

*Figure 9.* Query composition and its effect on QK alignment for ambiguous tasks. Panels **(a, c)** report the FV injection accuracy, while **(b, d)** show the attention proportion on unambiguous vs. ambiguous examples. The value **T** above each bar indicates the **total attention mass** allocated to all example segments. Results are averaged over 100 trials using 10-shot prompts. See Appendix J Fig.52-57 for the complete results.

studies of ICL in real-world LLMs has so far been limited. As one example, whereas theoretical constructions for ICL often assume that attention favors examples *based on match of $x_i$ to the ICL query $x_{n+1}$*, this view contrasts with the finding that LLM FV heads assign attention *based on how much an example $(x_i, y_i)$ disambiguates task identity*. Theoretical studies often target continuous parametric tasks such as linear regression, quite different from the discrete tasks mostly studied in the mechanistic literature on LLMs. In Appendix M, we show that a simple theoretical model that (i) accounts for nonlinear MLPs and softmax attention found in real LMs, rather than linearized transformers, and (ii) assumes a discrete space of tasks operating on discrete input spaces, accounts for some of our key mechanistic findings about LLMs: Function vectors are a linear superposition of per-example vectors, independent of $x_{n+1}$, but assigning higher weight to examples that are informative about task identity (see Theorem M.1).

**Unifying prior mechanistic views of ICL** Our finding that prompt-level FVs are well approximated by additive combinations of example-level sub-FVs, with contribution weights modulated by attention alignment, suggests a potential unifying perspective on several previously distinct mechanistic views of ICL. **(1) Reconciling additive superposition and attention reweighting** First, our results help reconcile the *additive superposition* perspective, often associated with induction circuit mechanisms (Olsson et al., 2022), with the *context-dependent attention reweighting* perspective found in retrieval-based accounts (Min et al., 2022; Abernethy et al., 2024; Yu & Ananiadou, 2024). These two views need not be mutually exclusive within the FV-formation pathway: FV-head attention can selectively route information through Query–Key alignment, while the resulting prompt-level FV can still be well approximated as a linear combination of example-level sub-FVs. In this

sense, attention reweighting controls which sub-FVs dominate the composition, while additive superposition describes how these routed signals combine into the extracted FV. **(2) Relating to superposition phenomena** Second, our framework offers a plausible mechanistic perspective on recently observed phenomena such as *task superposition* (Xiong et al., 2025) and *instruction-demonstration complementarity* (Davidson et al., 2025). Within the identified FV heads, example-level sub-FVs are aggregated into the prompt-level FV by a strong linear approximation, rather than by an obviously winner-take-all process at the FV-formation stage. This does not rule out nonlinear interactions or competition elsewhere in the model, including downstream execution, residual-stream interactions, or final readout. Instead, it suggests that FV-forming heads may provide a relatively linear interface through which multiple demonstration-derived task signals can be composed before later layers transform them into task behavior. **(3) Interpreting saturation and noise robustness** Third, the stability of the additive approximation across contextualized settings offers a possible interpretation of the **demonstration saturation** effect (Agarwal et al., 2024; Gu et al., 2025; Bertsch et al., 2025). Once a sufficiently strong task direction is formed from informative demonstrations, additional examples may contribute signals that are partially redundant with the existing FV, leading to diminishing marginal gains. Similarly, our findings provide a possible FV-level account of the **label noise robustness** reported by Min et al. (2022): contextualization can reduce the effective contribution of less informative or ambiguous examples through QK-mediated reweighting, while the subsequent FV aggregation may average residual noise in the task representation. We emphasize that these interpretations are suggestive connections to prior phenomena rather than complete explanations of the full ICL computation.

Together, these results point toward FV superposition as a useful bridge between circuit-level and representational accounts of ICL. Within the extracted FV representation, few-shot examples can be understood as contributing approximately additive task-relevant directions, whose effective weights are modulated by context-sensitive attention routing. This provides a bounded, testable account of FV formation, while leaving open how the resulting task representation interacts with downstream nonlinear computation and broader mechanisms of ICL.

## 8. Conclusion

We presented a prompt-level, intervention-based account of how few-shot demonstrations form function vectors (FVs). Unlike prior aggregate analyses, we isolate the *within-prompt* mechanism, characterizing how example-level signals compose and how contextualization modulates this process. Our findings support three primary conclusions.

First, $n$-shot FVs are well-approximated by a near-linear superposition of example-level sub-FVs. This additive structure remains robust before and after contextualization. Second, contextualization acts as an adaptive reweighting mechanism: attention routing shifts to emphasize informative demonstrations that resolve task identity, while preserving the underlying additive template. Third, causal decomposition reveals a functional divergence: Query–Key alignment consistently improves FV quality, especially under ambiguity, whereas Value-mediated refinement is heterogeneous–varying from beneficial to detrimental despite inducing coherent geometric shifts. We bridge retrieval-style accounts (selective routing) with representation-based views (additive superposition) by showing they are complementary facets of the same task-representational circuit.

**Limitations and future directions** (1) Our conclusions are bounded by the FV framework and by the FV-head localization procedure used to extract task vectors. (2) Our study focuses on discrete mapping toy tasks; extending this framework to long-horizon or compositional reasoning remains a key challenge. (3) Value effects are still unclear; characterizing when Value updates help versus harm, and whether training-time objectives can regularize this behavior, is an important direction, on which we provide some initial exploratory observations in Appendix L. (4) While we operationalize "ambiguity" through controlled task constructions, future work could benefit from introducing formal scalar metrics to enhance cross-task comparability.

## Impact Statement

This paper presents work whose goal is to advance the field of machine learning. There are many potential societal consequences of our work, none of which we feel must be specifically highlighted here.

## Acknowledgments

Funded by the Deutsche Forschungsgemeinschaft (DFG, German Research Foundation) – GRK 2853/1 "Neuroexplicit Models of Language, Vision, and Action" - project number 471607914.

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

## A. FAQ

1. *Why an uncontextualized ablation?*

   The *uncontextualized* ablation provides a counterfactual baseline that isolates the causal role of *cross-example interactions*. By masking cross-component attention edges, differences to the contextualized setting can be attributed to contextualization (information flow across components) rather than to local encoding or readout changes. Because within-example computation is preserved, each example keeps its intrinsic representation, while the aggregation mechanism remains comparable. This also enables clean sub-FV extraction and additivity tests (Sec. 3) and supports pathway attributions separating QK *reweighting* from V *representation changes* (Sec. 5). Intuitively, *uncontextualized* isolates components; *contextualized* allows earlier examples to shape later ones via attention.

2. *The function vector is the output of attention, which by definition is a linear combination. Why are the findings about linear superposition in Section 3 then nontrivial?*

   Conditioned on fixed attention weights, each head output is linear in $V$. But the weights themselves are *prompt-dependent* nonlinear functions via $\mathrm{softmax}(QK^\top)$, and $Q$ is produced by multi-layer contextual processing (with residual paths and nonlinearities). Thus, ablating demonstrations can change $Q/K$ geometry and globally reconfigure attention, so additivity is not guaranteed. Our test is further nontrivial because we fit a *global* shared ridge-regression weight vector across many prompts (not per-prompt fitting). Thus, a linear fit is nontrivial because the attention weights from $\mathrm{softmax}(QK^\top)$ are in general prompt-dependent.

   We further compare against null baselines (mismatched dictionaries, orthogonalized sub-FVs) that preserve marginals and the geometry while breaking directionality; the real-vs-null gap indicates structured example-level superposition rather than a degenerate linear artifact.

3. *Why do we need a symmetric Key patching?*

   Bidirectional (symmetric) Key swaps rule out a trivial *content-mismatch* account where attention differs only because unpatched examples happen to better match the last-token Query. Swapping Keys in both directions (unambiguous ↔ ambiguous) while holding other factors fixed isolates Keys as the causal driver: unambiguous Keys yield higher similarity to the last-token Query and attract attention even without contextualization, while disrupting this Key structure restores generic recency/positional bias. Symmetry ensures the effect tracks Key structure, not example identity or an asymmetric perturbation.

4. *The task suite is restricted to classification-style mappings, failing to represent complex behaviors like creative generation or complex algorithmic tasks.*

   We follow relevant prior work to focus on short-horizon, discrete mappings because their computational cost is desirable, and ambiguity can be introduced in a controlled way. We agree that open-ended generation and long-horizon algorithmic tasks are closer to real applications, but we can no longer evaluate on merely the last token; they add a multi-step attention transition when doing autoregressive generation. We therefore view richer tasks as a key extension, and present current results as mechanistic foundations established under controlled conditions.

5. *What does "on normal datasets, we do not observe a single universal alignment driver" in Section 6 mean?*

   On normal tasks, QK-alignment does not consistently trace to a single prompt component (examples vs query) across tasks/models. Some tasks show heterogeneous patterns (e.g., PERSON–SPORT, PARK–COUNTRY), while others exhibit *preferential* regimes: COUNTRY–CAPITAL is more query-driven (with $x_{n+1}$-only corruption more reliably reducing example attention and FV injection accuracy), whereas PRESENT–PAST is more example-driven. Overall, normal tasks reflect redundancy or task-/model-specific preference rather than a universal driver.

## B. Further Discussion of Related Work

A growing body of work proposes complementary mechanisms for ICL. **(1) Induction circuits** A mechanistic stream attributes few-shot generalization to *induction heads* (Olsson et al., 2022; Elhage et al., 2021), in which specific attention heads copy patterns from earlier examples to the query and stitch them together across layers. **(2) Retrieval** Retrieval explanation emphasizes query–key geometry: the query attends to example keys with high semantic alignment, and this selective routing–rather than value rewriting–drives performance gains (Min et al., 2022; Liu et al., 2022; Abernethy et al., 2024; Yu & Ananiadou, 2024). **(3) Implicit Bayesian and Optimization** Algorithmic interpretations view ICL as

implicit Bayesian updating or as an in-network optimizer that approximates gradient descent over a hypothesized linear model implemented by attention (Wei et al., 2023; Shi et al., 2024; Xie et al., 2022). **(4) Composition and superposition** Compositionality and superposition perspectives argue that ICL behaviors can combine up, explaining why multi-example prompts can act like additive control signals (Dong et al., 2026; Xiong et al., 2025; Lu et al., 2025; Vladymyrov et al., 2024). **(5) Contextualization and Aggregation** This emerging perspective dives into how individual examples are integrated and calibrated into a unified task representation. Bakalova et al. (2025) shows that the ICL process can be viewed as a two-stage strategy: Lower layers enrich the representations of individual few-shot examples with information from preceding examples (contextualization). Middle layers then aggregate the representations of individual examples into a single representation (aggregation), driving the prediction for the query. Cho et al. (2025) proposes a structured "three-stage circuit"—comprising Input Encode, Semantics Merge, and Feature Retrieval—to explain how the model aggregates disparate demonstration features into a calibrated prediction.

Despite these advances, no single account fully explains the breadth of ICL phenomena observed in practice. Induction circuits illuminate copy-and-continue behaviors yet underpredict cases where examples differ substantially from the query; retrieval-style accounts clarify query–key alignment and format sensitivity but do not directly identify where a task is represented; and composition/ superposition perspectives explain additive trends and multi-task coexistence but abstract away prompt-conditioned causal loci inside the network. Against this backdrop, the function vector view is promising but still nascent. Most FV studies estimate task representations by averaging activations across many prompts and validating them via causal injection. However, this dataset-level approach leaves open the natural and common setting of how a single few-shot prompt forms its own FV, how example and query specific contributions superpose or interact, and how FV relates mechanistically to induction and retrieval pathways. Very recent evidence begins to separate FV heads from induction heads and tries to explain how FV heads form during the training phase. However, the mechanism by which FV heads play a role in single-prompt-level ICL remains poorly understood.

## C. ICL tasks

In this study, we define the In-Context Learning (ICL) process as a mapping function $\mathcal{F} : x \to y$ derived from a few-shot demonstration context. We categorize the tasks into two groups based on the clarity of this mapping:

**Normal Tasks** are characterized by a deterministic and globally consistent transformation rule. In these tasks, the semantic or syntactic relationship between the input $x$ and the output $y$ remains invariant across all examples, providing a clear signal for the model to retrieve specific factual or linguistic knowledge (e.g., mapping a country to its capital).

**Ambiguous Tasks** are designed such that the demonstration context contains a latent mapping space where a single input could plausibly satisfy multiple transformation hypotheses. The specific sources of ambiguity for each task are detailed: **Present-Past Ambiguous (PP-A):** The ambiguity arises from *morphological invariance* in certain English irregular verbs. For verbs such as "let" or "set," the past tense form is orthographically identical to the present tense. This creates a functional conflict between a temporal transformation (tense shifting) and a null transformation, making the intended rule indistinguishable based on such samples. **Chinese Ambiguous (ZH-A):** The ambiguity stems from the *partial orthographic overlap* between Simplified and Traditional Chinese character sets. While the task objective is script conversion (Simplified → Traditional), many characters are "inherited characters" that remain unchanged in both systems. This forces the model to resolve the competition between a script-based transition and literal character preservation. **Japanese Ambiguous (JA-A):** The ambiguity here originates from *cross-lingual task interference*. In this setup, the model is prompted with English-Japanese pairs. However, some Japanese translations are also correct in Chinese. Hence, the ambiguous examples consist of English-Japanese pairs that would also be correct English-Chinese pairs. **French Ambiguous (FR-A):** This task exploits *cross-lingual lexical cognates*–words with identical spelling and meaning across languages. In an English-to-French translation context, tokens such as "menu" exist in both vocabularies. The ambiguity emerges from the competition between a cross-lingual translation mapping and a mono-lingual persistence mapping within the shared embedding space. **Capitalize Ambiguous (CAP-A):** The source of ambiguity is the *pre-training priors* associated with proper nouns. When the task is to perform a lowercase-to-capitalized transformation, words that are inherently capitalized in the pre-training corpus (e.g., "Jupiter") satisfy the output requirement without undergoing any change. This creates a conflict between the task's formatting rule and the tokens' natural distribution in the training data.

| Task Name | Task Abbreviation | Example | Task Source |
|---|---|---|---|
| **Normal Task** | | | |
| COUNTRY-CAPITAL | **CC** | China: Beijing | Todd et al. (2024) |
| PERSON-SPORT | **PS** | Lionel Messi: soccer | Hernandez et al. (2024) |
| PARK-COUNTRY | **PC** | Talampaya National Park: Argentina | Todd et al. (2024) |
| PRESENT-PAST | **PP** | swim: swam | Todd et al. (2024) |
| ENGLISH-FRENCH | **EN-FR** | west: ouest | Conneau et al. (2018) |
| **Ambiguous Task** | | | |
| PRESENT-PAST AMBIGUOUS | **PP-A** | let: let; make: made | Todd et al. (2024) |
| CHINESE AMBIGUOUS | **ZH-A** | 堂: 堂 卫: 衛 | new task |
| JAPANESE AMBIGUOUS | **JA-A** | university: 大学 onion: 玉ねぎ | new task |
| FRENCH AMBIGUOUS | **FR-A** | menu: menu; soldier: soldat | Conneau et al. (2018) |
| CAPITALIZE AMBIGUOUS | **CAP-A** | Jupiter: Jupiter; indigo: Indigo | Nguyen et al. (2017) |

*Table 1.* Summary of ICL tasks. **Normal Tasks** follow a consistent transformation rule. **Ambiguous Tasks** introduce mapping uncertainty through different mechanisms: *morphological invariance* (PP-A), *orthographic overlap* in scripts (ZH-A) or cognates (FR-A), *pre-training priors* on proper nouns (CAP-A), and *cross-lingual task interference* (JA-A) where the target output is valid in multiple language contexts. For each ambiguous task, we show both an *ambiguous* (e.g., let: let in the PP-A task) and an *unambiguous* (e.g., make: made in the PP-A task) example. See Appendix C for definitions.

## D. Models used

| Model (Huggingface ID) | $|L|$ | $|a|$ | $d_{model}$ |
|---|---|---|---|
| google/gemma-2-2b | 26 | 8 | 2304 |
| google/gemma-2-9b | 42 | 16 | 3584 |
| google/gemma-2-27b | 46 | 32 | 4608 |
| meta-llama/Llama-3.2-1B | 16 | 32 | 2048 |
| meta-llama/Llama-3.2-3B | 28 | 24 | 3072 |
| meta-llama/Llama-3.1-8B-Instruct | 32 | 32 | 4096 |

*Table 2.* Models studied in this work. We use Huggingface implementations for all models. We report the number of layers $|L|$, the number of attention heads per layer $|a|$, and the dimension of hidden states $d_{model}$ for each model.

## E. Methodology details

### E.1. In-Context Learning

In this work, we study a family of tasks $\mathcal{T}$. Each task $t \in \mathcal{T}$ defines a distribution $\mathcal{D}_t$ over input–output pairs $(x, y) \in \mathcal{X} \times \mathcal{Y}$ and a mapping $g_t : \mathcal{X} \to \mathcal{Y}$. Let $\mathcal{V}$ denote the vocabulary of an LLM, with $\mathcal{X}, \mathcal{Y} \subseteq \mathcal{V}$.

An $n$-shot ICL prompt for task $t$ is a concatenation of $n$ example segments followed by a single *ICL query*:

$$p_n^{(t)} = \big((x_1, y_1), \ldots, (x_n, y_n), x_{n+1}\big)$$

where $(x_i, y_i) \sim \mathcal{D}_t$ are the example pairs. The *ICL query* is $x_{n+1}$. Specifically, we use the term *0-shot* to refer to prompts that contain only the ICL query, i.e., $p_0 = (x_1)$. In practice, the input $x_i$ and output $y_i$ are also followed by an input/output separator $t_i, n_i$.

Let $f_\theta$ denote a pretrained model with parameters $\theta$ that maps a prompt to a distribution over the next tokens. Given $p_n^{(t)}$, the model is expected to produce the output $y_{n+1} = g_t(x_{n+1})$ consistent with the task specified by the examples.

$$\hat{y}_{n+1} = \arg\max_{y \in \mathcal{V}} f_\theta(y \mid p_n^{(t)})$$

### E.2. Function Vector

**Localization of FV heads**   We localize FV heads using the *causal mediation analysis* procedure introduced by Todd et al. (2024). For each task $t \in \mathcal{T}$ with prompt set $P_t = \{p_i^{(t)}\}$, we compute each attention head's task-conditioned mean activation $a_{\ell j}$ at the last token $t_{n+1}$:

$$\bar{a}_{\ell j}^t = \frac{1}{|P_t|} \sum_{p_i^{(t)} \in P_t} a_{\ell j}(p_i^{(t)}).$$

Then we create corrupted prompts $\tilde{p}_i^{(t)}$ by randomly shuffling output labels $\tilde{y}_i$ while keeping inputs $x_i$ fixed. Running the model on these uninformative prompts, we replace $a_{\ell j}$ with $\bar{a}_{\ell j}^t$ and measure its *causal indirect effect* (CIE) toward producing the correct answer $y_{iq}$:

$$\text{CIE}(a_{\ell j} \mid \tilde{p}_i^{(t)}) = f_\theta\big(\tilde{p}_i^{(t)} \mid a_{\ell j} := \bar{a}_{\ell j}^t\big)[y_{iq}] - f_\theta\big(\tilde{p}_i^{(t)}\big)[y_{iq}].$$

A larger CIE implies that substituting the "correct" mean activation increases the probability of the target answer under the corrupted prompt, indicating that the head contributes causally to ICL. Each head's average indirect effect (AIE) is obtained by averaging CIE across all corrupted prompts in a specific task:

$$\text{AIE}(a_{\ell j}) = \frac{1}{|\tilde{P}_t|} \sum_{\tilde{p}_i^{(t)} \in \tilde{P}_t} \text{CIE}(a_{\ell j} \mid \tilde{p}_i^{(t)}).$$

Heads with consistently high AIE values across tasks are identified as *FV heads*. The quantity of FV heads is an empirical value which is positively correlated with the overall number of attention heads in the model (Table. 3).

**Extracting per-prompt function vectors**   Whereas previous FV work typically averaged activations across many prompts to obtain a dataset-level task representation, we extract **per-prompt** FVs–capturing how a single few-shot prompt forms its own task vector. For a specific prompt $p_n^{(t)}$, the FV is the sum of all identified FV heads' activations:

$$v_{\text{FV}}(p_n^{(t)}) = \sum_{a \in \mathcal{A}_{\text{FV}}} a\big(p_n^{(t)}, t_{n+1}\big),$$

where $\mathcal{A}_{\text{FV}}$ is the set of FV heads localized via AIE. This per-prompt vector enables us to analyze how example-level contributions superpose linearly and how contextualization alters the final task direction.

**Injection and Evaluation**   To test whether a per-prompt FV is sufficient to re-induce the task, we perform *function vector injection*. Given a 0-shot prompt $\tilde{p}_0^{(t)}$, we add $v_{\text{FV}}(p_n^{(t)})$ into the residual stream at the FV heads' output positions:

$$h_{n+1}^{(\ell)} \leftarrow h_{n+1}^{(\ell)} + \alpha\, v_{\text{FV}}(p_n^{(t)}),$$

where $h_{n+1}^{(\ell)}$ is the hidden state of the last token $t_{n+1}$ at layer $\ell$, and $\alpha$ is a scaling factor controlling the injection strength. After injection, the correct answer $y_{n+1}$ should receive the *highest probability mass* in the output distribution $f_\theta^+(y_{n+1} \mid \tilde{p}_0^{(t)})$, demonstrating that the injected FV successfully reinstates the task function $g_t$ within an otherwise uninformative context. To evaluate the quality of an FV, we follow a systematic injection protocol across layers and scaling factors. First, we inject the same FV sequentially from the input layer ($\ell = 0$) up to approximately one-third of the model's total layers ($L/3$). As shown in Todd et al. (2024), the injection accuracy remains smooth and stable in early layers, reaching a peak around $L/3$, beyond which performance drops sharply as deeper layers encode task-specific decoding states. Second, for each layer $\ell$ and scaling factor $\alpha \in \mathcal{A}$, we measure the accuracy $\text{Acc}^{(\ell,\alpha)}$ using the zero-shot prompt set $\tilde{P}_t$ that randomly samples the ICL query $x_{n+1}$ from the input space. We then compute the **maximum injection accuracy** across all combinations of $(\ell, \alpha)$:

$$\text{Acc}^{(\ell,\alpha)}(v_{\text{FV}}) = \frac{1}{M} \sum_{i=1}^{M} \mathbf{1}\Big[ \arg\max_{y \in \mathcal{V}} f_\theta^+\big(y \mid p_{0,i}^{(t)}\big) = \tilde{y}_{n+1,i} \Big]$$

$$\text{Acc}_{\max}(v_{\text{FV}}) = \max_{\ell \leq L',\, \alpha \in \mathcal{A}} \text{Acc}^{(\ell,\alpha)}(v_{\text{FV}})$$

where $\tilde{y}_{n+1,i}$ denotes the answer of the $i$-th prompt in the test dataset and $L$ denotes the upper layer limit for stable injection. Higher $\text{Acc}_{\max}$ indicates stronger task reinstatement and higher-quality FVs. This layer- and scale-sweep evaluation provides a robust measure of how well a per-prompt FV captures the underlying task function across the model's representational hierarchy.

| Model Name | FV Head Number | Scaled Factor Range | Injection Range |
|---|---|---|---|
| gemma-2-2b | 20 | [1.0, 25.0] | Layer 0: Layer 9 |
| gemma-2-9b | 20 | [1.0, 30.0] | Layer 0: Layer 13 |
| gemma-2-27b | 70 | [1.0, 30.0] | Layer 0: Layer 15 |
| Llama-3.2-1B | 20 | [1.0, 4.0] | Layer 0: Layer 5 |
| Llama-3.2-3B | 50 | [0.5, 4.0] | Layer 0: Layer 19 |
| Llama-3.1-8B-Instruct | 50 | [1.0, 24.0] | Layer 0: Layer 22 |

*Table 3.* Detailed hyperparameters used in FV extraction and injection.

### E.3. Contextualization and Aggregation

**Edge ablation**   The goal of edge ablation is to isolate how each individual example or the ICL query contributes to the overall prompt's task representation. By severing specific communication channels between components (e.g., examples → examples or examples → ICL query), we can measure the causal effect of each component on performance and thus infer its functional role. Concretely, we construct a token-position graph of the prompt where nodes correspond to tokens and directed edges represent attention flows. Following the method of Bakalova et al. (2025), we keep intact all edges within each example component $(x_i, y_i)$ and all edges from prompt components to the last token $t_{n+1}$ but apply an attention ablation mask to zero-out all attention weights between any token in component $i$ and any token in component $j$ (for $i \neq j$), including the ICL query component. We use the term **uncontextualized** (Fig. 10) to refer to this ablation setting.

Let the prompt consist of components (examples and the ICL query) indexed by

$$\mathcal{C} = \{Ex_1, Ex_2, \ldots, Ex_n, \mathbf{Q}\},$$

where component $c_i \in \{Ex_1, \ldots, Ex_n\}$ corresponds to example $(x_i, y_i)$, and $c_i = \mathbf{Q}$ corresponds to the ICL query input $x_{n+1}$. Define an attention graph $G = (V, E)$ at a given attention head $\mathcal{H}$, where each token position is a node $v \in V$, and

$$E = \{(v \to w) : \alpha_{attn}^{\mathcal{H}, (v \to w)} \neq 0 \text{ for attention head } \mathcal{H}\}$$

represents attention edges. We keep all intra-component edges:

$$E_{\text{intra}} = \big\{(v \to w) \mid v, w \in c_i\big\}$$

We then ablate (zero-out) all edges between distinct components, including ICL query by defining:

$$E_{\text{ablated}} = \big\{(v \to w) \mid v \in c_i, \; w \in c_j, \; i \neq j\big\}$$

The ablated model replaces attention score $\alpha_{attn}$:

$$\alpha_{attn}^{\mathcal{H}, (v \to w)} \leftarrow 0, \quad \forall (v \to w) \in E_{\text{ablated}}$$

**Q/K/V patching**   We perform Q/K/V patching to study how the individual components of the attention mechanism contribute to the formation of the function vector and to bridge this with prior findings on linear additive and attention-weight reweighting. Specifically, we first run a "patching dataset" (which may be ablated or intact) to collect the query, key, and value vectors for target token positions. We then run the original prompt dataset, and replace the $\mathbf{q}^{\mathcal{H}}$, $\mathbf{k}^{\mathcal{H}}$, or $\mathbf{v}^{\mathcal{H}}$ vectors with those recorded from the patching dataset on **all attention heads**. By selectively substituting Q, K or V activations, we dissect how each component affects the downstream FV formation. [1]

$$f_\theta(p_n^{(t)} | \mathbf{q}^{\mathcal{H}}, \mathbf{k}^{\mathcal{H}}, \mathbf{v}^{\mathcal{H}} := \mathbf{q}_{\text{patched}}^{\mathcal{H}}, \mathbf{k}_{\text{patched}}^{\mathcal{H}}, \mathbf{v}_{\text{patched}}^{\mathcal{H}}) \to v_{\text{FV}}$$

---

[1]In all Q patching experiments we only replace the Query vector at the last token position. For K/V patching we only replace the Key/Value vectors used in the attention update of the last token (i.e., the keys and values that the last token attends to), while keeping the keys and values used for all other query positions unchanged.

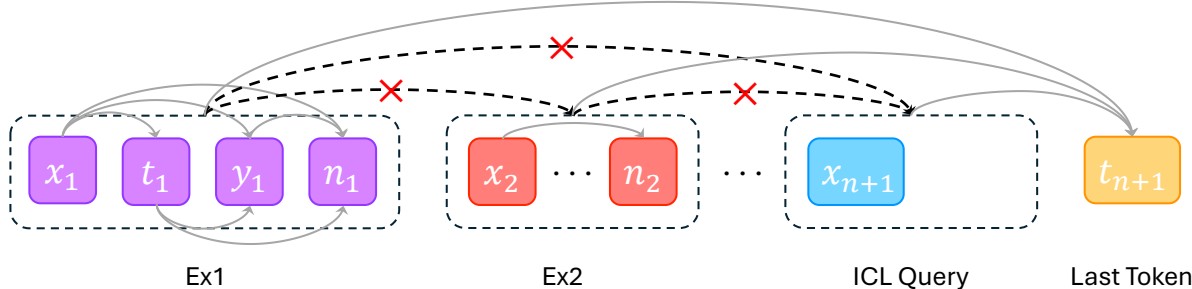

*Figure 10.* A diagram of **uncontextualized** ablation. This diagram illustrates the intervention used to isolate ICL component-level contributions by severing specific attention pathways. We keep all edges within each example and all edges from prompt components to the last token $t_{n+1}$ while zeroing out the rest of the edges.

## F. Linear superposition

To quantify the extent to which an $n$-shot FV can be explained as a linear superposition of per-example sub-FVs, we perform a *global* ordinary least squares (OLS) regression with weights shared across a batch of $B$ prompts. Let $\{v_i^{(b)} \in \mathbb{R}^D\}_{i=1}^n$ denote the sub-FVs extracted for the $b$-th prompt instance, and let $v_{\mathrm{FV}}^{(b)} \in \mathbb{R}^D$ be the corresponding full FV. Rather than fitting prompt-specific coefficients, we solve a single ridge-regularized regression,

$$\min_{w \in \mathbb{R}^n} \sum_{b=1}^{B} \left\| \sum_{i=1}^{n} w_i \, v_i^{(b)} - v_{\mathrm{FV}}^{(b)} \right\|_2^2 \; + \; \lambda \|w\|_2^2,$$

which yields a *global optimal weight vector* $w^*$ shared across all $B$ prompts. This formulation enforces that the same linear combination of sub-FVs must simultaneously explain the full FV across diverse prompt realizations, making the test substantially stricter than per-prompt fitting. We solve the associated normal equations $\left( \sum_b X_b^\top X_b + \lambda I \right) w = \sum_b X_b^\top y_b$, and evaluate reconstruction quality using per-prompt cosine similarity and $R^2$.

**Null hypothesis**   To demonstrate that the observed linear superposition is non-trivial, we compare the real fit against two null baselines designed to rule out distinct degenerate explanations. (**Null-1: mismatched dictionary**) We randomly permute the batch dimension of all sub-FVs, breaking the prompt-level correspondence between sub-FVs and the target FV while preserving their marginal distributions. Poor performance under this null indicates that reconstruction depends on the correct prompt-specific alignment between sub-FVs and the corresponding target FV, rather than on fitting to the overall distribution of sub-FVs. (**Null-2: orthogonalized sub-FVs**) We apply a random orthogonal transformation (dimension permutation with random sign flips) to all sub-FVs, preserving norms and intra-set geometry while destroying alignment with the target direction. Failure here shows that the result is not explained by generic low-rank structure or scale effects in hidden states, but instead relies on meaningful directional alignment.

Across tasks and models, the real fits achieve substantially higher cosine similarity and $R^2$ than both nulls. In particular, the orthogonal null collapses to near-zero cosine and near-zero (often negative) $R^2$, indicating that the observed superposition crucially depends on semantic alignment between sub-FVs and the full FV rather than on distributional or geometric invariants alone. The mismatched-dictionary null remains well below the real fit, yet still attains a non-trivial mean cosine similarity (often $\approx 0.80$–$0.85$) and $R^2$ (often $\approx 0.70$–$0.80$). This residual performance suggests that sub-FVs share some cross-prompt structure–e.g., common task-level representation or a partially shared subspace–so that a global linear map can recover part of the target even without the correct prompt-level pairing; however, the large gap to the real fit confirms that accurate reconstruction ultimately relies on the correct association between examples and prompts.

Together, these tests establish that the high-quality linear reconstruction is not a tautological consequence of the linearity of attention, but reflects a robust and non-trivial additive structure in how example-level information is aggregated into the prompt-level FV.

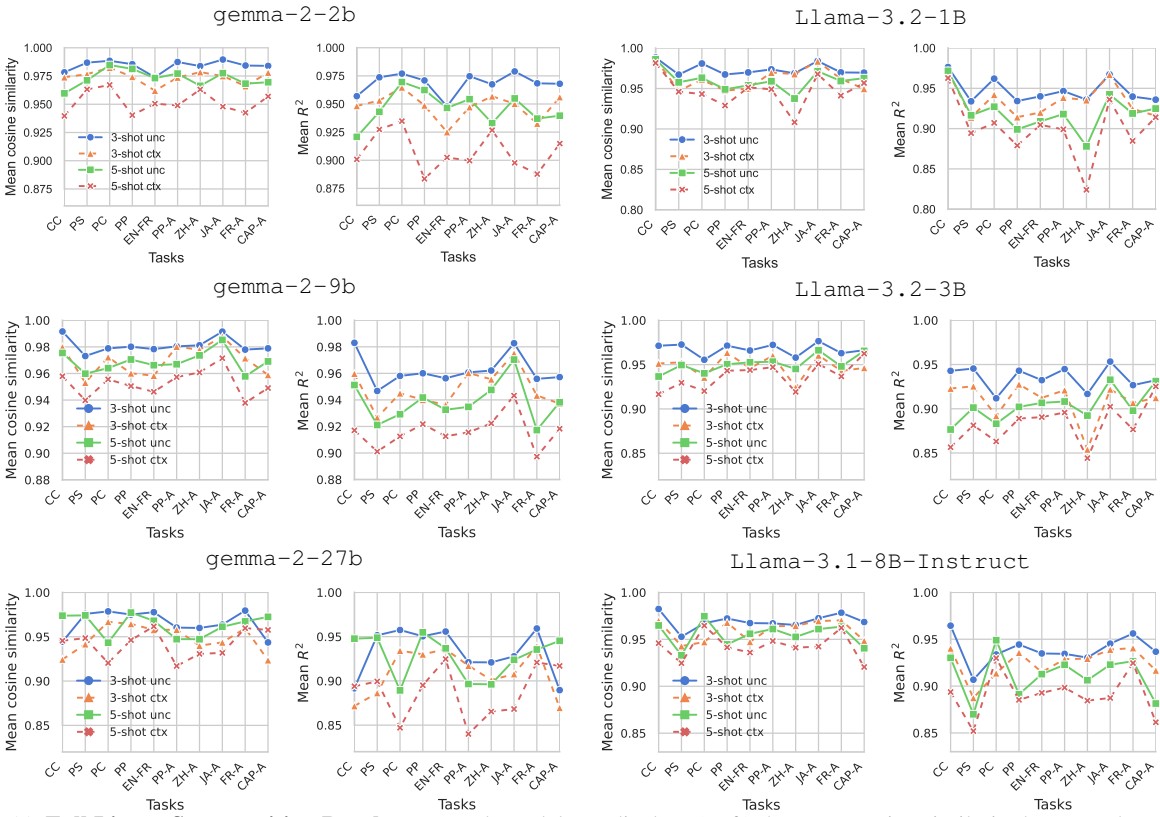

*Figure 11.* **Full Linear Superposition Results.** For each model, we display: (Left) the mean cosine similarity between the observed Function Vector (FV) and the OLS reconstruction; (Right) the mean $R^2$ of the fit across all layers. This is an extension of Main Paper Fig. 2.

| Model | Task | Cosine | $R^2$ | OLS / Full-FV |
|-------|------|--------|-------|---------------|
| gemma-2-9b | COUNTRY–CAPITAL (ctx) | 0.846 | 0.694 | 1.116 |
| | COUNTRY–CAPITAL (unc) | 0.923 | 0.830 | 1.000 |
| | PRESENT–PAST AMBIGUOUS (ctx) | 0.910 | 0.812 | 0.818 |
| | PRESENT–PAST AMBIGUOUS (unc) | 0.936 | 0.826 | 1.000 |
| gemma-2-2b | COUNTRY–CAPITAL (ctx) | 0.918 | 0.838 | 0.936 |
| | COUNTRY–CAPITAL (unc) | 0.866 | 0.733 | 0.975 |
| | PRESENT–PAST AMBIGUOUS (ctx) | 0.874 | 0.756 | 0.884 |
| | PRESENT–PAST AMBIGUOUS (unc) | 0.965 | 0.899 | 0.923 |

*Table 4.* **Preliminary 20-shot validation of linear superposition.** We reconstruct full-prompt FVs from example-level sub-FVs in 20-shot prompts and report both representation-level reconstruction quality and causal recovery under FV injection. **OLS / Full-FV** denotes the ratio between the injection accuracy of the OLS-reconstructed FV and that of the true full-prompt FV.

| Task Name (5-shot) | unc | ctx |
|---|---|---|
| COUNTRY-CAPITAL | 0.1565, 0.1973, 0.2143, 0.2538, **0.3118** | 0.1732, **0.3590**, 0.2267, 0.1868, 0.1839 |
| PERSON-SPORT | 0.1416, 0.1699, 0.2144, 0.2340, **0.3203** | 0.1296, **0.4070**, 0.2808, 0.1508, 0.1341 |
| PARK-COUNTRY | 0.1302, **0.4176**, 0.2754, 0.1842, 0.1066 | 0.1819, 0.1923, 0.1992, 0.2778, **0.3428** |
| PRESENT-PAST | 0.1209, 0.1598, 0.1965, 0.2489, **0.4201** | 0.1631, **0.3238**, 0.2367, 0.2796, 0.2394 |
| ENGLISH-FRENCH | 0.1819, 0.1923, 0.1992, 0.2778, **0.3428** | 0.1995, **0.2876**, 0.2332, 0.2743, 0.2634 |
| PRESENT-PAST AMBIGUOUS | 0.0989, 0.2090, 0.1637, 0.3152, **0.3501** | 0.2091, 0.3947, 0.0656, **0.4971**, -0.0113 |
| CHINESE AMBIGUOUS | 0.0425, 0.2689, 0.0616, 0.3816, **0.4063** | 0.1835, **0.4628**, -0.0501, 0.4132, 0.1829 |
| JAPANESE AMBIGUOUS | 0.0864, 0.2033, 0.1591, 0.3292, **0.3848** | 0.1575, **0.4154**, 0.1683, 0.3679, 0.0251 |
| FRENCH AMBIGUOUS | 0.0997, 0.3154, 0.1540, **0.3767**, 0.2767 | 0.2138, 0.4389, 0.0542, **0.4488**, 0.1245 |
| CAPITALIZE AMBIGUOUS | 0.1426, 0.2978, 0.1785, **0.3088**, 0.2192 | 0.1789, **0.5102**, -0.0012, 0.4849, -0.0874 |

*Table 5.* A demonstration of relative contribution of examples to the overall FV. We report the **normalized global optimal** Ordinary Least Squares (OLS) weights obtained by decomposing the overall FV into individual per-example sub-FVs on `gemma-2-2b`. Uncontextualzied (**unc**) weights correspond to sub-FVs extracted from examples processed in isolation. Contextualized (**ctx**) weights represent the influence of examples processed within the full ICL sequence. Bold values indicate the dominant example within each sequence. In this case, the unambiguous examples for ambiguous tasks are fixed at the positions of **Ex2**, **Ex4**.

# G. Supplemental Evidence for Section 4.1: Intrinsic Attractiveness of Key

We implement *Key patching* as a controlled intervention on the attention computation at FV heads. To construct corrupted prompts for Key patching, we ensure that the tokenization structure of the prompt remains unchanged. Specifically, for each ambiguous dataset, we prepare two aligned prompt pools drawn from the *same task*: one pool of ambiguous examples and one pool of unambiguous examples. When replacing examples, we substitute each example segment with another example from the same task that has the *same number of tokens after tokenization*, including both input and output sub-tokens. This constraint guarantees that the overall prompt length, token positions, and positional encodings remain identical to the original prompt, preventing confounds due to changes in sequence length or token alignment. The ICL query $x_{n+1}$ and the overall prompt template are kept unchanged.

Key patching is implemented using a two-stage forward procedure. In the first forward pass, we run the *uncontextualized* model on the prompt and cache the Query activations at the last token position $t_{n+1}$ for all FV heads and all layers. This cached Query is treated as fixed throughout the intervention. In the second forward pass, we run the uncontextualized model on the same prompt and intervene at the attention computation by replacing the Key activations at the target example token positions with the cached Keys from the corresponding example type and keeping the Query of the last token unchanged.

(a)

(b)

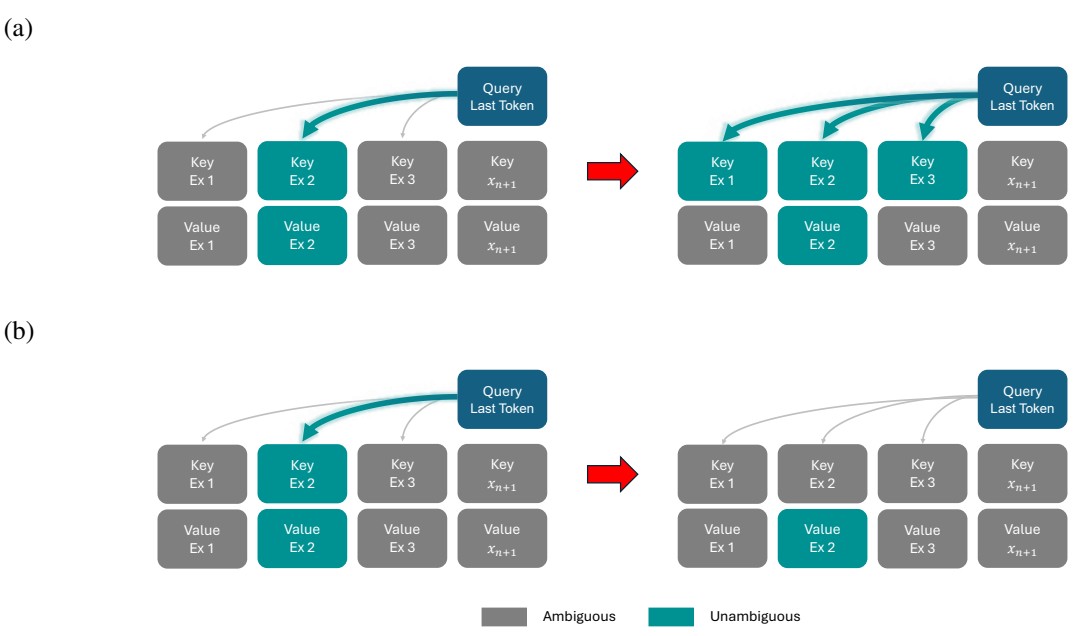

*Figure 12.* Diagram of the Key patching process in Section 4.1. (a) Patching ambiguous Key to unambiguous Key. (b) Patching unambiguous Key to ambiguous Key.

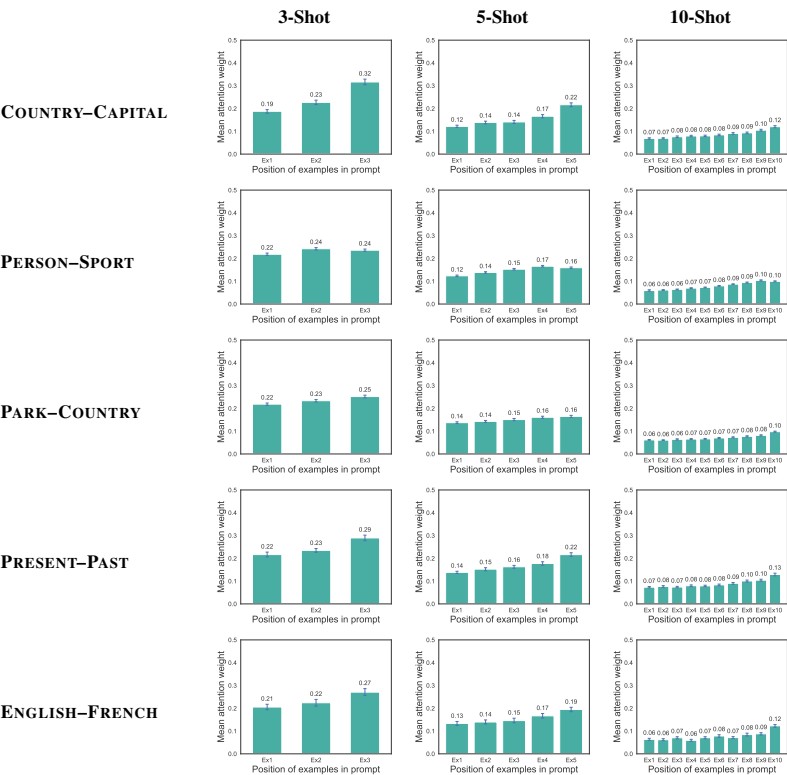

Figure 13. **Normal Tasks:** `gemma-2-2b` mean attention weights of individual examples on FV heads across 3/5/10-shot settings. This is an extension of Main Paper Fig. 5

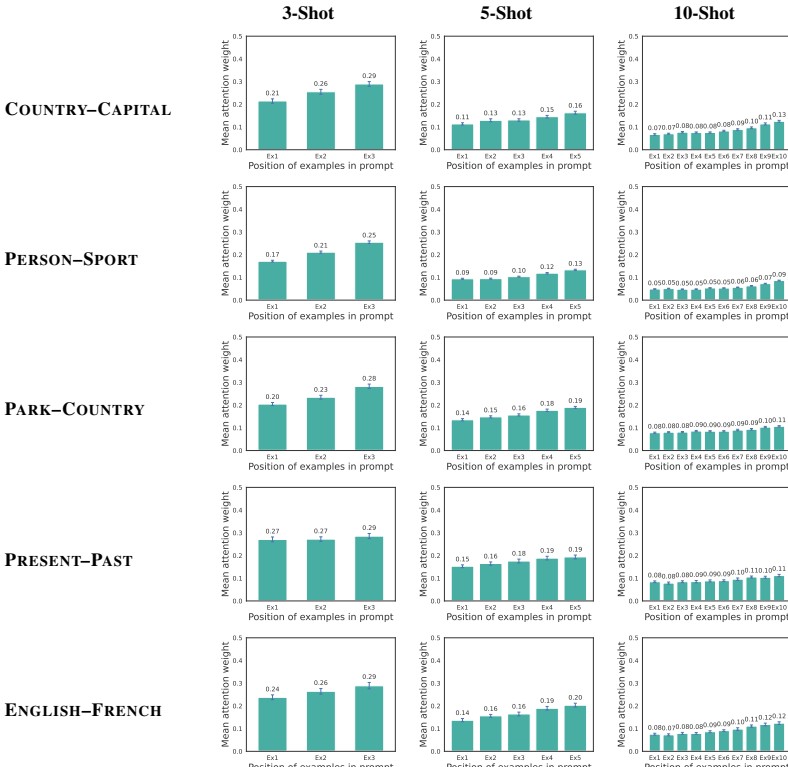

Figure 14. **Normal Tasks:** `gemma-2-9b` mean attention weights of individual examples on FV heads across 3/5/10-shot settings.

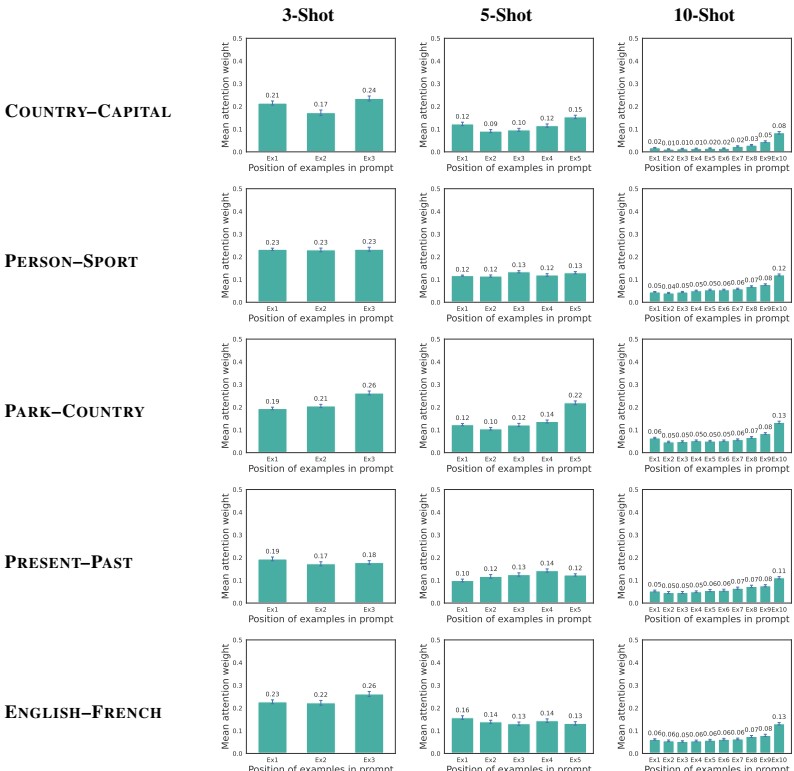

*Figure 15.* **Normal Tasks:** gemma-2-27b mean attention weights of individual examples on FV heads across 3/5/10-shot settings.

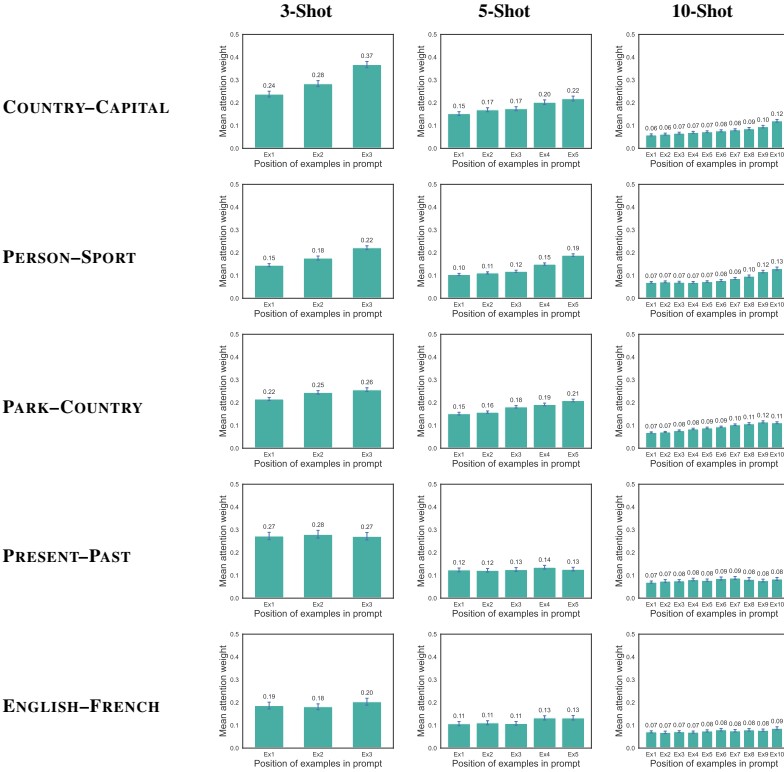

*Figure 16.* **Normal Tasks:** Llama-3.2-1B mean attention weights of individual examples on FV heads across 3/5/10-shot settings.

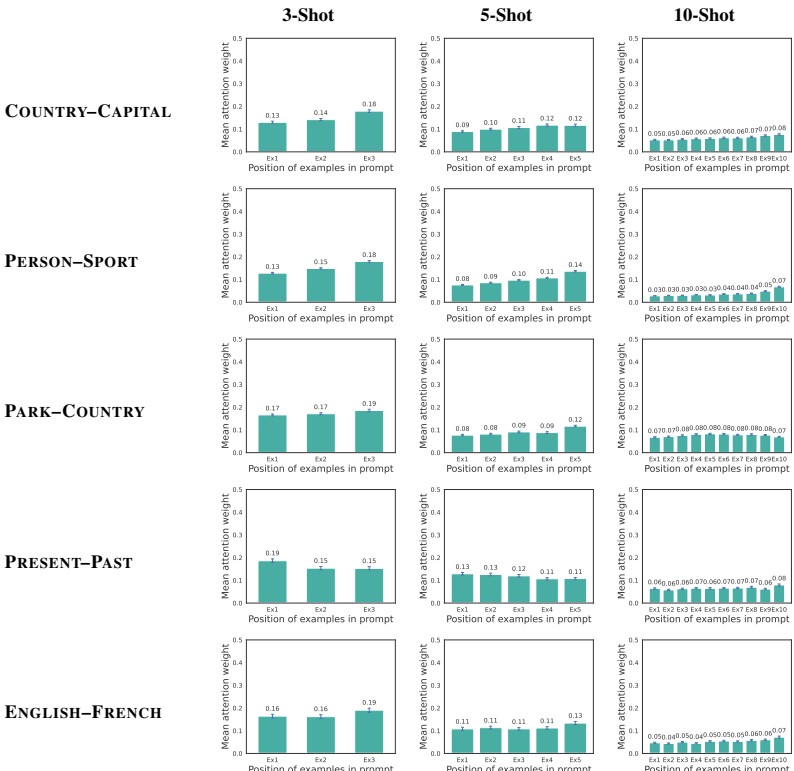

*Figure 17.* **Normal Tasks:** `Llama-3.2-3B` mean attention weights of individual examples on FV heads across 3/5/10-shot settings.

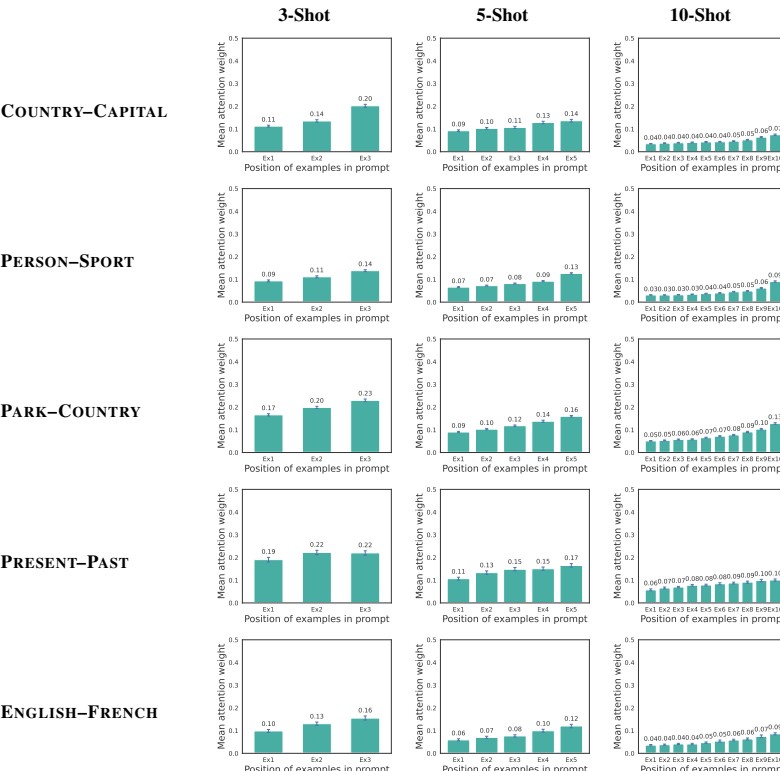

*Figure 18.* **Normal Tasks:** `Llama-3.1-8B-Instruct` mean attention weights of individual examples on FV heads across 3/5/10-shot settings.

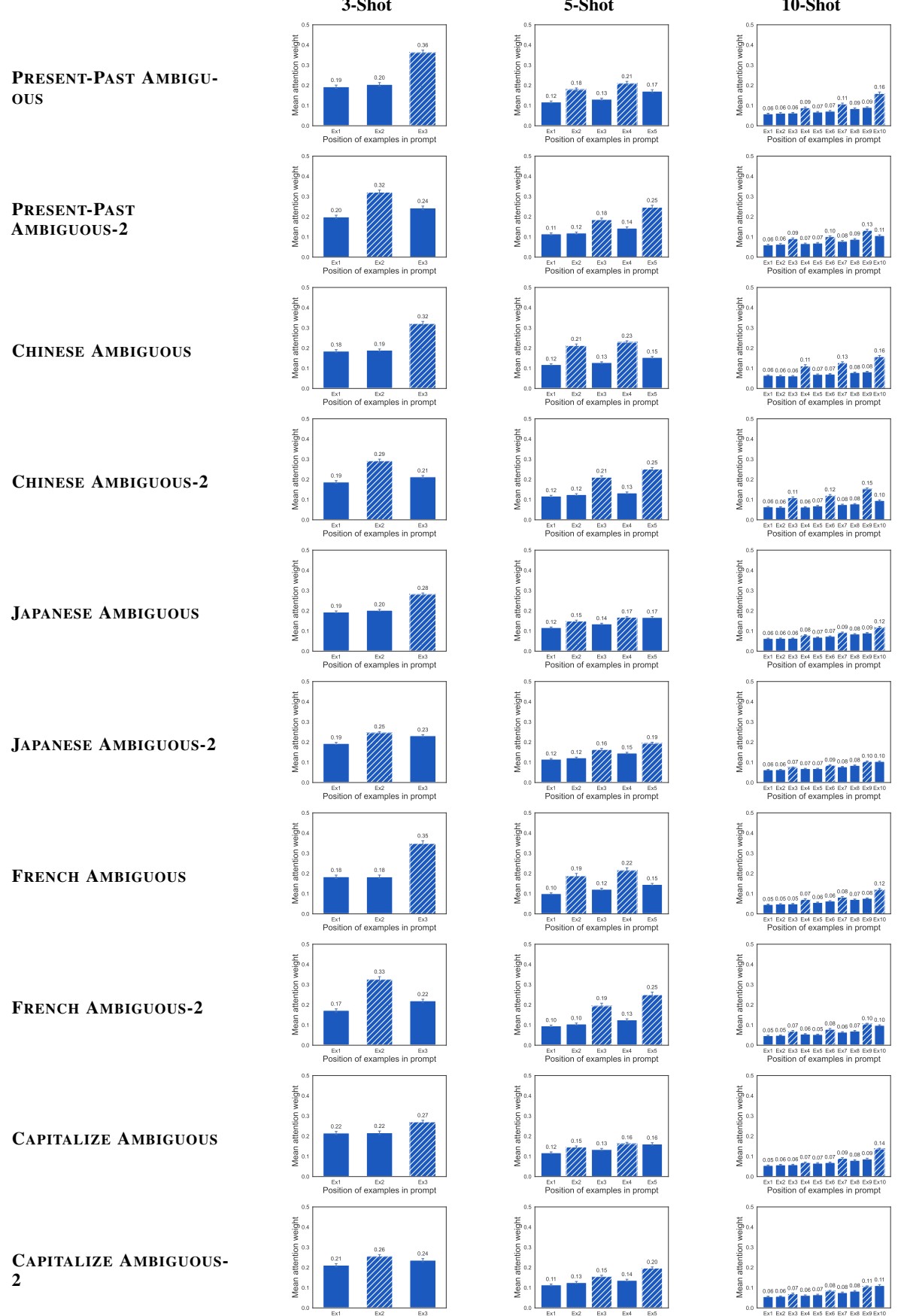

*Figure 19.* **Ambiguous Tasks:** `gemma-2-2b` mean attention weights of individual examples on FV heads across 3/5/10-shot settings. This is an extension of Main Paper Fig. 5.

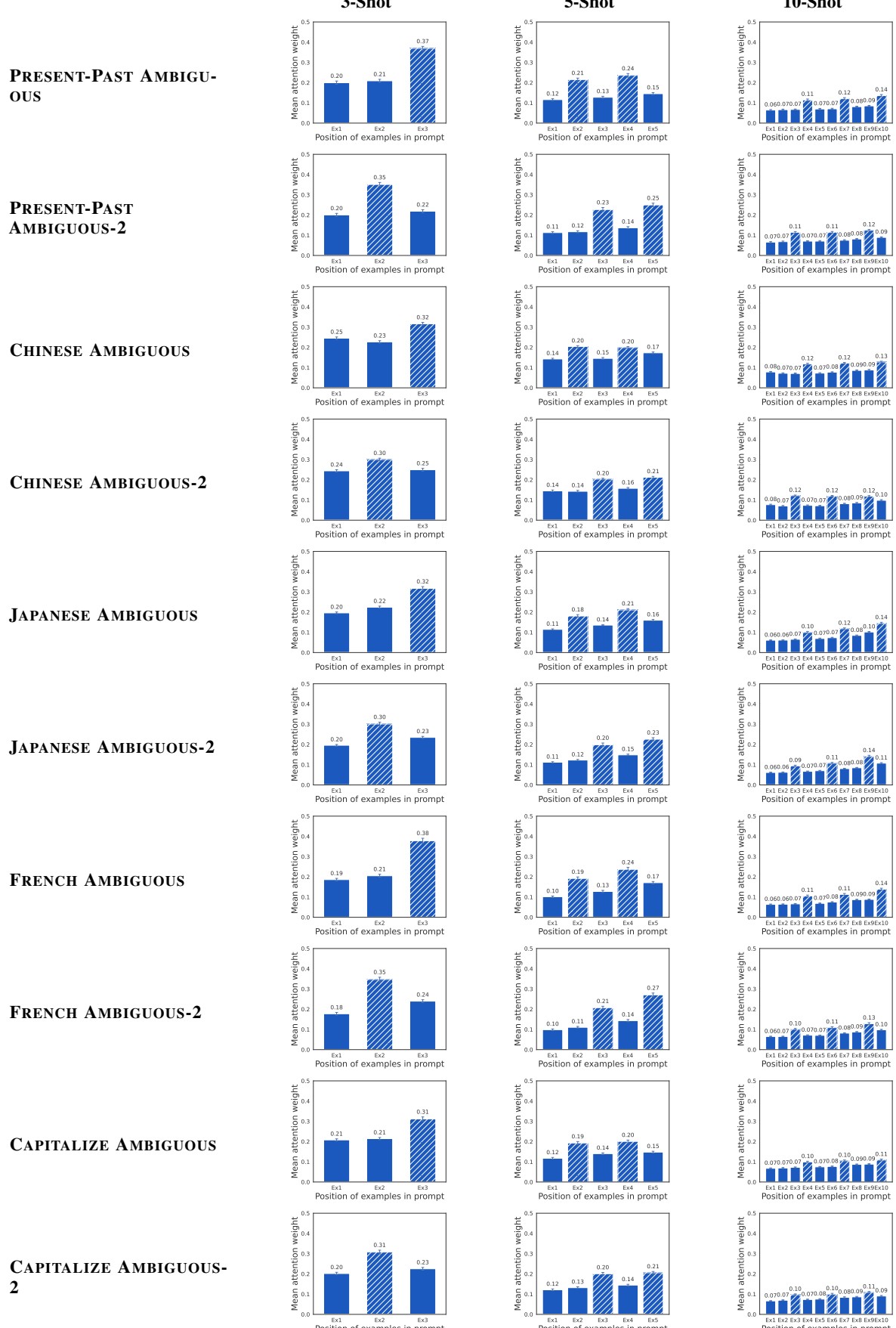

*Figure 20.* **Ambiguous Tasks:** `gemma-2-9b` mean attention weights of individual examples on FV heads across 3/5/10-shot settings.

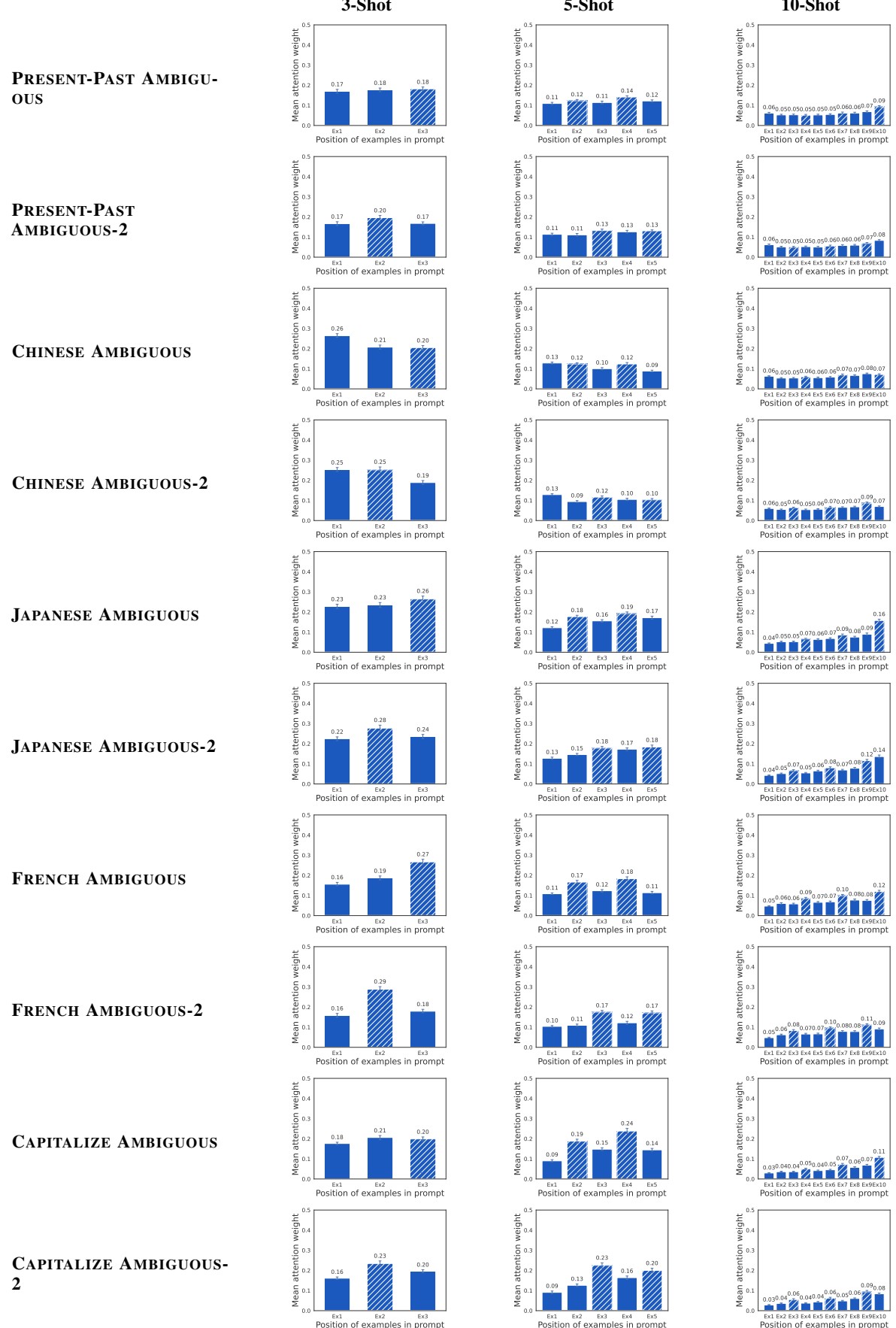

*Figure 21.* **Ambiguous Tasks:** `gemma-2-27b` mean attention weights of individual examples on FV heads across 3/5/10-shot settings.

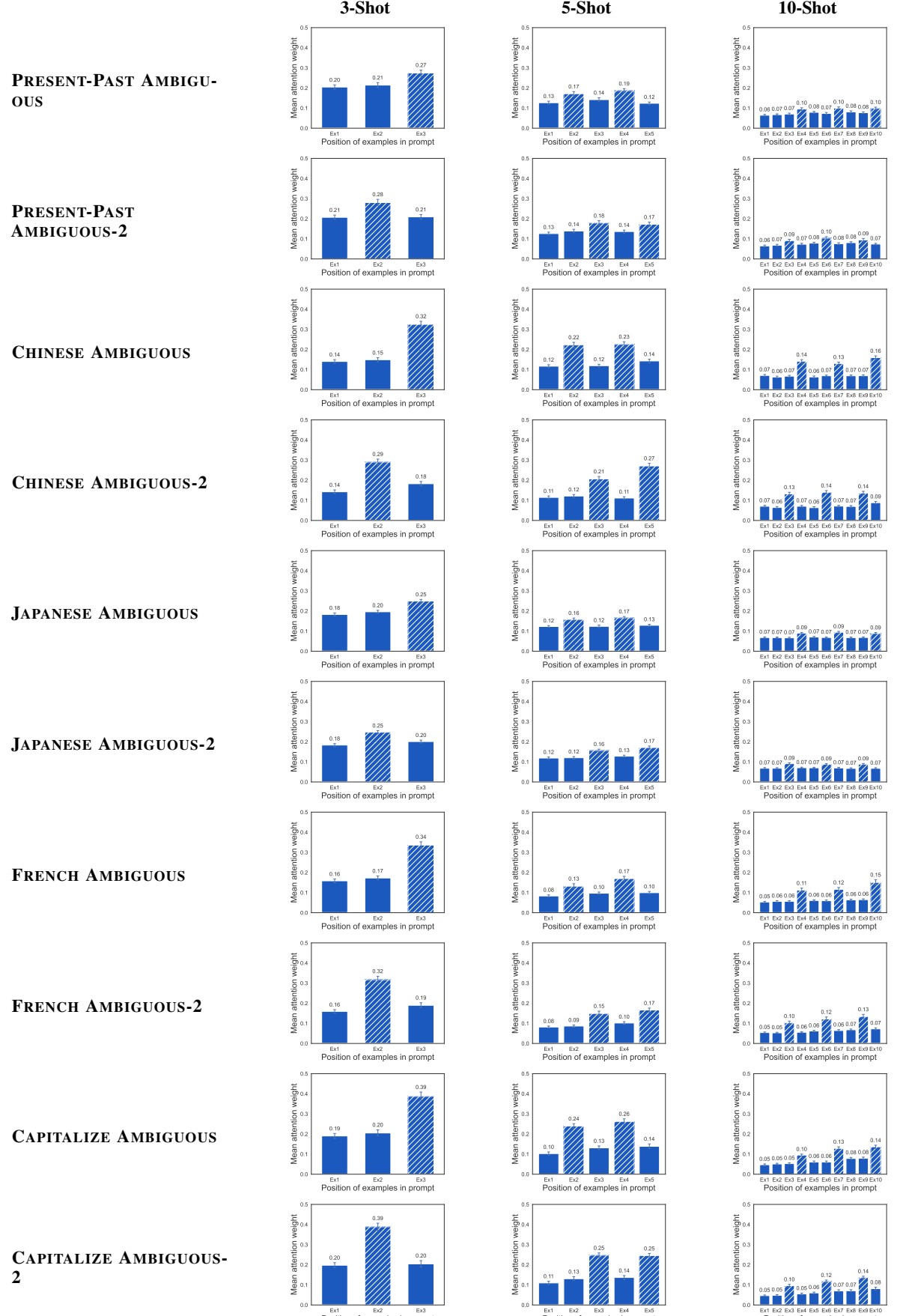

*Figure 22.* **Ambiguous Tasks:** `Llama-3.2-1B` mean attention weights of individual examples on FV heads across 3/5/10-shot settings.

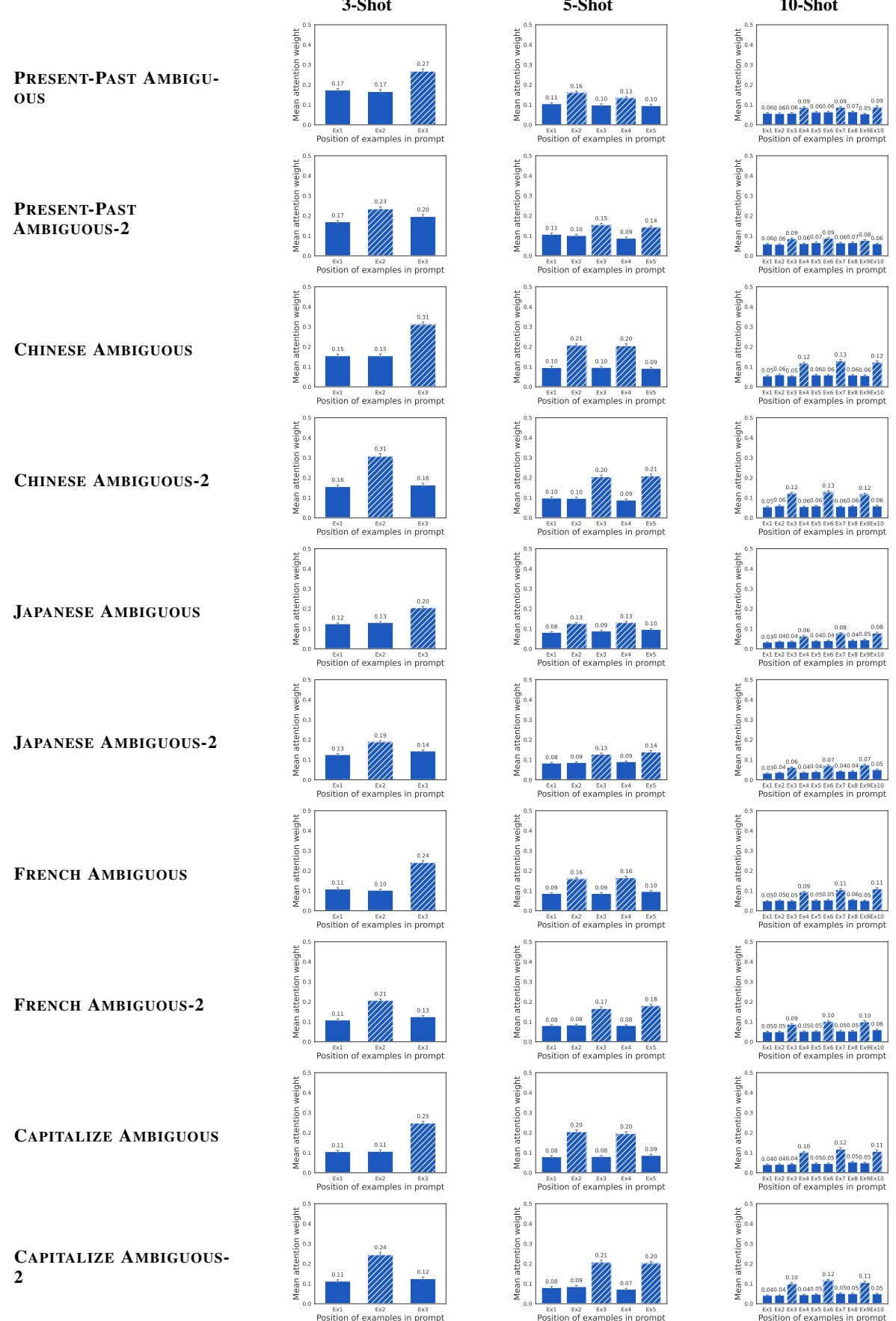

*Figure 23.* **Ambiguous Tasks:** `Llama-3.2-3B` mean attention weights of individual examples on FV heads across 3/5/10-shot settings.

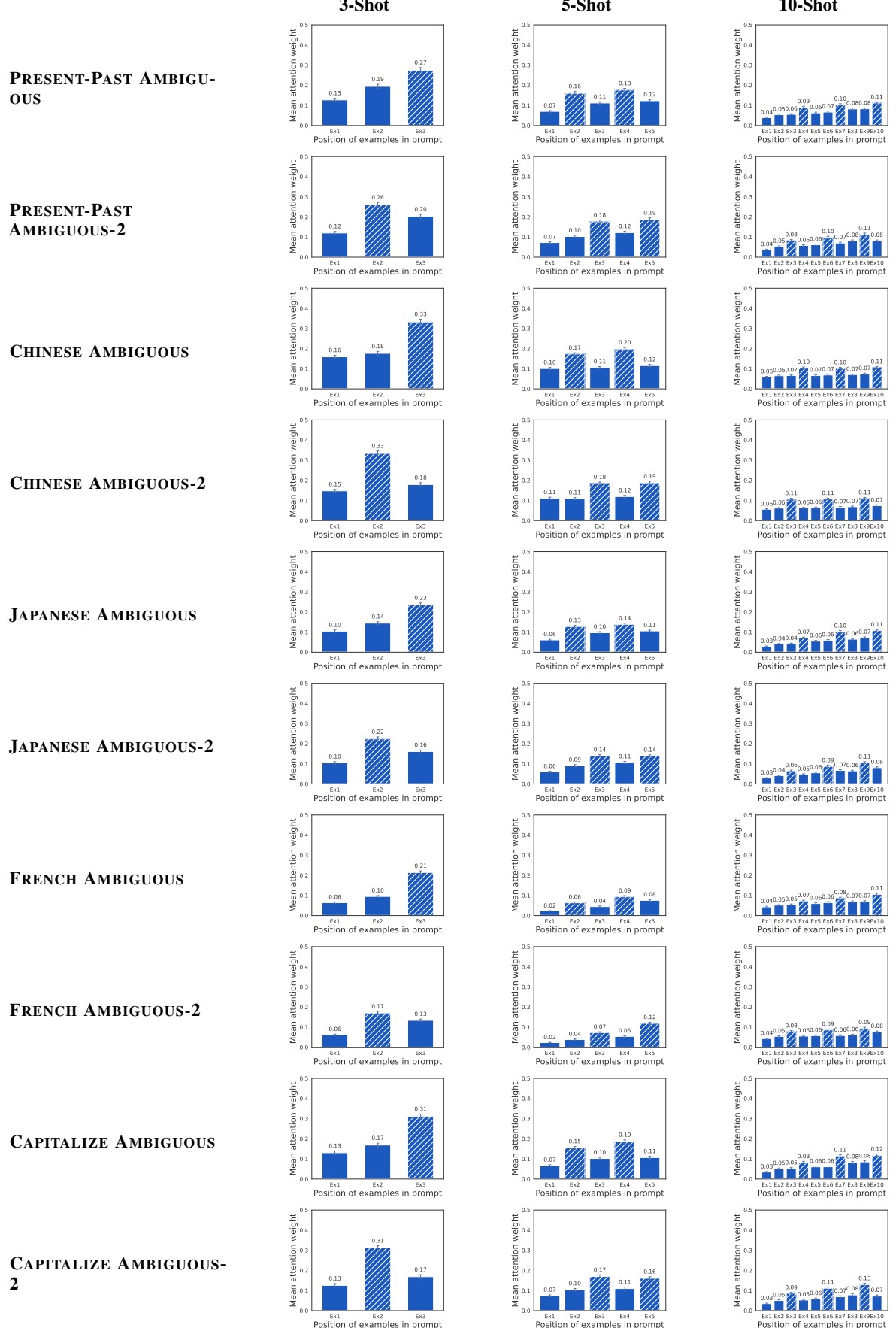

*Figure 24.* **Ambiguous Tasks:** `Llama-3.1-8B-Instruct` mean attention weights of individual examples on FV heads across 3/5/10-shot settings.

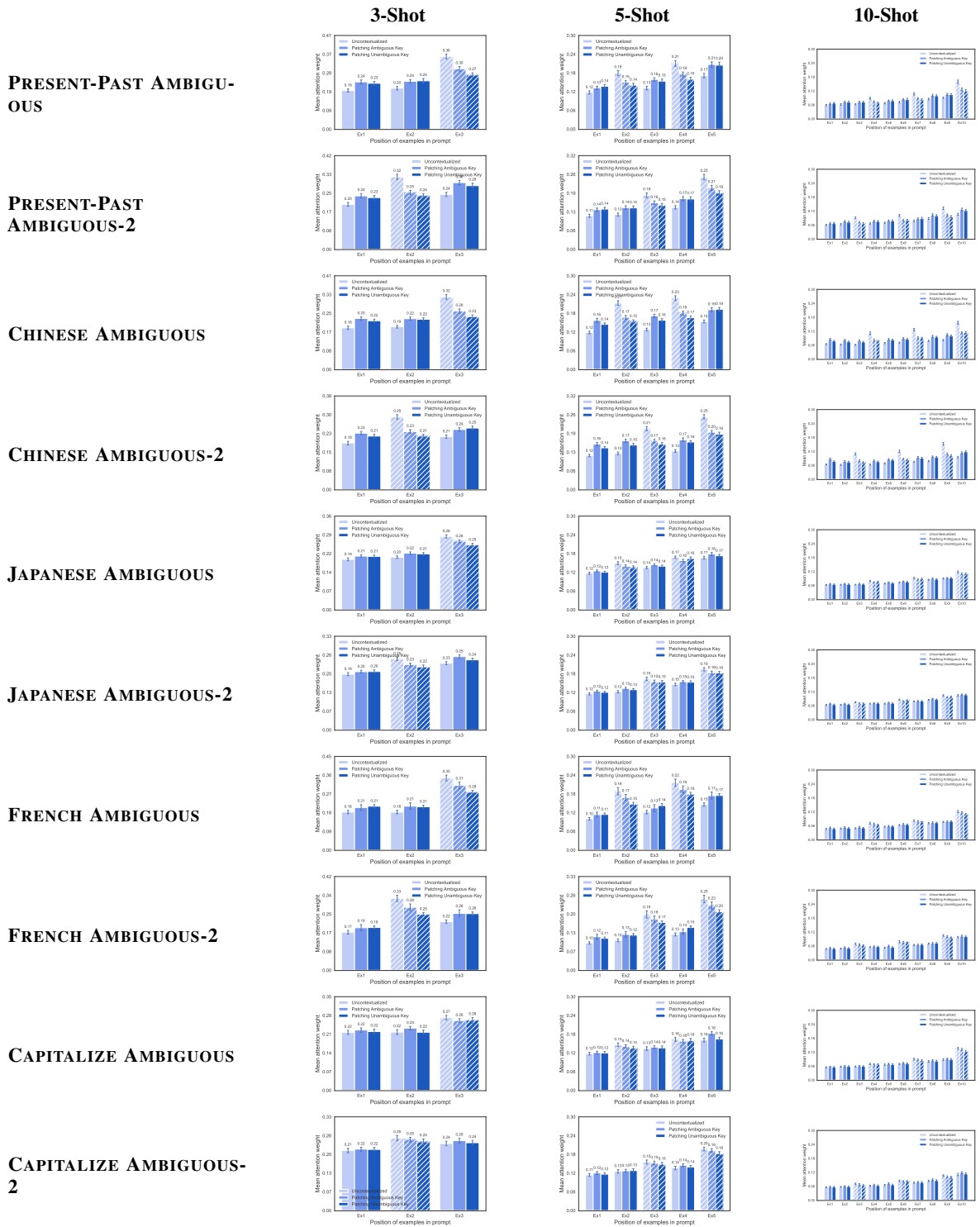

*Figure 25.* Key patching experiment results of all ambiguous tasks on `gemma-2-2b`. This is an extension of Main Paper Fig. 6.

## H. Supplemental Evidence for Section 4.2: Query-Key alignment on contextualization

| | 3-SHOT | | 5-SHOT | | 10-SHOT | |
|---|---|---|---|---|---|---|
| TASK NAME | $\mathcal{C}$ | $\hat{H}$ | $\mathcal{C}$ | $\hat{H}$ | $\mathcal{C}$ | $\hat{H}$ |
| COUNTRY-CAPITAL | **2.16** (2.18) | **.977** (.978) | **3.08** (3.28) | **.991** (.987) | **5.44** (5.98) | **.994** (.993) |
| PERSON-SPORT | **2.12** (2.03) | **.968** (.999) | **3.07** (3.13) | **.986** (.997) | **5.39** (6.05) | **.994** (.992) |
| PARK-COUNTRY | **2.11** (2.05) | **.969** (.999) | **3.12** (3.10) | **.986** (.999) | **5.58** (5.89) | **.996** (.995) |
| PRESENT-PAST | **2.20** (2.10) | **.968** (.993) | **3.15** (3.21) | **.987** (.993) | **5.47** (5.99) | **.994** (.993) |
| ENGLISH-FRENCH | **2.21** (2.09) | **.960** (.994) | **3.30** (3.19) | **.980** (.994) | **5.91** (6.04) | **.991** (.990) |
| PRESENT-PAST AMBIGUOUS | — | **.848** (.971) | — | **.893** (.980) | — | **.881** (.983) |
| CHINESE AMBIGUOUS | — | **.815** (.976) | — | **.877** (.974) | — | **.899** (.978) |
| JAPANESE AMBIGUOUS | — | **.940** (.991) | — | **.967** (.992) | — | **.976** (.992) |
| FRENCH AMBIGUOUS | — | **.824** (.961) | — | **.889** (.966) | — | **.917** (.983) |
| CAPITALIZE AMBIGUOUS | — | **.855** (.996) | — | **.901** (.992) | — | **.910** (.986) |

*Table 6.* Attention metrics **after (before)** contextualization on `gemma-2-2b`. $\mathcal{C}$ and $\hat{H}$ denote the center of FV attention mass and normalized FV attention Entropy, respectively. For ambiguous tasks, entropy is averaged across both position settings. Bold values represent the results after contextualization, with original values in parentheses.

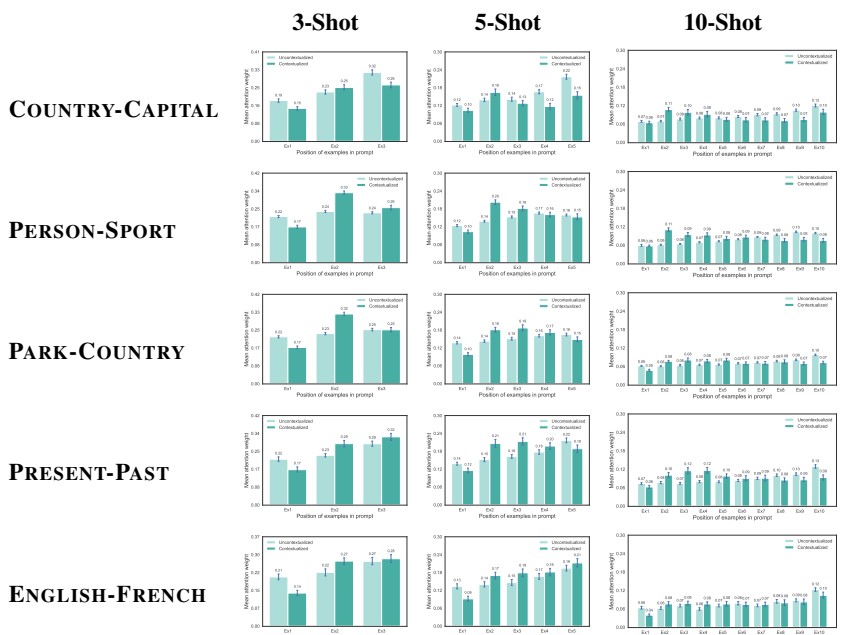

*Figure 26.* **Normal Tasks:** `gemma-2-2b` Influence of contextualization on mean attention weights of individual examples on FV heads. Each row displays the mean attention for a normal task across 3-shot, 5-shot, and 10-shot settings before and after contextualization. This is an extension of Main Paper Fig. 7.

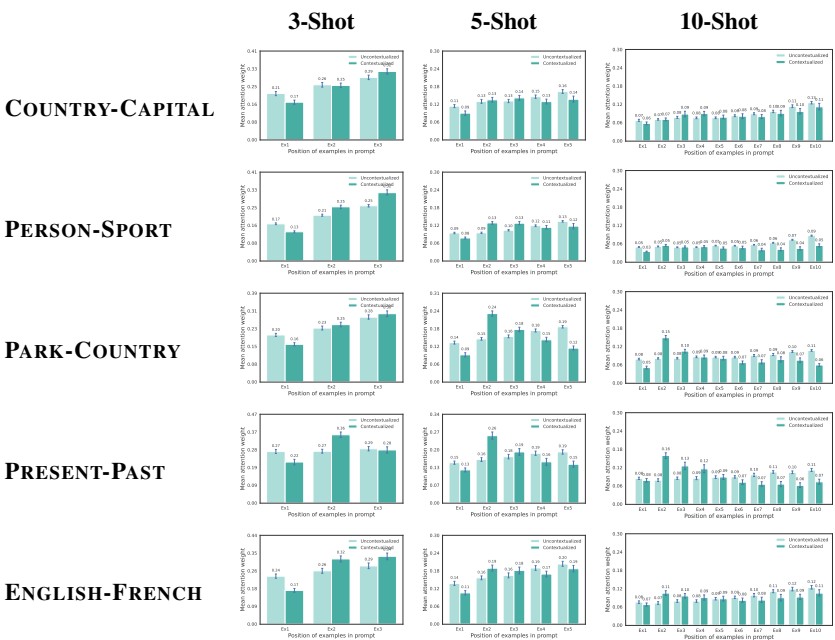

*Figure 27.* **Normal Tasks:** `gemma-2-9b` Influence of contextualization on mean attention weights of individual examples on FV heads. Each row displays the mean attention for a normal task across 3-shot, 5-shot, and 10-shot settings before and after contextualization. This is an extension of Main Paper Fig. 7.

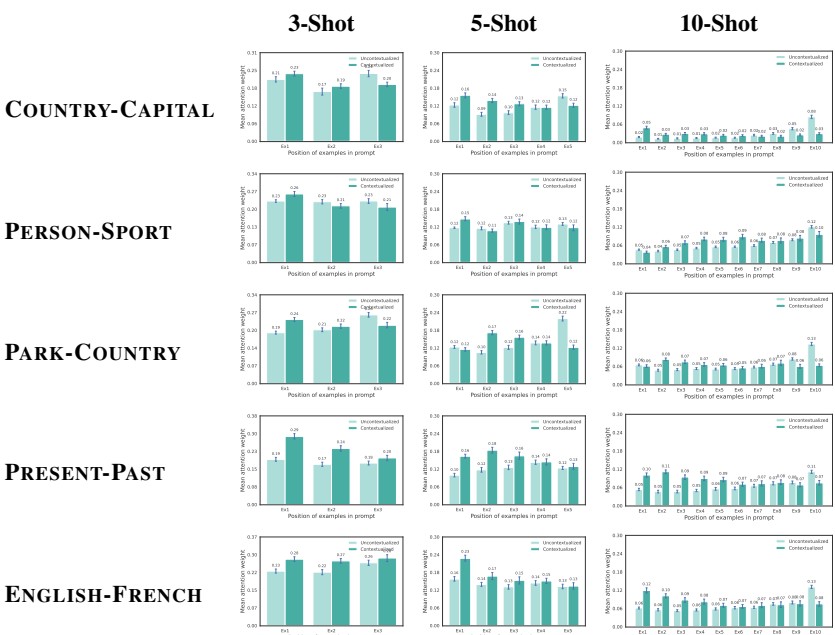

*Figure 28.* **Normal Tasks:** `gemma-2-27b` Influence of contextualization on mean attention weights of individual examples on FV heads. Each row displays the mean attention for a normal task across 3-shot, 5-shot, and 10-shot settings before and after contextualization. This is an extension of Main Paper Fig. 7.

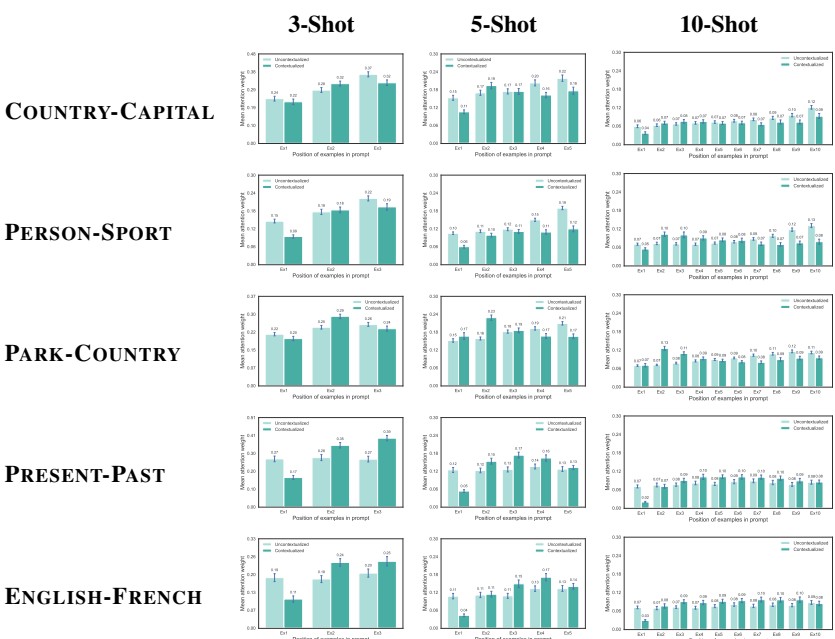

*Figure 29.* **Normal Tasks:** `Llama-3.2-1B` Influence of contextualization on mean attention weights of individual examples on FV heads. Each row displays the mean attention for a normal task across 3-shot, 5-shot, and 10-shot settings before and after contextualization. This is an extension of Main Paper Fig. 7.

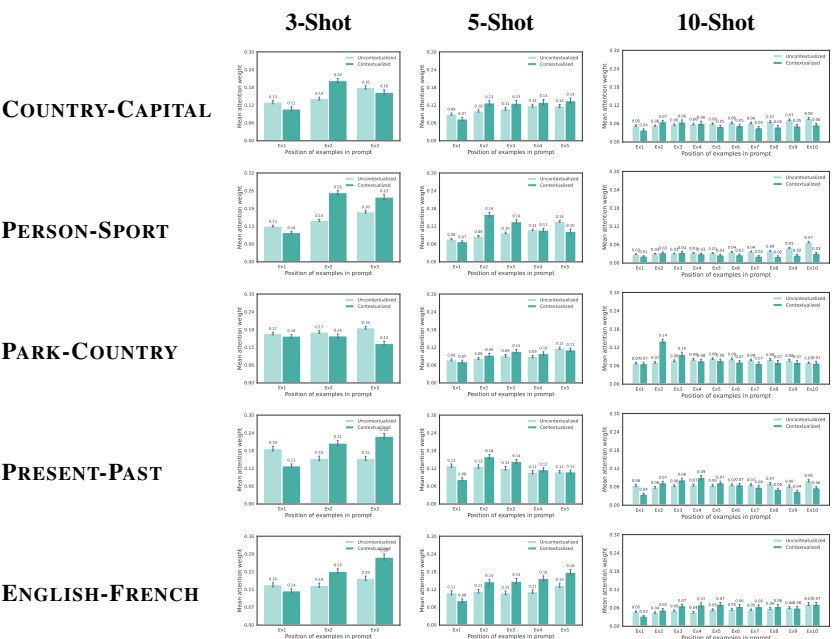

*Figure 30.* **Normal Tasks:** `Llama-3.2-3B` Influence of contextualization on mean attention weights of individual examples on FV heads. Each row displays the mean attention for a normal task across 3-shot, 5-shot, and 10-shot settings before and after contextualization. This is an extension of Main Paper Fig. 7.

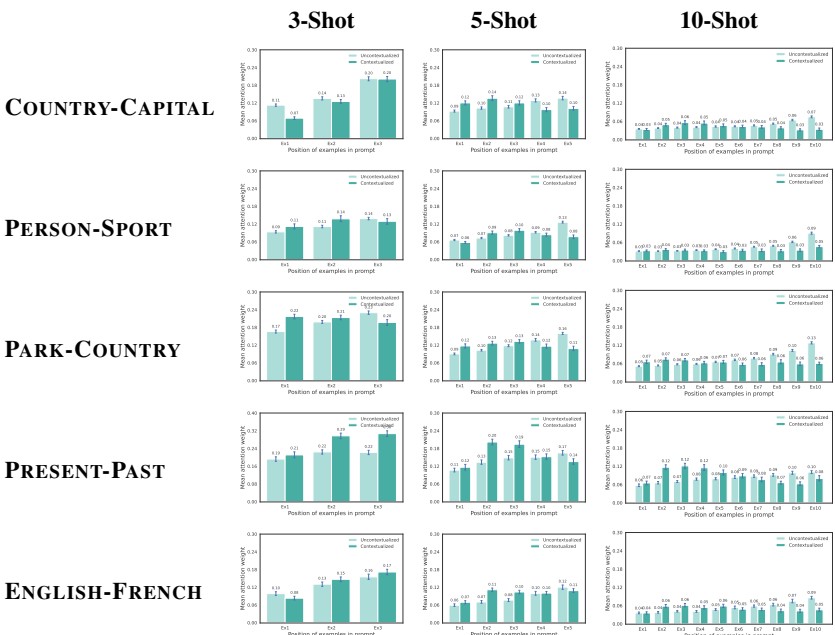

*Figure 31.* **Normal Tasks:** `Llama-3.1-8B-Instruct` Influence of contextualization on mean attention weights of individual examples on FV heads. Each row displays the mean attention for a normal task across 3-shot, 5-shot, and 10-shot settings before and after contextualization. This is an extension of Main Paper Fig. 7.

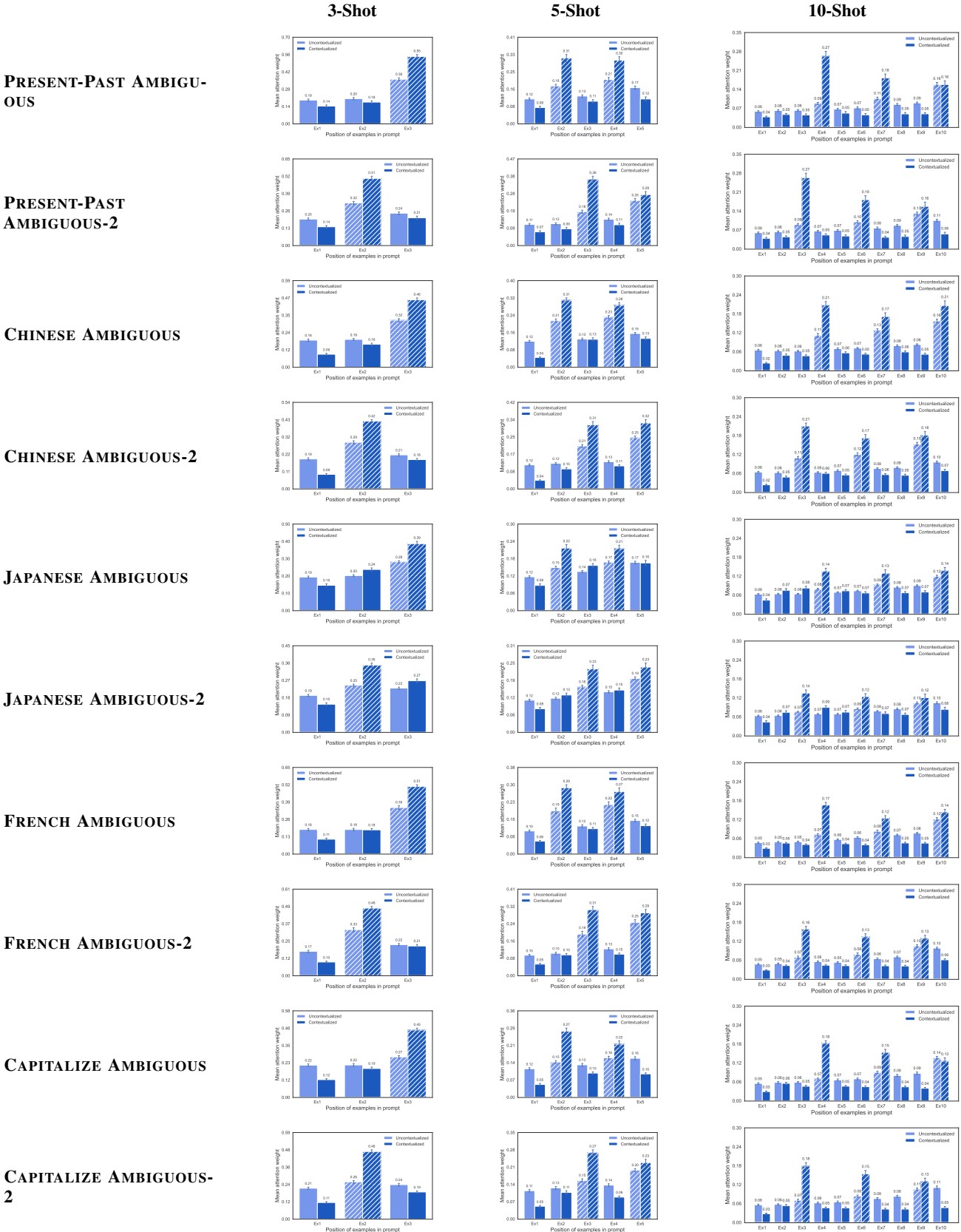

*Figure 32.* **Ambiguous Tasks:** `gemma-2-2b` Influence of contextualization on mean attention weights of individual examples on FV heads. Each row displays the mean attention for a normal task across 3-shot, 5-shot, and 10-shot settings before and after contextualization. This is an extension of Main Paper Fig. 7.

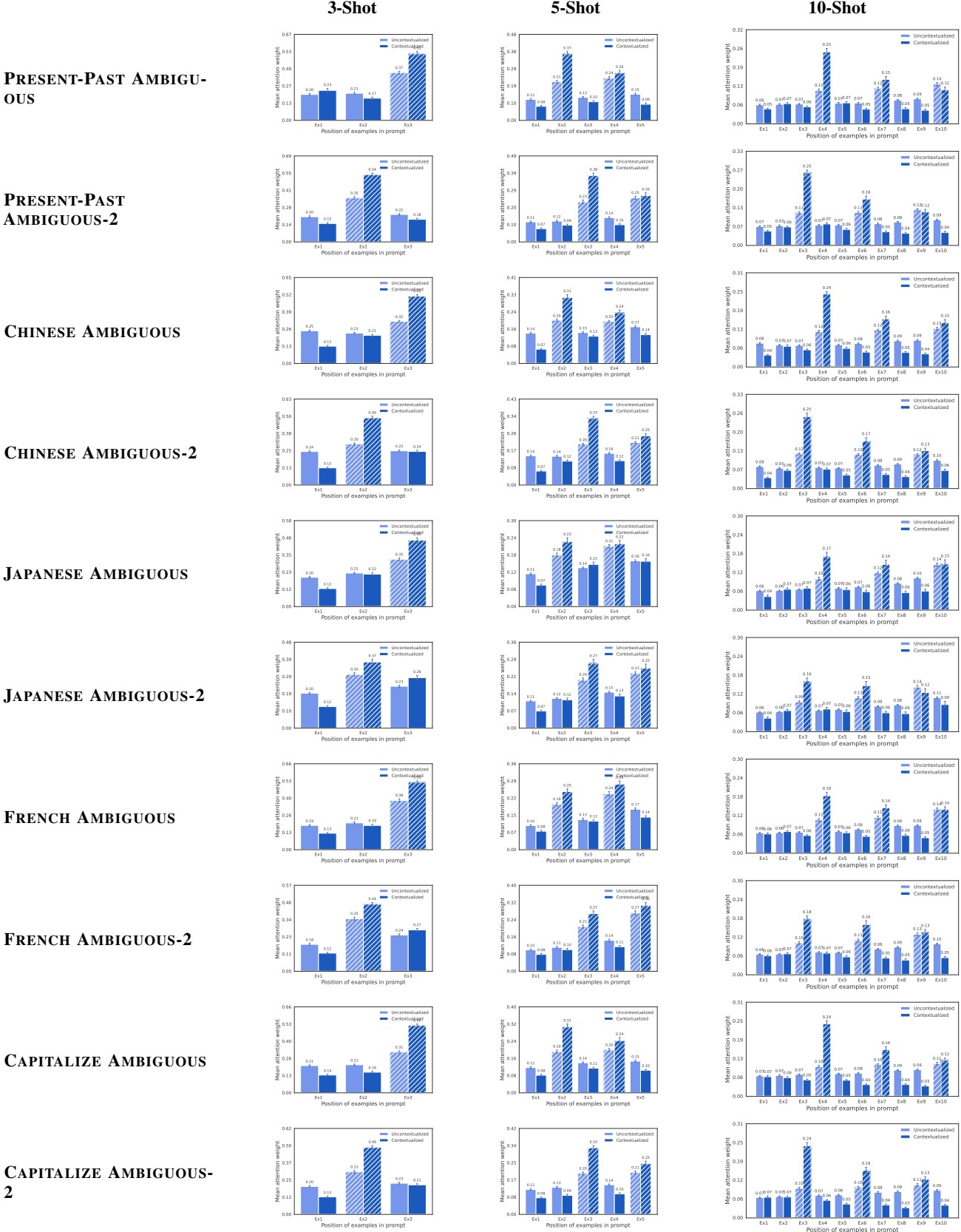

*Figure 33.* **Ambiguous Tasks:** `gemma-2-9b` Influence of contextualization on mean attention weights of individual examples on FV heads. Each row displays the mean attention for a normal task across 3-shot, 5-shot, and 10-shot settings before and after contextualization. This is an extension of Main Paper Fig. 7.

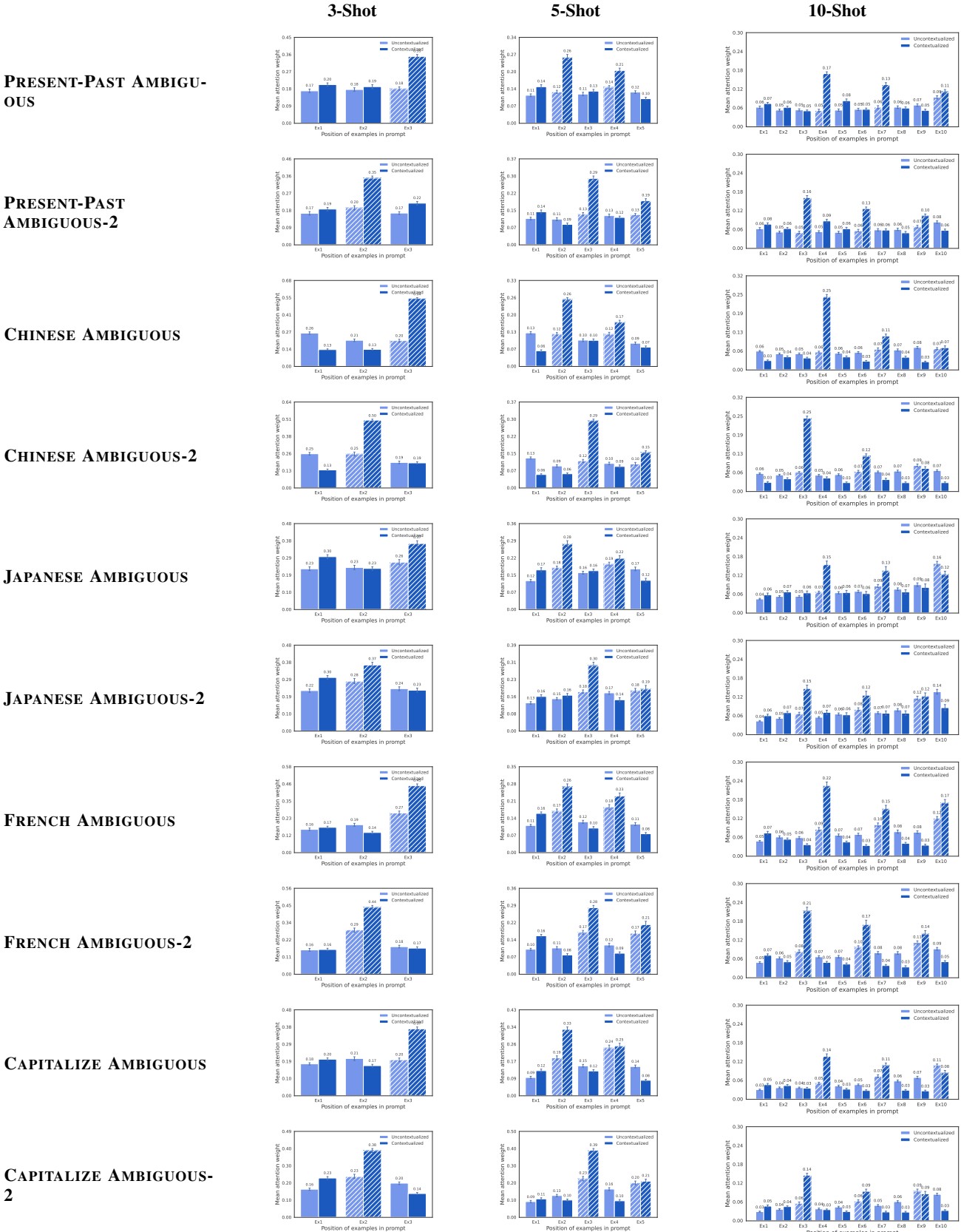

*Figure 34.* **Ambiguous Tasks:** `gemma-2-27b` Influence of contextualization on mean attention weights of individual examples on FV heads. Each row displays the mean attention for a normal task across 3-shot, 5-shot, and 10-shot settings before and after contextualization. This is an extension of Main Paper Fig. 7.

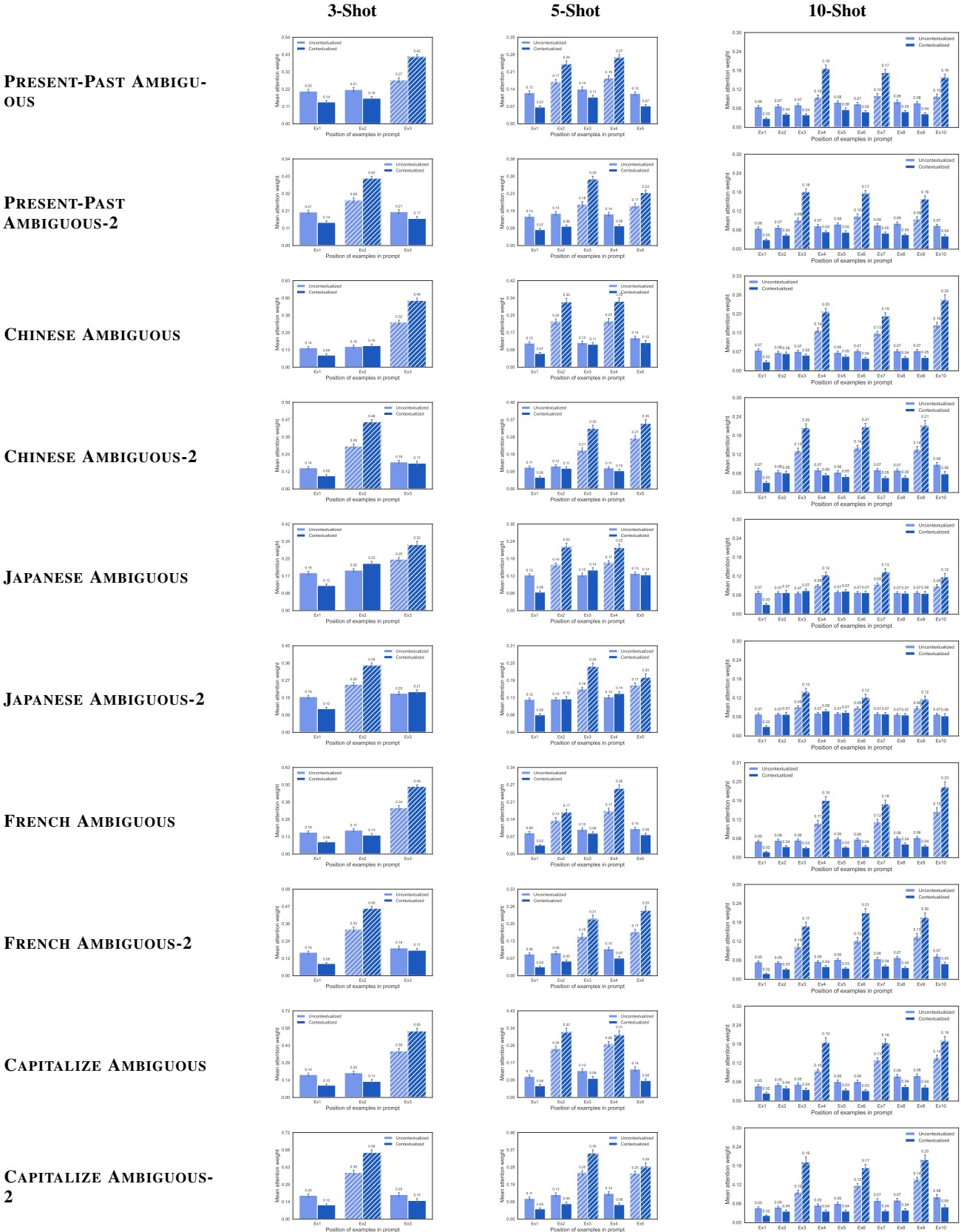

*Figure 35.* **Ambiguous Tasks:** `Llama-3.2-1B` Influence of contextualization on mean attention weights of individual examples on FV heads. Each row displays the mean attention for a normal task across 3-shot, 5-shot, and 10-shot settings before and after contextualization. This is an extension of Main Paper Fig. 7.

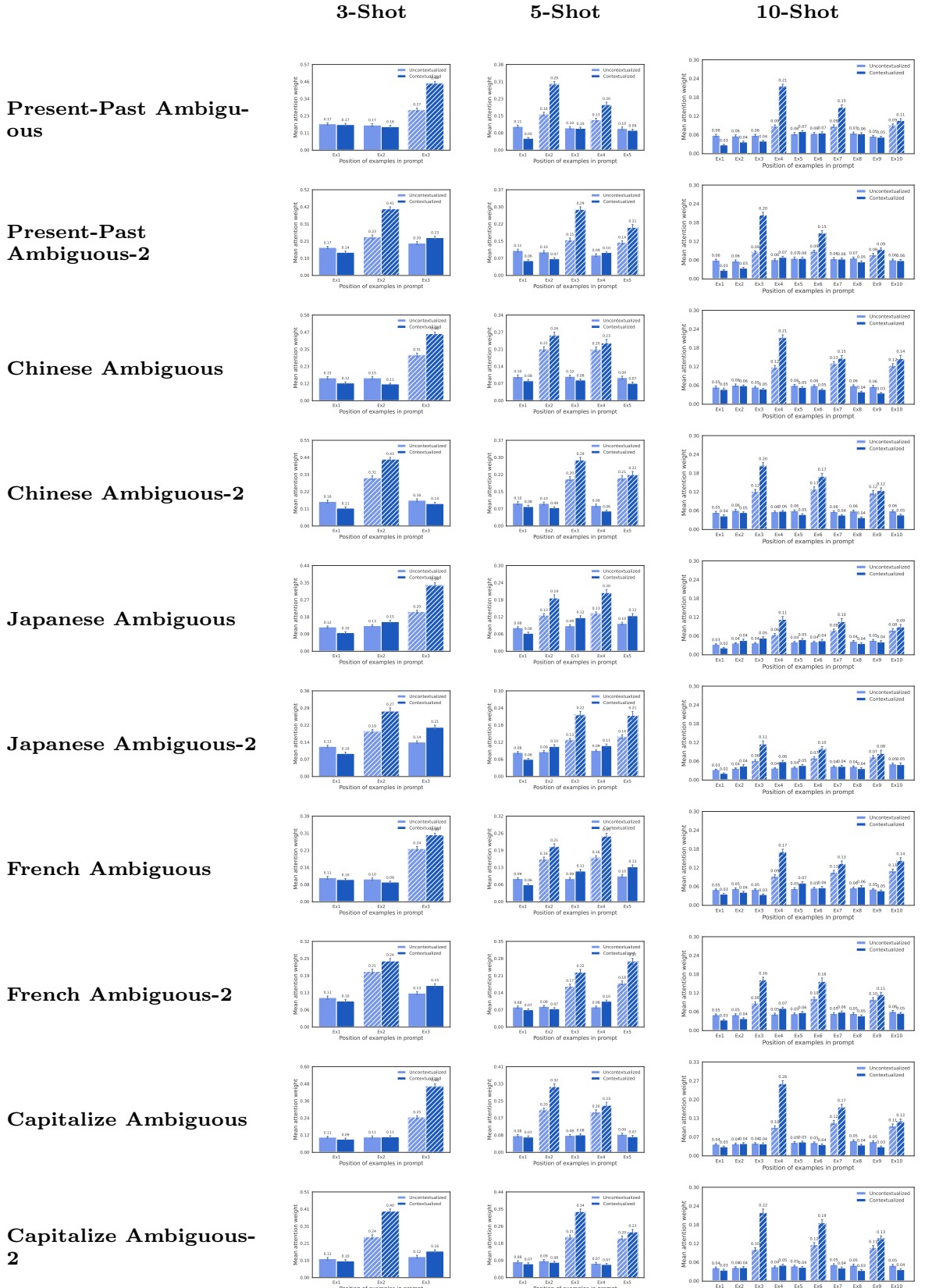

*Figure 36.* **Ambiguous Tasks:** `Llama-3.2-3B` Influence of contextualization on mean attention weights of individual examples on FV heads. Each row displays the mean attention for a normal task across 3-shot, 5-shot, and 10-shot settings before and after contextualization. This is an extension of Main Paper Fig. 7.

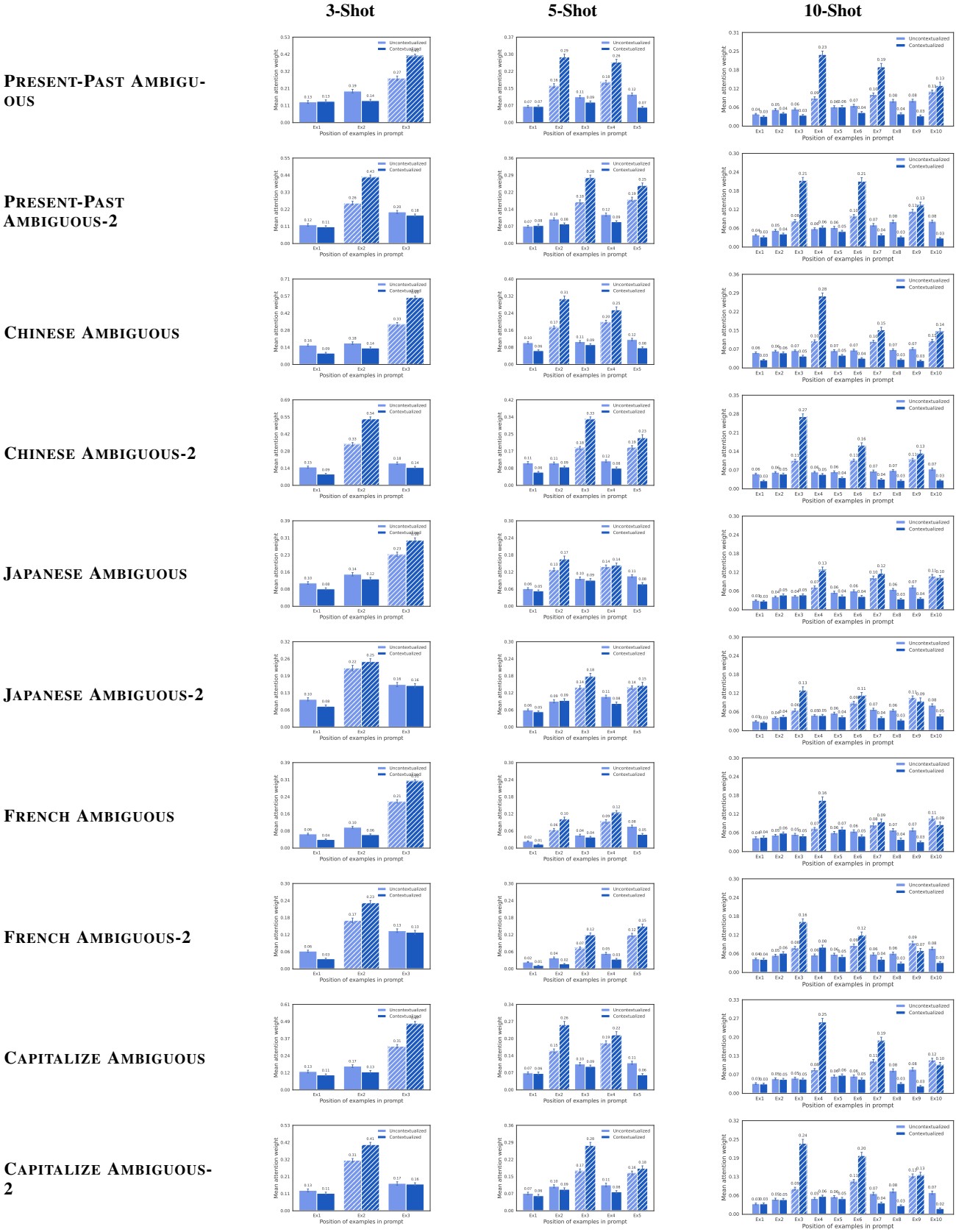

*Figure 37.* **Ambiguous Tasks:** `Llama-3.1-8B-Instruct` Influence of contextualization on mean attention weights of individual examples on FV heads. Each row displays the mean attention for a normal task across 3-shot, 5-shot, and 10-shot settings before and after contextualization. This is an extension of Main Paper Fig. 7.

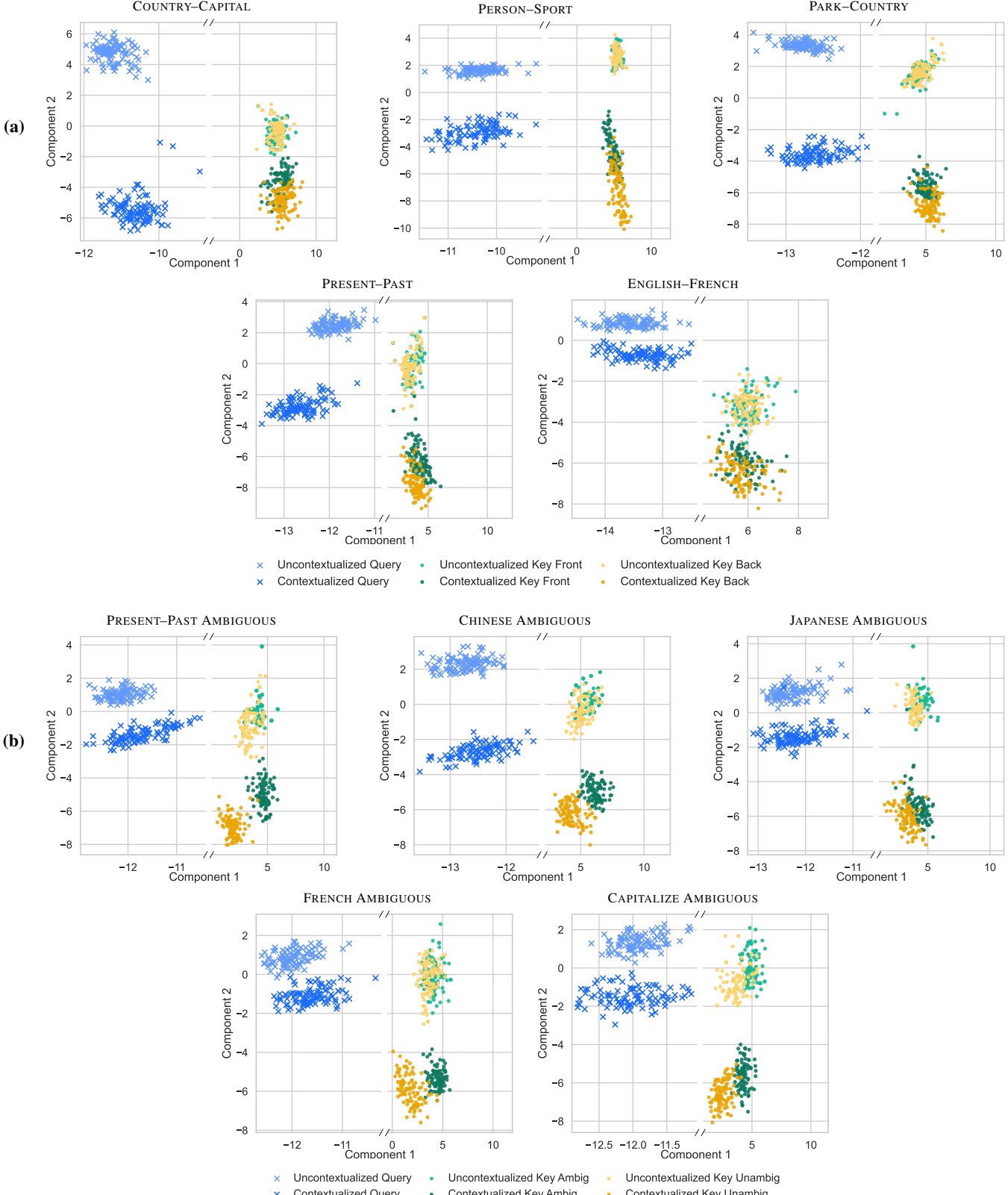

*Figure 38.* Shared PCA visualization of Query and Key vectors extracted from the Top-1 FV head on `gemma-2-2b`. (a) Normal tasks. (b) Ambiguous tasks. For each panel, each category of Q/K vectors is randomly sampled from 100 different prompts. The Q vectors correspond to the Query of the final (prediction) token in the prompt. The K vectors correspond to the Key of the *single token* within each example that receives the highest attention from this last-token Query (i.e., the Top-1 attended token within that example). For normal tasks, we additionally label whether the selected Top-1 token lies in the **Front** (first half) or **Back** (second half) of the example token sequence, reflecting the dominant positional bias. In ambiguous tasks (b), before contextualization, ambiguous Keys and unambiguous Keys largely overlap and appear mixed within the same cluster. After contextualization, the Key distributions reorganize and exhibit a clear separation: unambiguous Keys form a distinct cluster that lies noticeably closer to the corresponding Query vectors on Component 1 axis than ambiguous Keys.

# I. Supplemental Evidence for Section 5: QK-mediated attention routing drives FV quality improvements

(a)

(b)

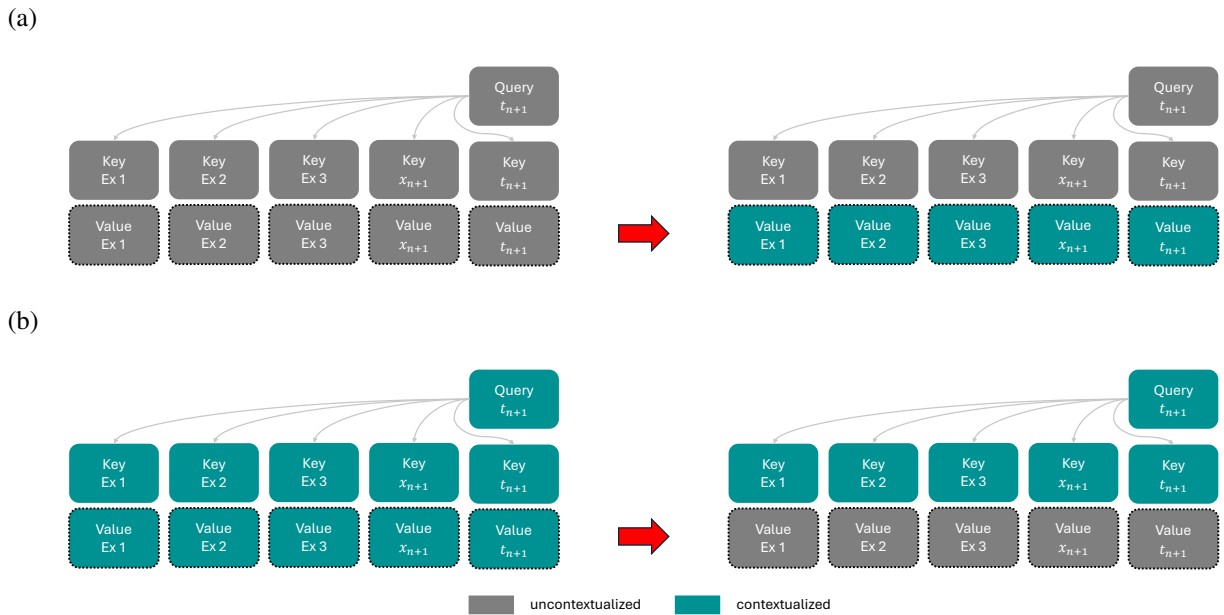

*Figure 39.* Diagram of the Value patching process in Section 5. (a) Patching V under uncontextualized ablation, replacing uncontextualized V with contextualized V, keeping Q/K fixed. (b) Patching V under contextualized ablation, replacing contextualized V with uncontextualized V, keeping Q/K fixed. Both patching experiments are done on all attention heads.

In the main text (Sec. 5), we report aggregated Shapley trends showing that contextualization gains in FV quality are most consistently mediated by the QK pathway, while V acts as a less ubiquitous, often complementary contributor. Here we provide the full *configuration-level* statistics underlying those summaries by enumerating all *model × dataset × n-shot × positional-setting* combinations and assigning each to a causal regime (QK-only, V-only, QK+V, Others) using the same activity threshold $\phi \geq 0.05$. This detailed breakdown makes transparent how the aggregate counts arise and verifies that the qualitative conclusion holds across individual models and task categories rather than being driven by a small subset of settings (Table. 7). Under this criterion, contextualization is rarely negligible on ambiguous tasks (inactive in only 31/180 configurations, i.e., $\phi_{QK} < 0.05$ and $\phi_V < 0.05$ ) where 180 counts all *models × datasets × n-shot × positional settings* combinations on ambiguous prompts, and QK is active in the vast majority of cases (**146/180**); consequently, purely V-driven improvements are rare (**V-only: 3/180**). V is active in 56/180 configurations and most often provides additional gains on top of QK rather than dominating it. In contrast, on normal tasks contextualization is frequently weak or inconsistent (inactive in 48/90 configurations; among the remaining settings where contextualization has a non-negligible effect, QK is active in most cases (37/42), whereas V is active in fewer (17/42), with V-only behavior again uncommon (5/90).

| Model | Task Category | Contextualization Causal Regime | | | | Total |
|---|---|---|---|---|---|---|
| | | QK-only | V-only | QK+V | Others* | |
| gemma-2-2b | Normal | 2 | 4 | 2 | 7 | 15 |
| | Ambiguous | 13 | 1 | 12 | 4 | 30 |
| gemma-2-9b | Normal | 3 | 0 | 3 | 9 | 15 |
| | Ambiguous | 8 | 2 | 20 | 0 | 30 |
| gemma-2-27b | Normal | 7 | 0 | 0 | 8 | 15 |
| | Ambiguous | 24 | 0 | 3 | 3 | 30 |
| Llama-3.2-1B | Normal | 3 | 0 | 4 | 8 | 15 |
| | Ambiguous | 10 | 0 | 9 | 11 | 30 |
| Llama-3.2-3B | Normal | 3 | 1 | 3 | 8 | 15 |
| | Ambiguous | 15 | 0 | 8 | 7 | 30 |
| Llama-3.1-8B-Instruct | Normal | 7 | 0 | 0 | 8 | 15 |
| | Ambiguous | 23 | 0 | 1 | 6 | 30 |
| Total | Normal | 25 | 5 | 12 | 48 | 90 |
| | Ambiguous | 93 | 3 | 53 | 31 | 180 |

*Table 7.* **Distribution of Causal Regimes across Models and Task Categories.** We categorize each experimental configuration (defined by a unique model, $n$-shot, and task type) into one of four regimes based on the Shapley value ($\phi$) of the QK and V circuits. This is the detailed statistics of Main Paper Sec.5.

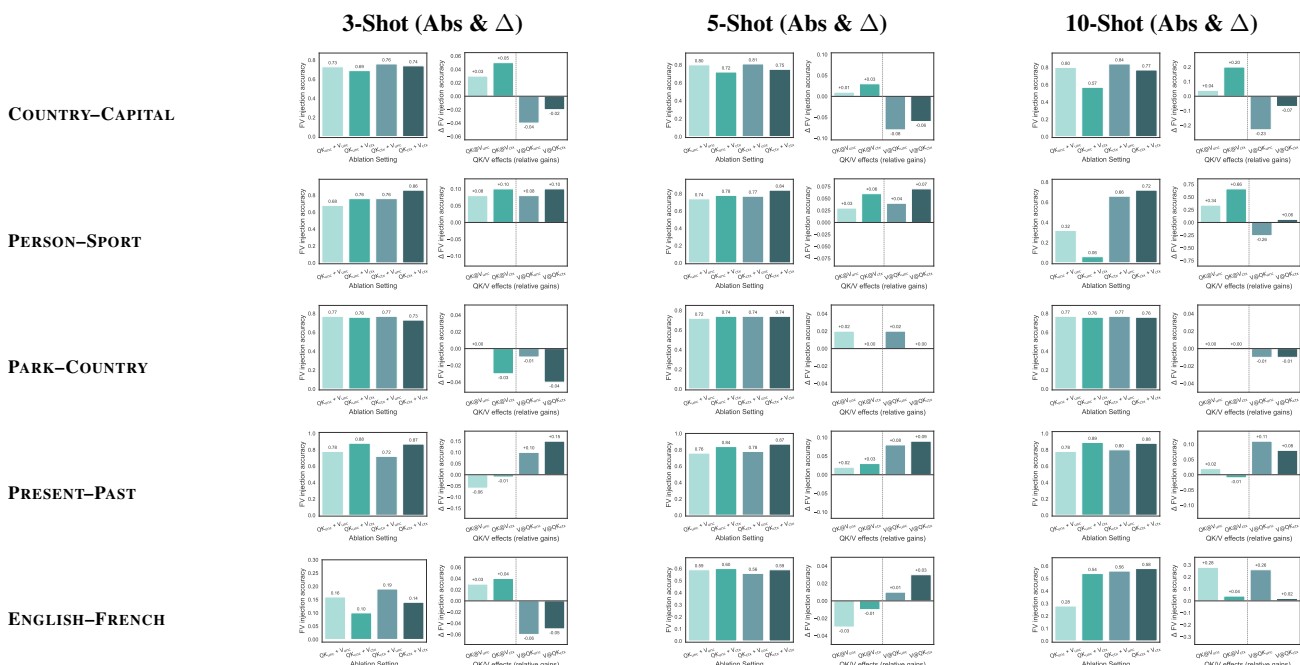

*Figure 40.* Causal Decomposition of Contextualization Gains on gemma-2-2b. (Normal Tasks) For each task and shot count, the **left plot** shows absolute FV injection accuracy $F(\text{QK}, \text{V})$ across four factorial intervention settings: (1) *Uncontextualized* $F(0,0)$; (2) *QK-contextualized* $F(1,0)$; (3) *V-contextualized* $F(0,1)$; (4) *Full contextualized* $F(1,1)$. We plot the marginal effects: $\text{QK@V}_{unc} := F(1,0) - F(0,0)$, $\text{QK@V}_{ctx} := F(1,1) - F(0,1)$, $\text{V@QK}_{unc} := F(0,1) - F(0,0)$, $\text{V@QK}_{ctx} := F(1,1) - F(1,0)$. The Shapley values are then $\phi_{QK} := \frac{1}{2}(\text{QK@V}_{unc} + \text{QK@V}_{ctx})$, $\phi_V := \frac{1}{2}(\text{V@QK}_{unc} + \text{V@QK}_{ctx})$. The **right plot** displays the marginal gains ($\Delta$) used in the Shapley decomposition: the first two bars show the effect of contextualizing the QK pathway ($\phi_{\text{QK}}$ components) under different V settings, while the latter two show the effect of contextualizing the V pathway ($\phi_{\text{V}}$ components) under different QK settings. This is the detailed result of Main Paper Fig. 8.

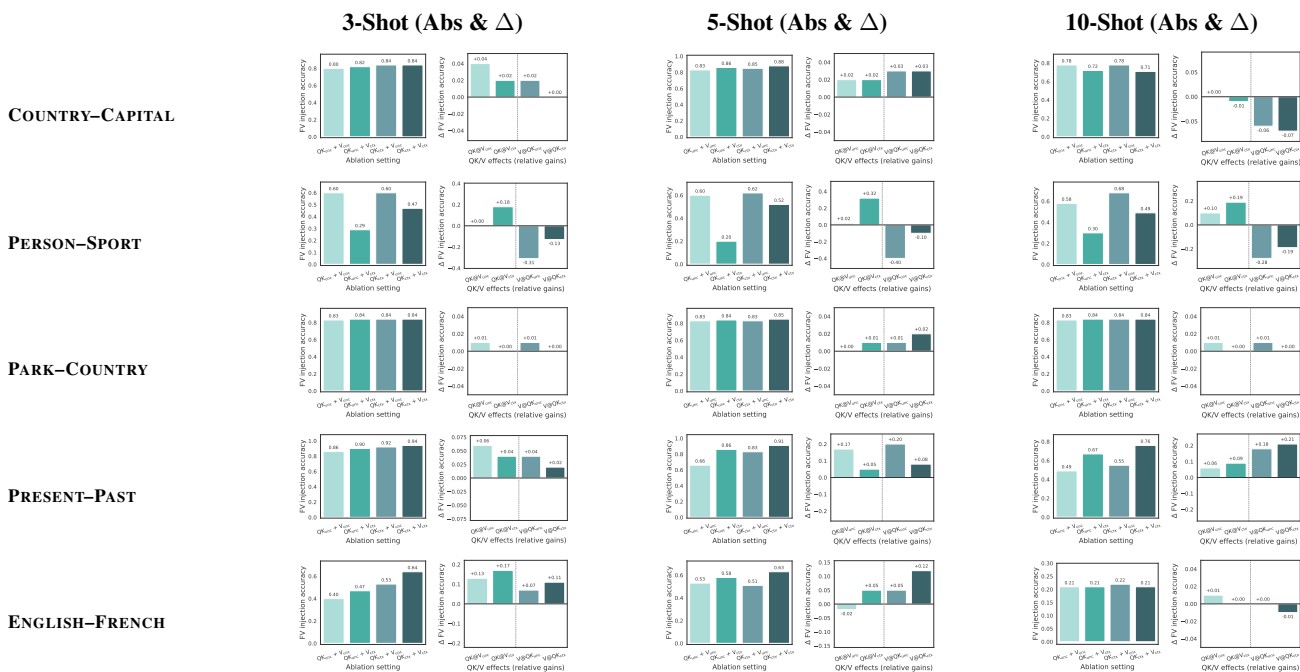

*Figure 41.* Causal Decomposition of Contextualization Gains on `gemma-2-9b`. (Normal Tasks) This is the detailed result of Main Paper Fig. 8.

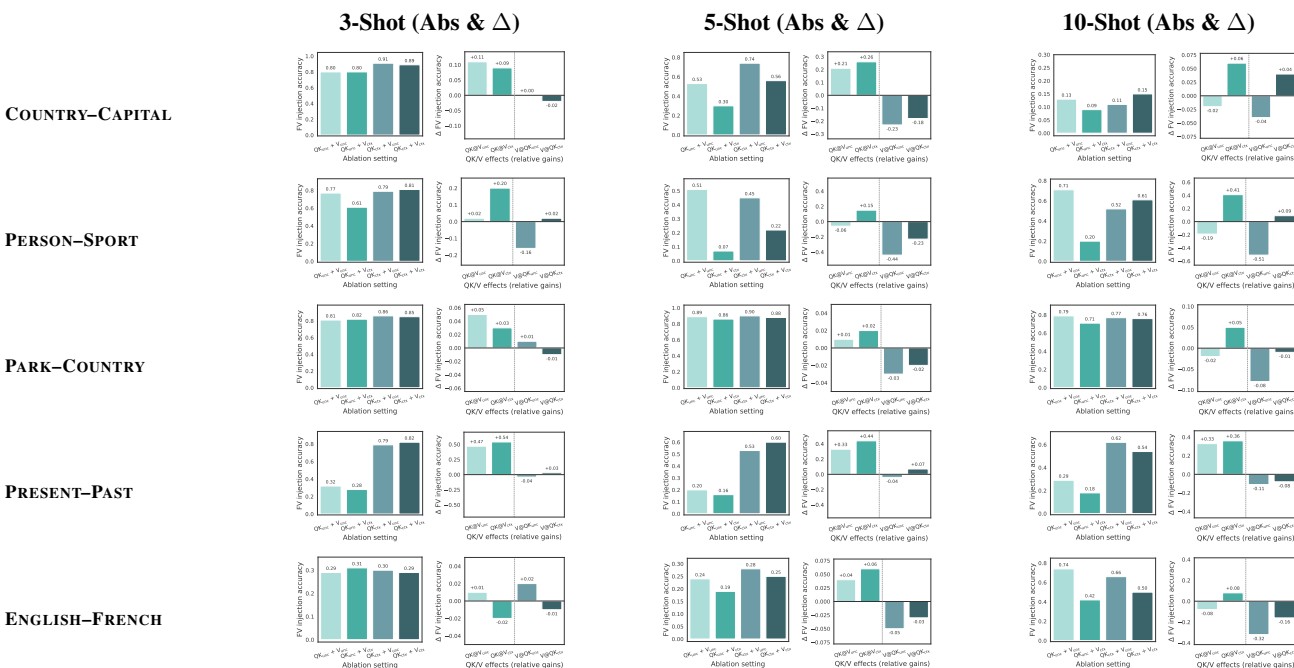

*Figure 42.* Causal Decomposition of Contextualization Gains on `gemma-2-27b`. (Normal Tasks) This is the detailed result of Main Paper Fig. 8.

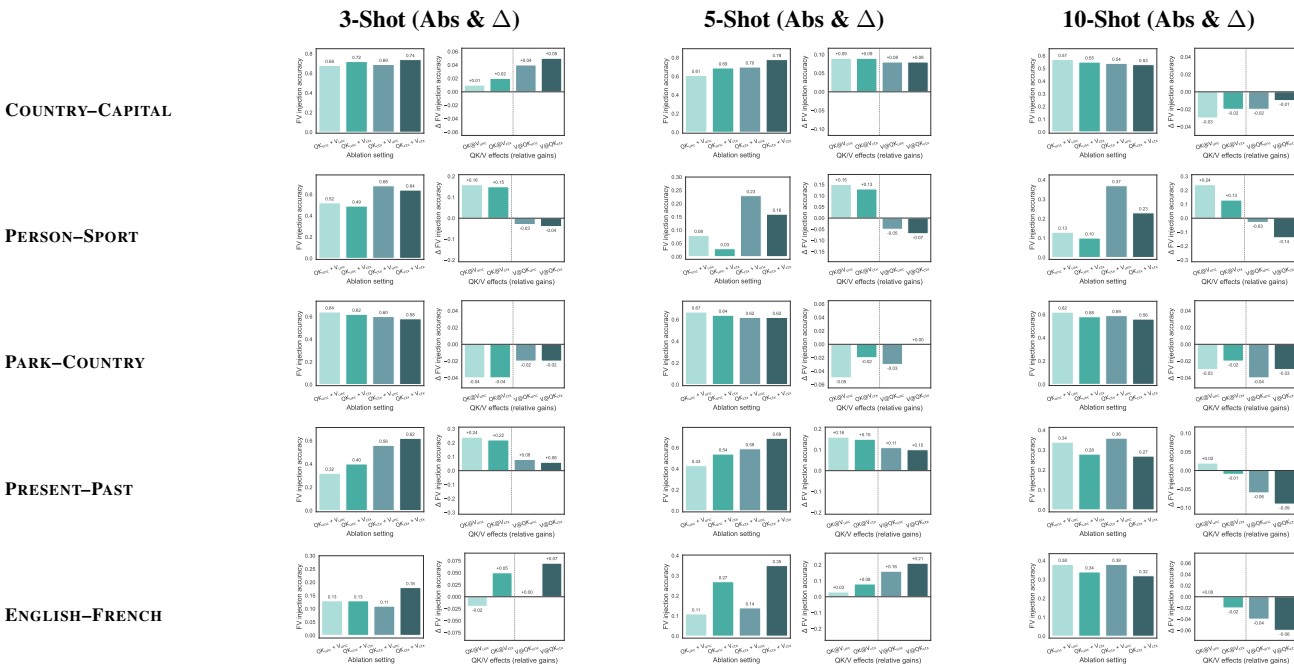

*Figure 43.* Causal Decomposition of Contextualization Gains on `Llama-3.2-1B`. (Normal Tasks) This is the detailed result of Main Paper Fig. 8.

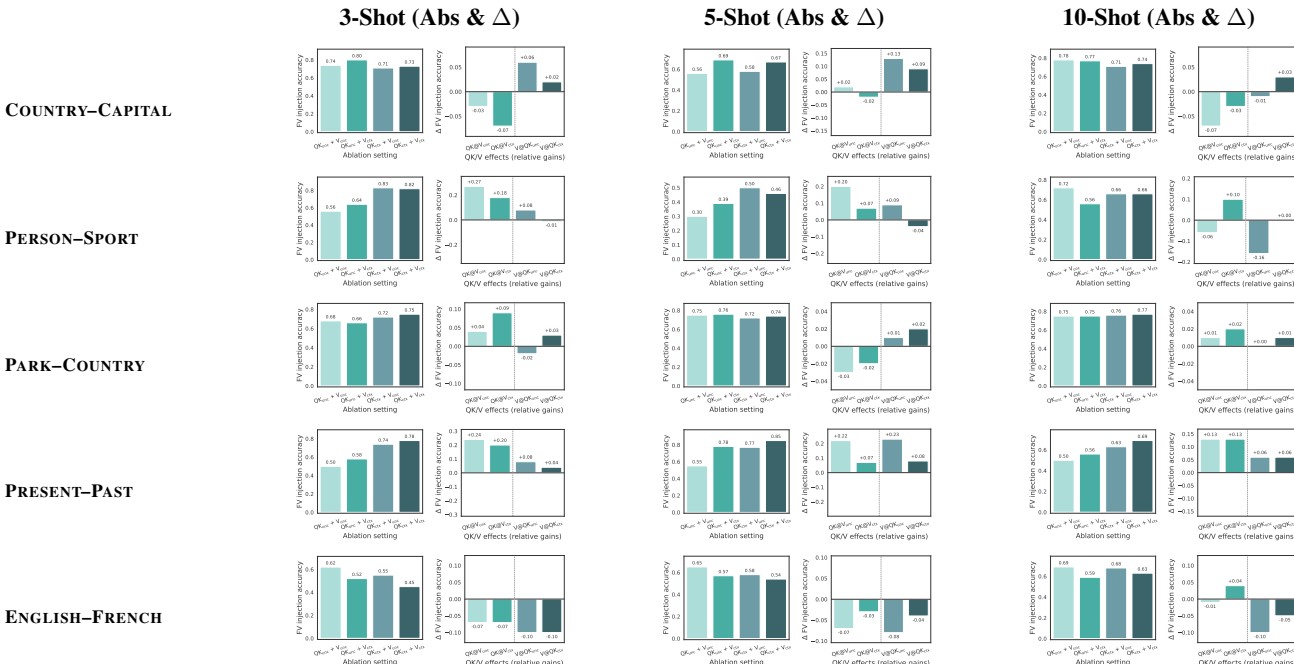

*Figure 44.* Causal Decomposition of Contextualization Gains on `Llama-3.2-3B`. (Normal Tasks) This is the detailed result of Main Paper Fig. 8.

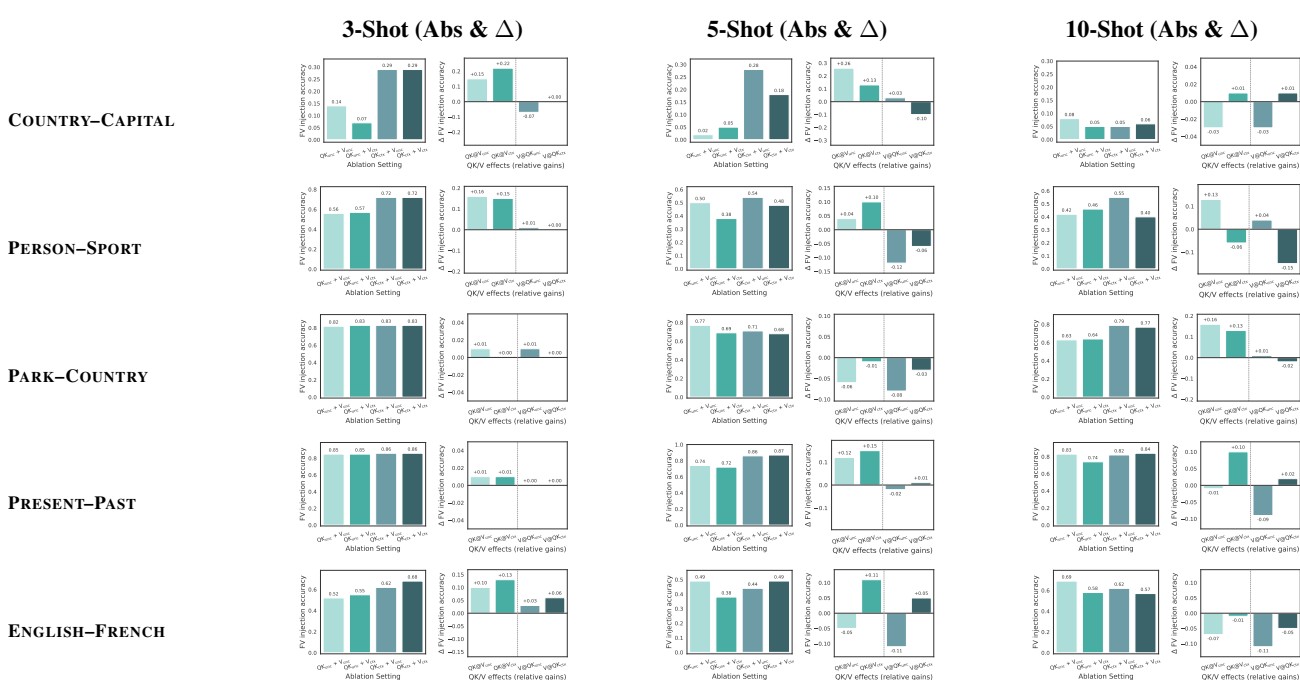

*Figure 45.* Causal Decomposition of Contextualization Gains on `Llama-3.1-8B-Instruct`. (Normal Tasks) This is the detailed result of Main Paper Fig. 8.

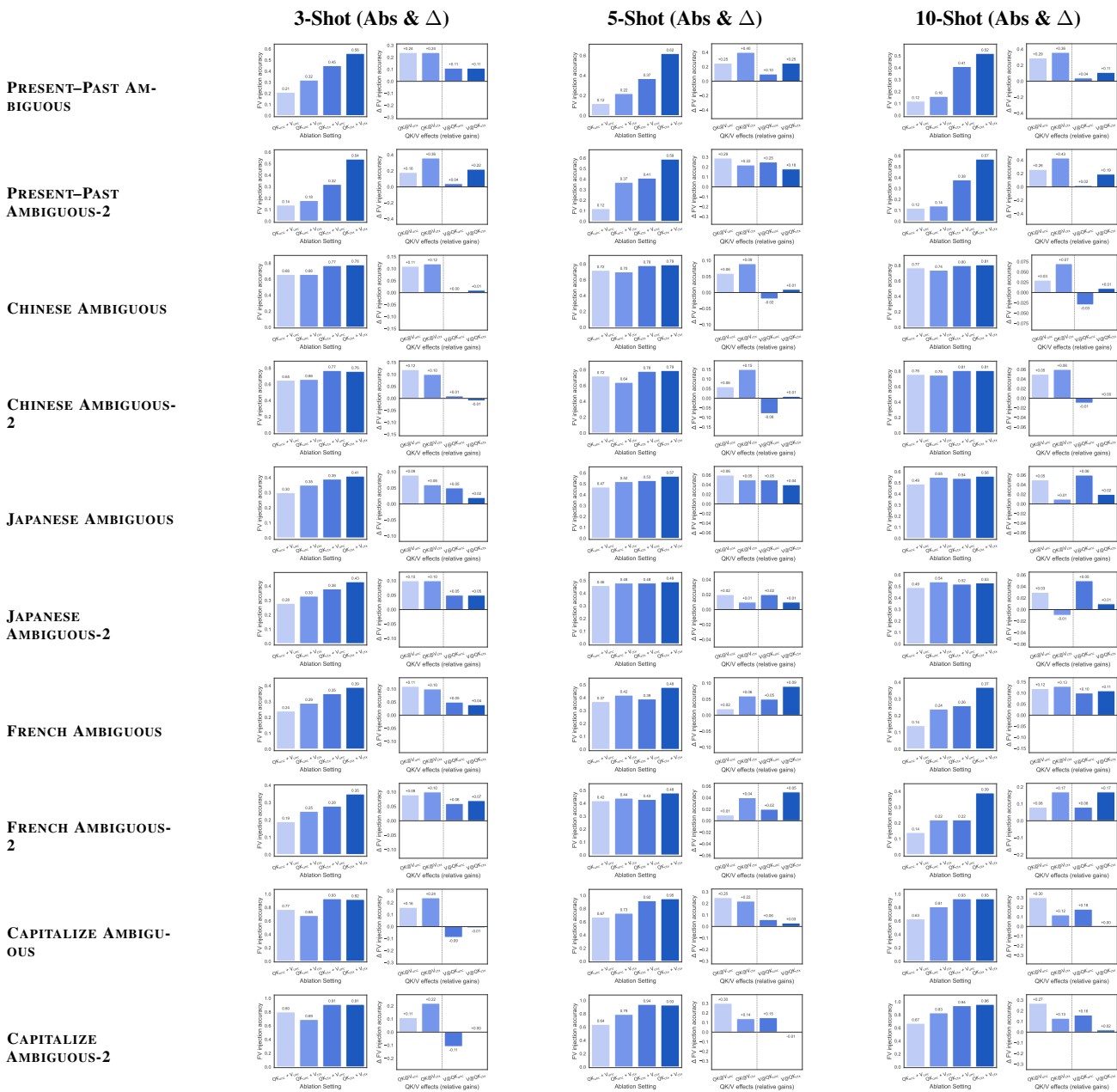

*Figure 46.* Causal Decomposition of Contextualization Gains on `gemma-2-2b`. (Ambiguous Tasks) For each task and shot count, the **left plot** shows absolute FV injection accuracy $F(QK, V)$ across four factorial intervention settings: (1) *Uncontextualized* $F(0,0)$; (2) *QK-contextualized* $F(1,0)$; (3) *V-contextualized* $F(0,1)$; (4) *Full contextualized* $F(1,1)$. We plot the marginal effects: $QK@V_{unc} := F(1,0) - F(0,0)$, $QK@V_{ctx} := F(1,1) - F(0,1)$, $V@QK_{unc} := F(0,1) - F(0,0)$, $V@QK_{ctx} := F(1,1) - F(1,0)$. The Shapley values are then $\phi_{QK} := \frac{1}{2}(QK@V_{unc} + QK@V_{ctx})$, $\phi_V := \frac{1}{2}(V@QK_{unc} + V@QK_{ctx})$. The **right plot** displays the marginal gains ($\Delta$) used in the Shapley decomposition: the first two bars show the effect of contextualizing the QK pathway ($\phi_{QK}$ components) under different V settings, while the latter two show the effect of contextualizing the V pathway ($\phi_V$ components) under different QK settings. This is the detailed result of Main Paper Fig. 8.

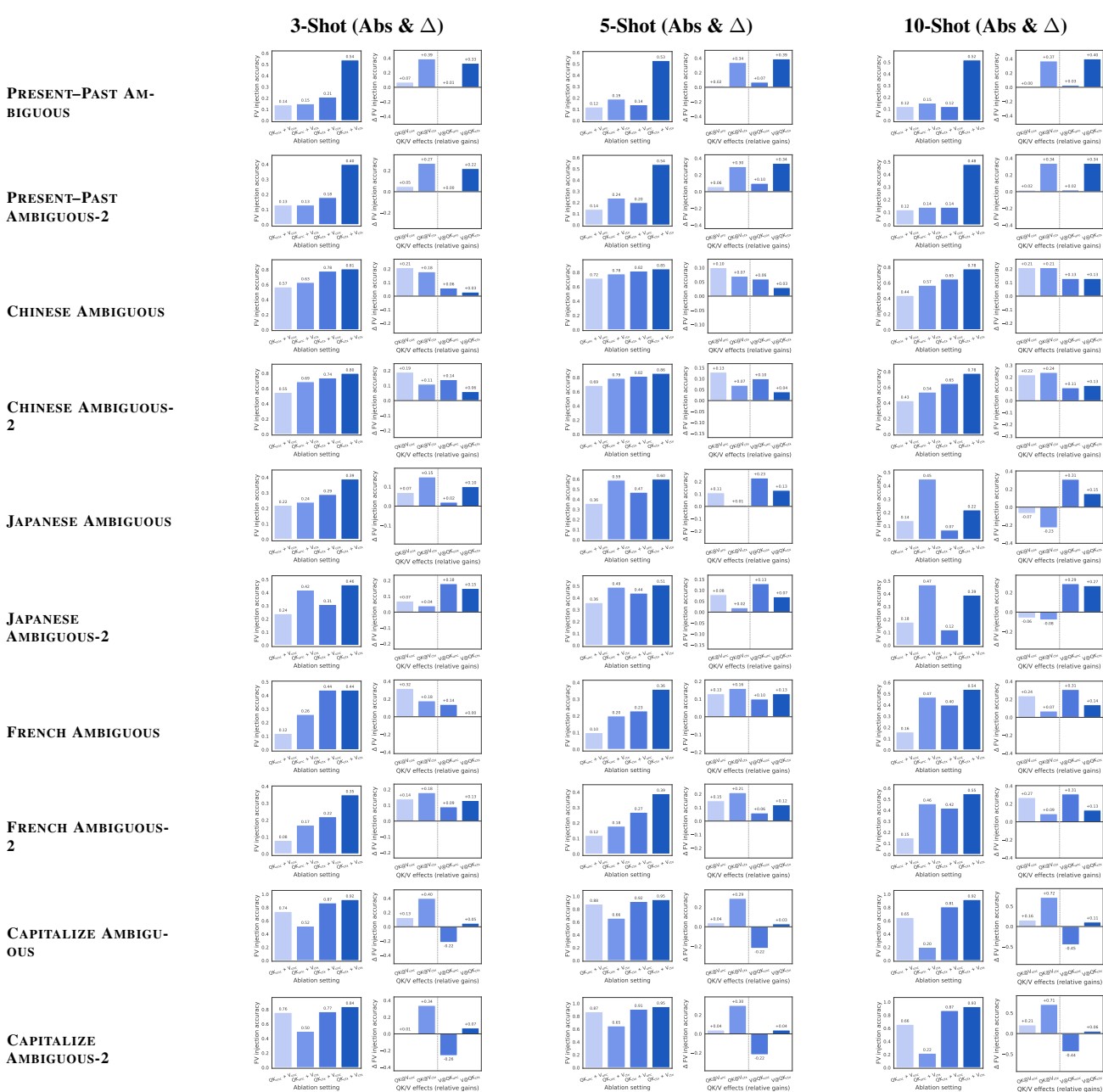

*Figure 47.* Causal Decomposition of Contextualization Gains on `gemma-2-9b`. (Ambiguous Tasks) This is the detailed result of Main Paper Fig. 8.

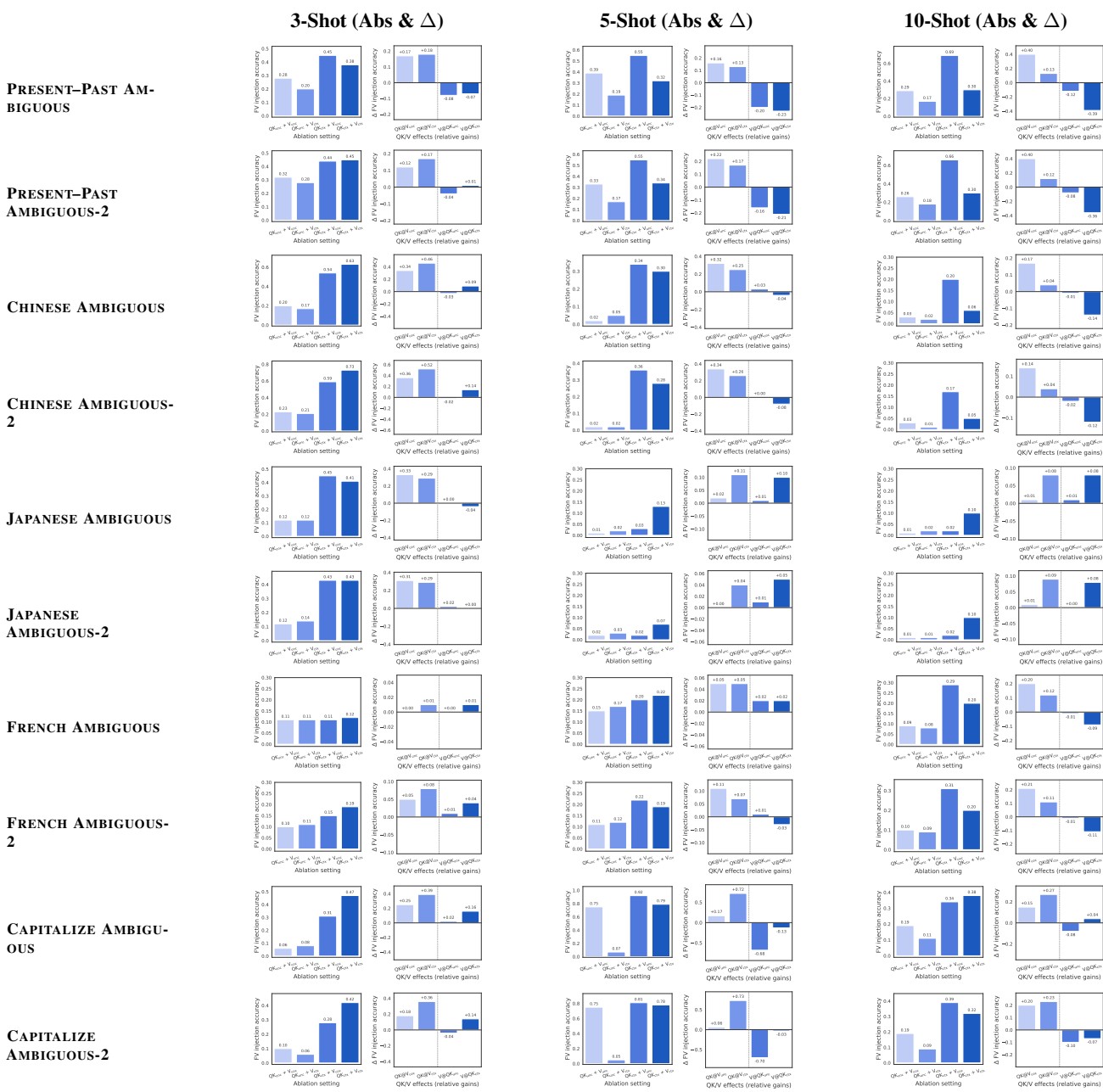

*Figure 48.* Causal Decomposition of Contextualization Gains on `gemma-2-27b`. (Ambiguous Tasks) This is the detailed result of Main Paper Fig. 8.

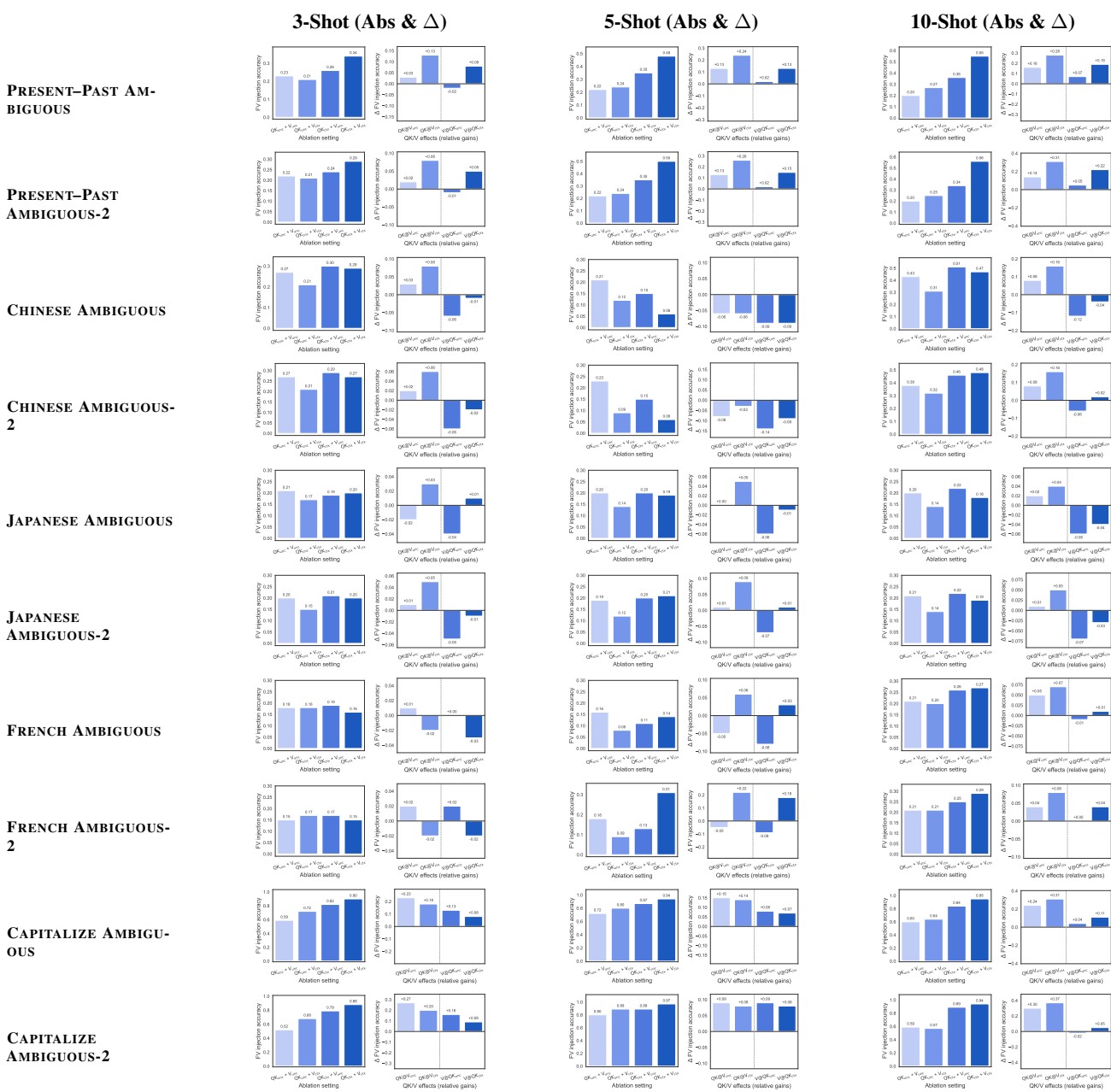

*Figure 49.* Causal Decomposition of Contextualization Gains on `Llama-3.2-1B`. (Ambiguous Tasks) This is the detailed result of Main Paper Fig. 8.

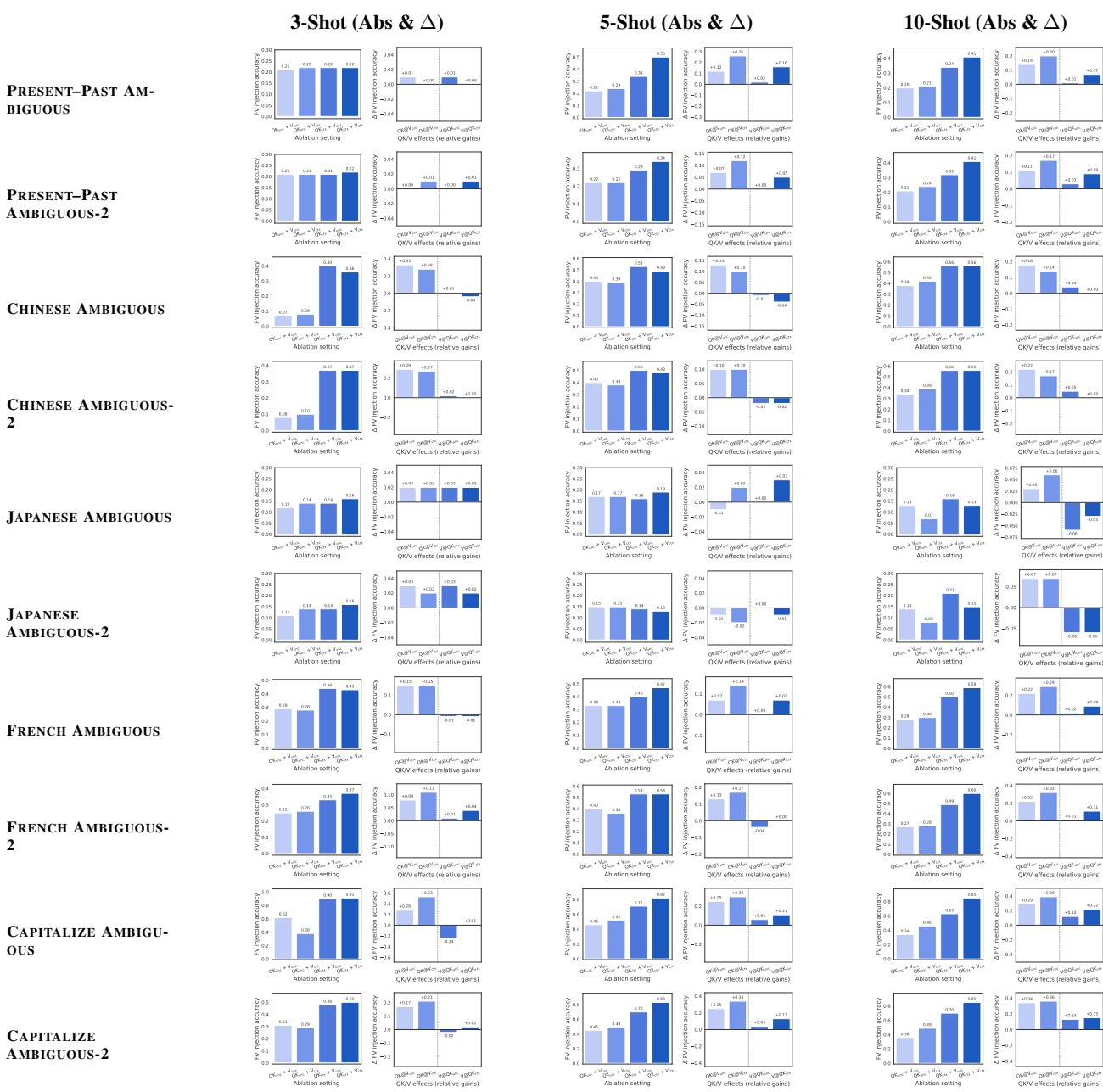

*Figure 50.* Causal Decomposition of Contextualization Gains on `Llama-3.2-3B`. (Ambiguous Tasks) This is the detailed result of Main Paper Fig. 8.

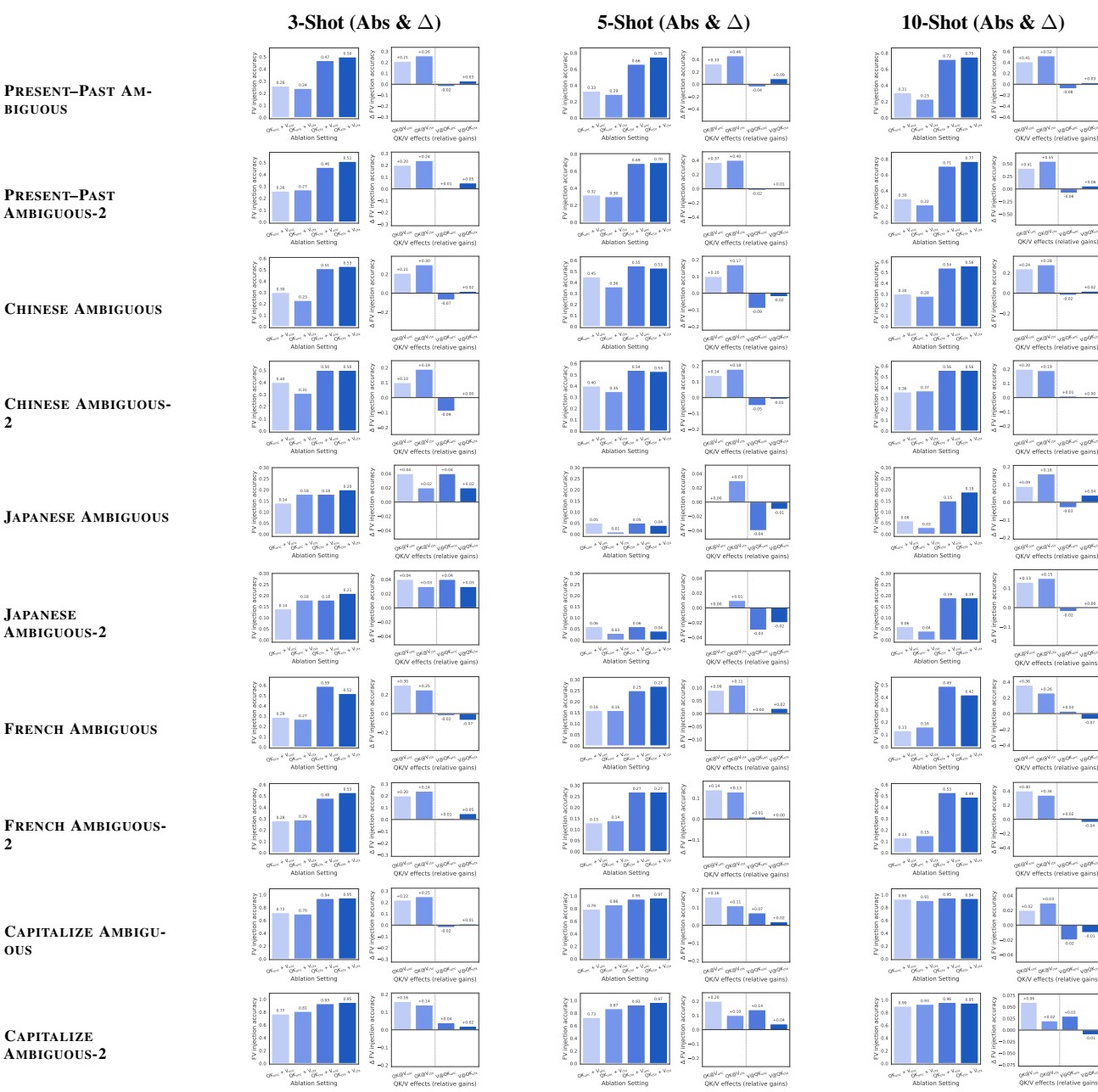

*Figure 51.* Causal Decomposition of Contextualization Gains on `Llama-3.1-8B-Instruct`. (Ambiguous Tasks) This is the detailed result of Main Paper Fig. 8.

## J. Supplemental Evidence for Section 6: dissecting the origins of Query–Key alignment within ICL prompt components

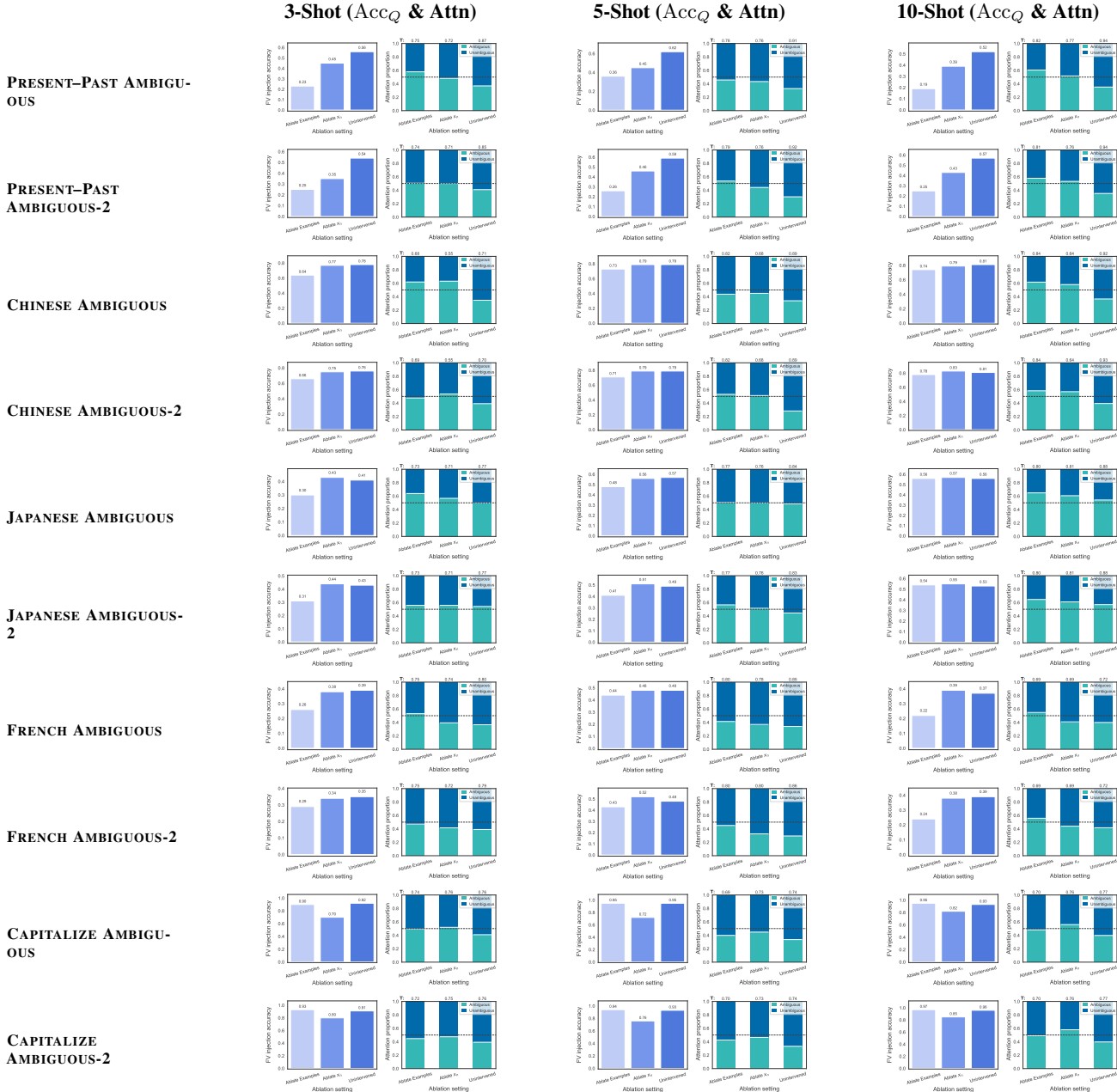

*Figure 52.* Query-patching FV Injection Accuracy ($\text{Acc}_Q$) and Attention Proportion for Ambiguous Tasks on `gemma-2-2b`. Each shot compares the Query-patching FV injection accuracy (left) with the attention mass allocation (right). This is an extension of Main Paper Fig. 9.

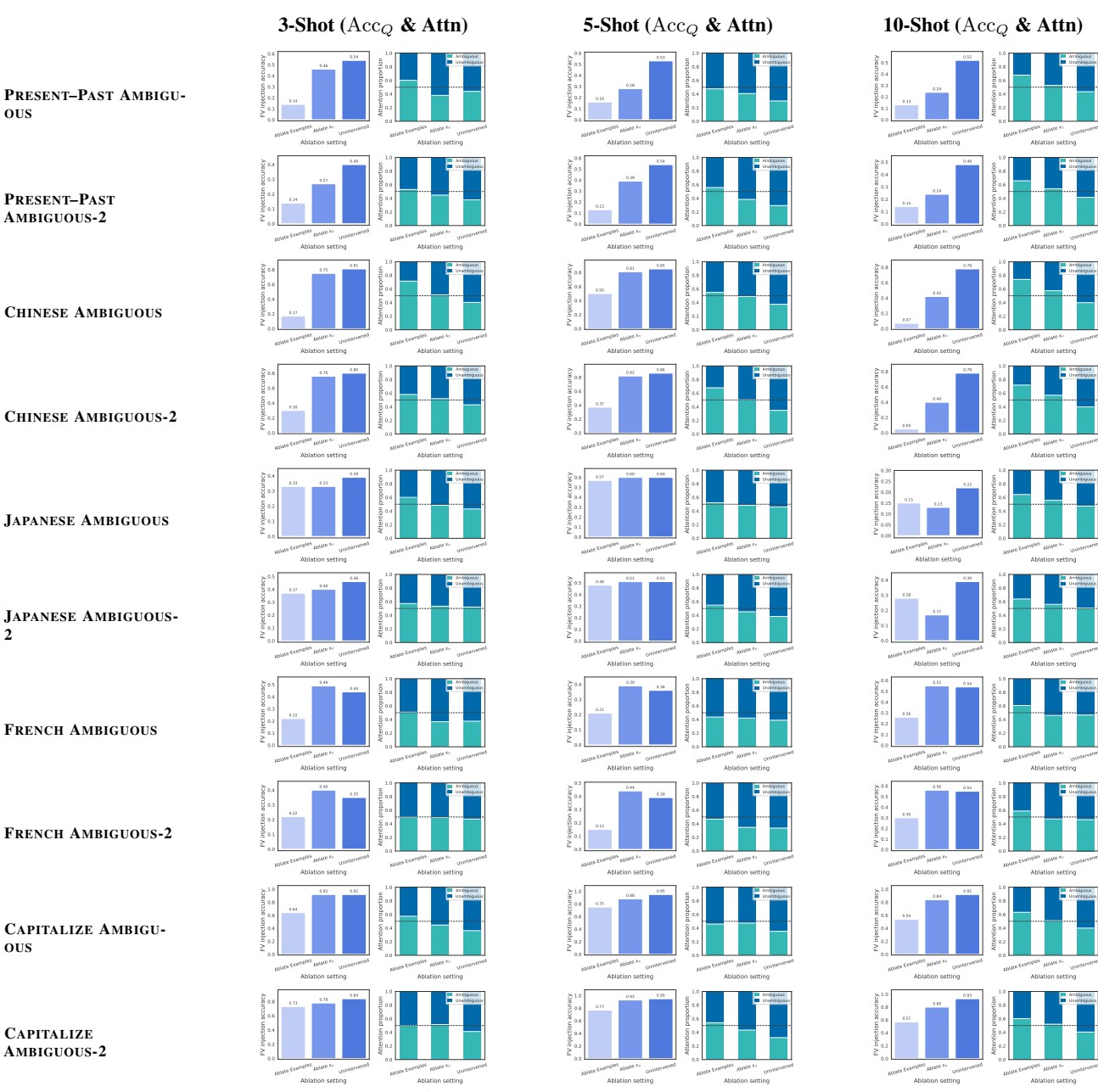

*Figure 53.* Query-patching FV Injection Accuracy ($\text{Acc}_Q$) and Attention Proportion for Ambiguous Tasks on `gemma-2-9b`. Each shot compares the Query-patching FV injection accuracy (left) with the attention mass allocation (right). This is an extension of Main Paper Fig. 9.

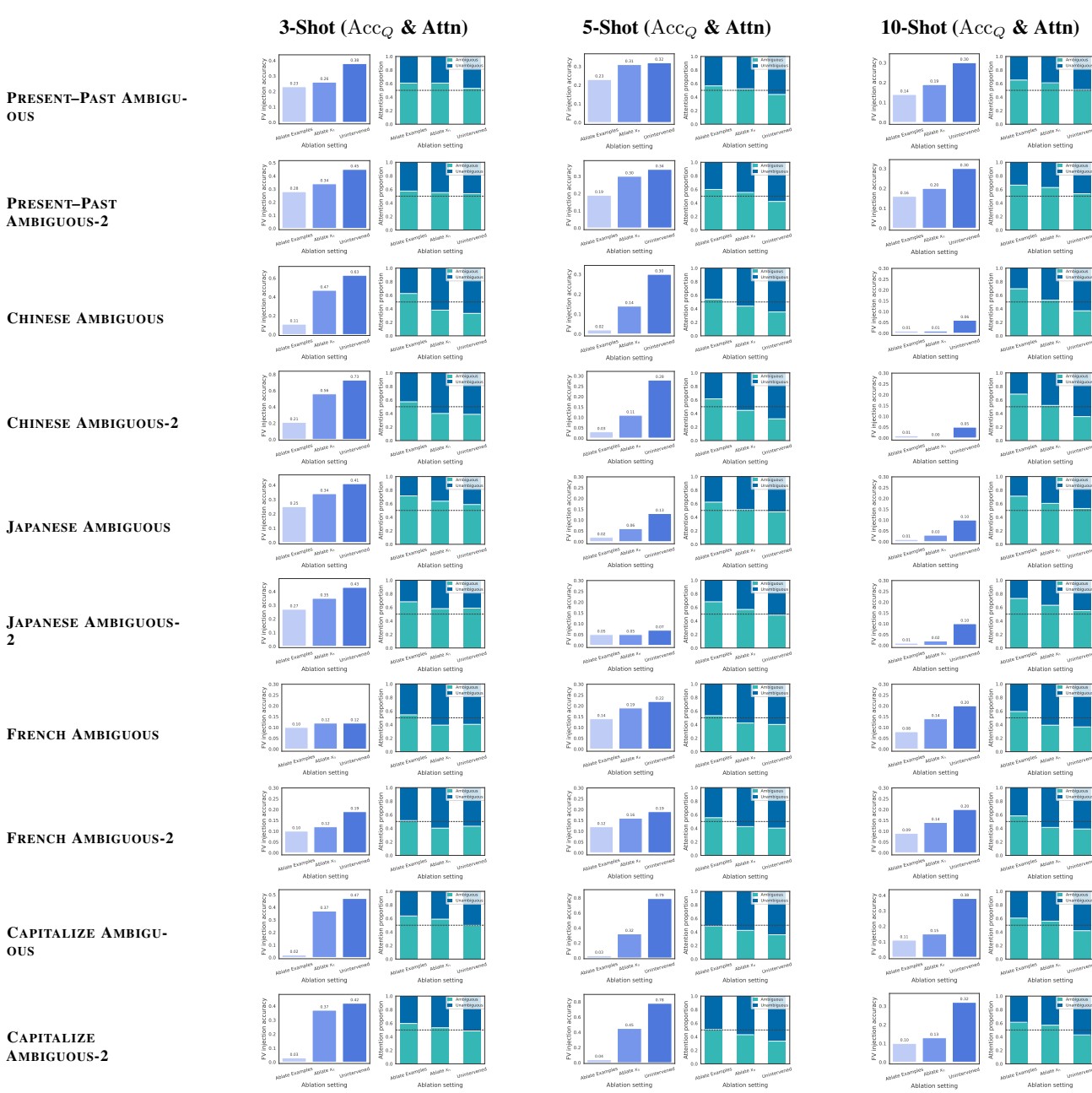

*Figure 54.* Query-patching FV Injection Accuracy (Acc$_Q$) and Attention Proportion for Ambiguous Tasks on `gemma-2-27b`. Each shot compares the Query-patching FV injection accuracy (left) with the attention mass allocation (right). This is an extension of Main Paper Fig. 9.

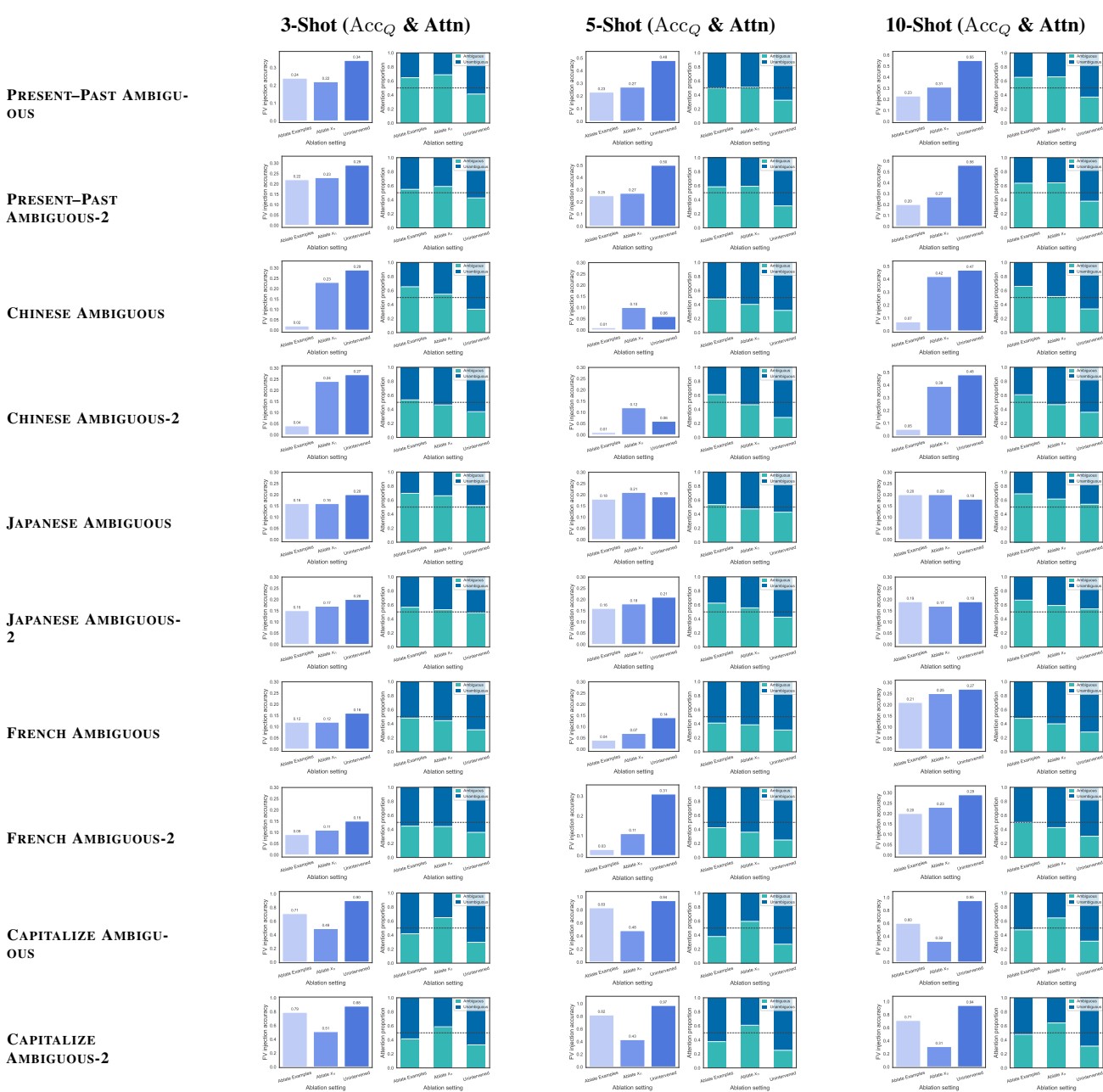

*Figure 55.* Query-patching FV Injection Accuracy ($\mathrm{Acc}_Q$) and Attention Proportion for Ambiguous Tasks on `Llama-3.2-1B`. Each shot compares the Query-patching FV injection accuracy (left) with the attention mass allocation (right). This is an extension of Main Paper Fig. 9.

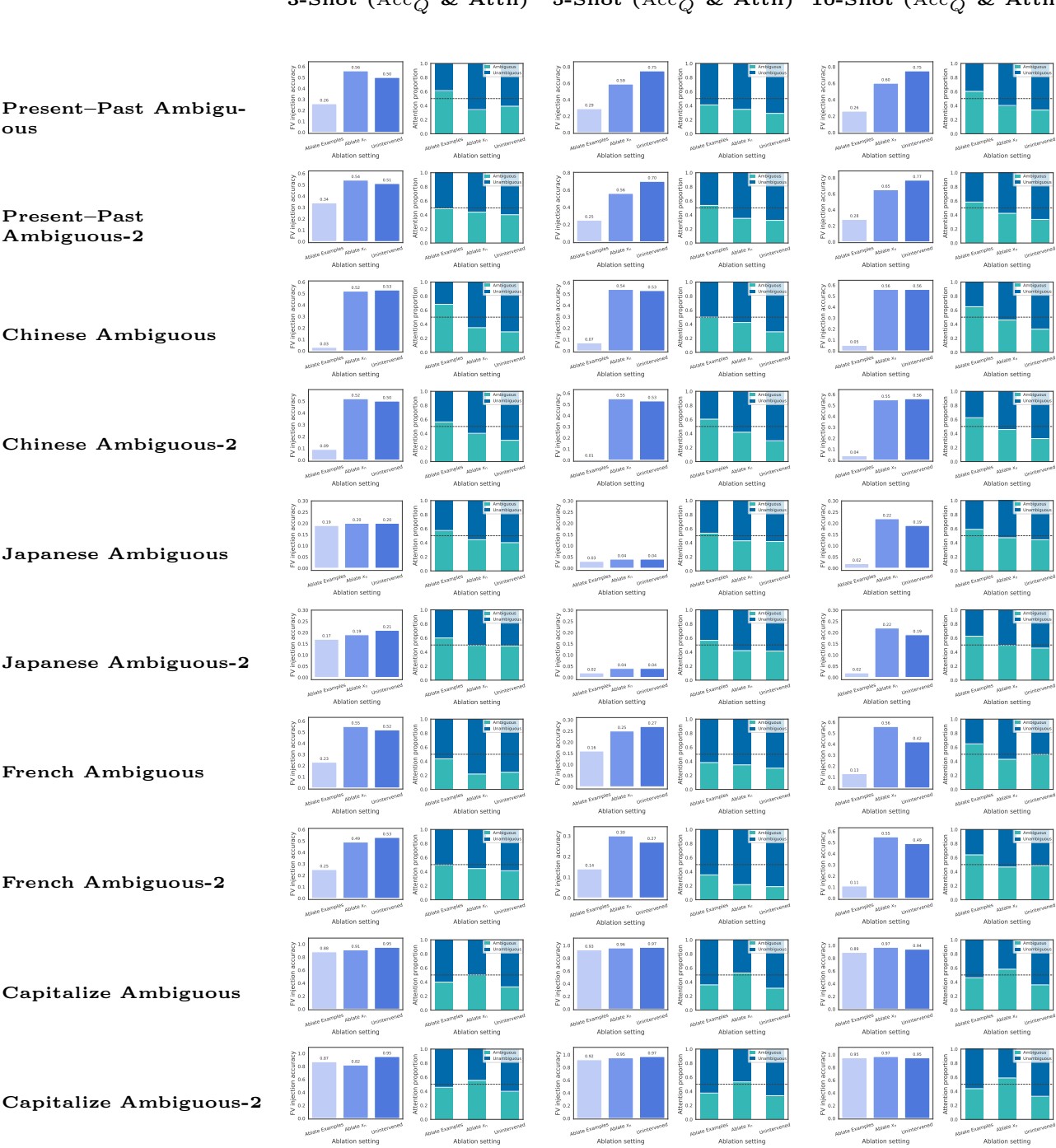

*Figure 56.* Query-patching FV Injection Accuracy ($\mathrm{Acc}_Q$) and Attention Proportion for Ambiguous Tasks on `Llama-3.2-3B`. Each shot compares the Query-patching FV injection accuracy (left) with the attention mass allocation (right). This is an extension of Main Paper Fig. 9.

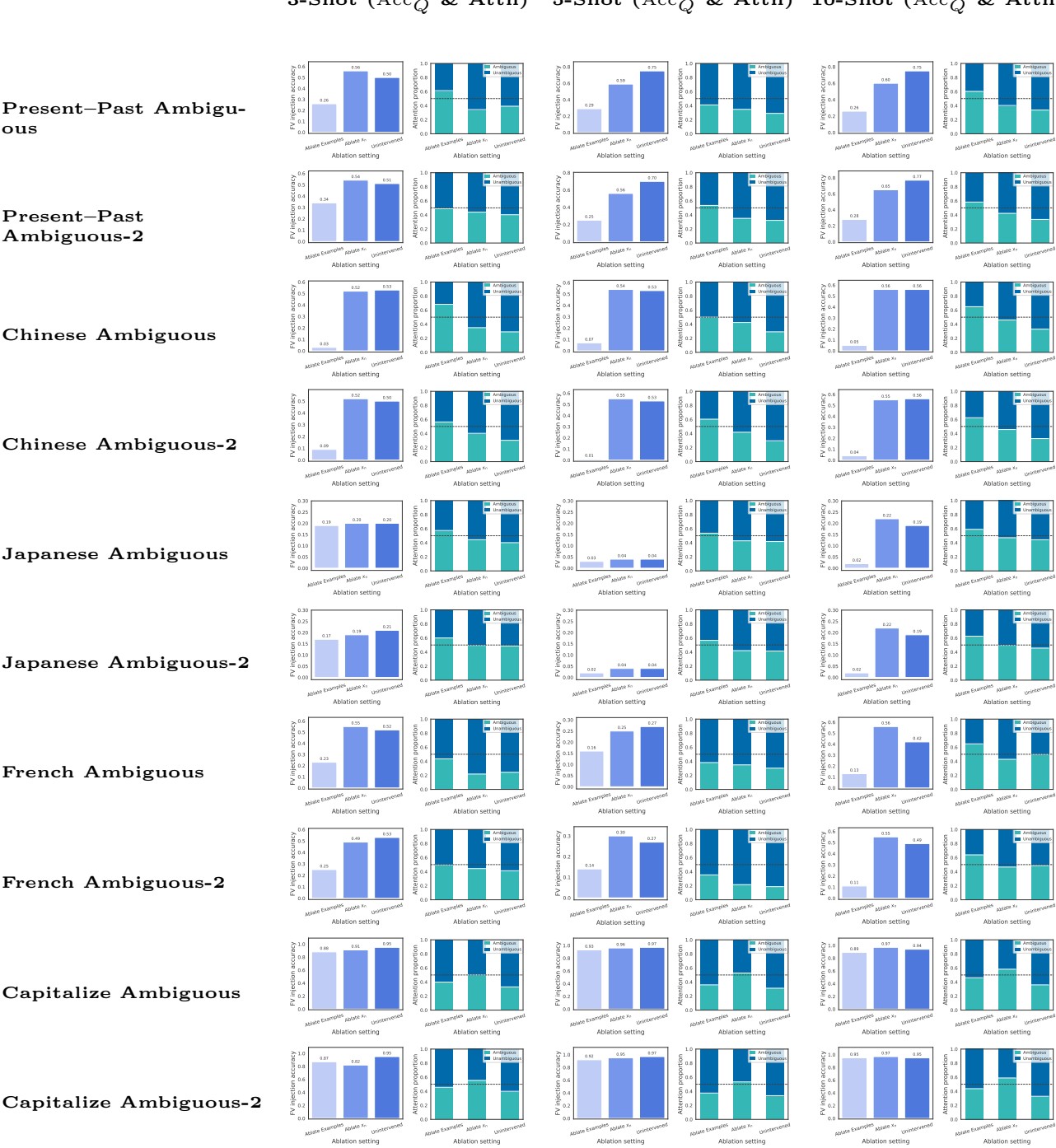

*Figure 57.* Query-patching FV Injection Accuracy ($\mathrm{Acc}_Q$) and Attention Proportion for Ambiguous Tasks on `Llama-3.1-8B-Instruct`. Each shot compares the Query-patching FV injection accuracy (left) with the attention mass allocation (right). This is an extension of Main Paper Fig. 9.

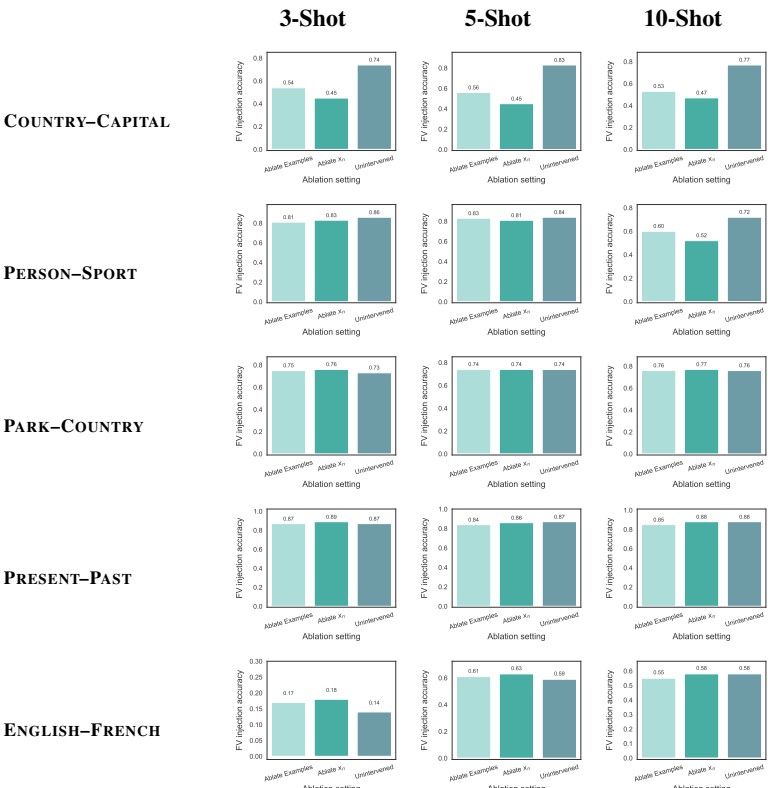

*Figure 58.* Query-patching FV Injection Accuracy ($\text{Acc}_Q$) for Normal Tasks on `gemma-2-2b` across few-shot settings. This is an extension of Main Paper Fig. 9.

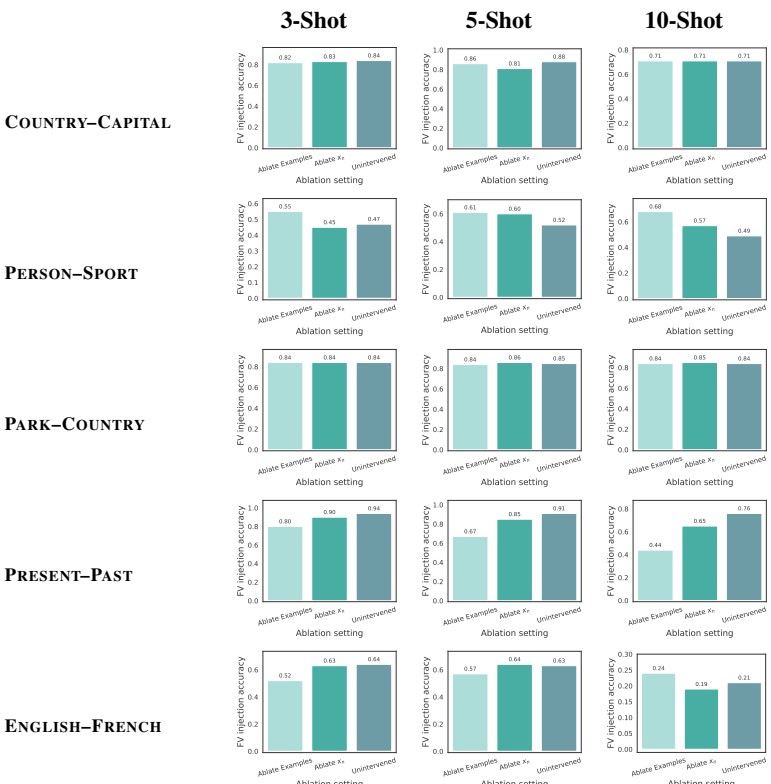

*Figure 59.* Query-patching FV Injection Accuracy ($\text{Acc}_Q$) for Normal Tasks on `gemma-2-9b` across few-shot settings. This is an extension of Main Paper Fig. 9.

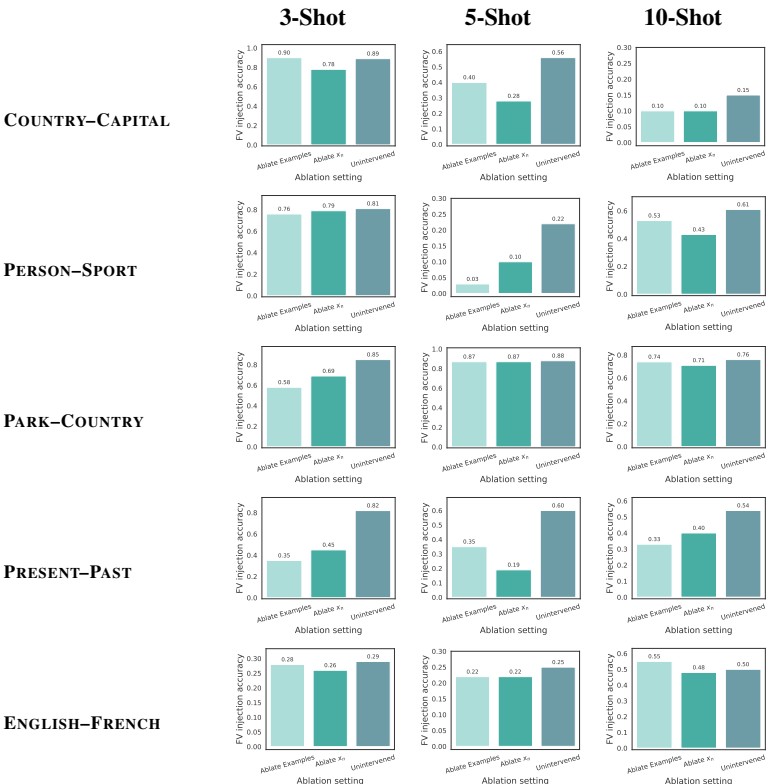

*Figure 60.* Query-patching FV Injection Accuracy (Acc$_Q$) for Normal Tasks on `gemma-2-27b` across few-shot settings. This is an extension of Main Paper Fig. 9.

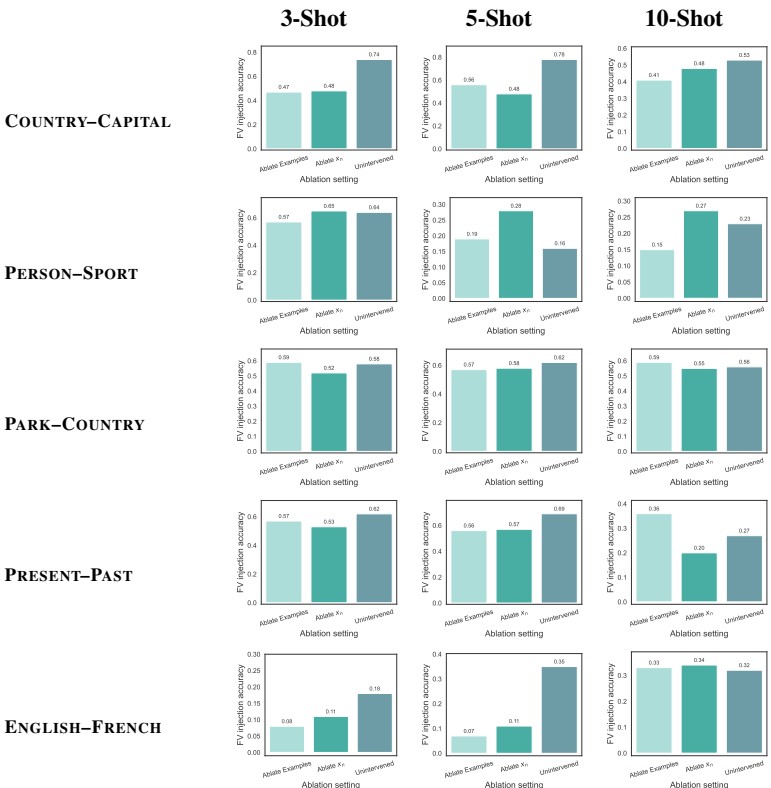

*Figure 61.* Query-patching FV Injection Accuracy (Acc$_Q$) for Normal Tasks on `Llama-3.2-1B` across few-shot settings. This is an extension of Main Paper Fig. 9.

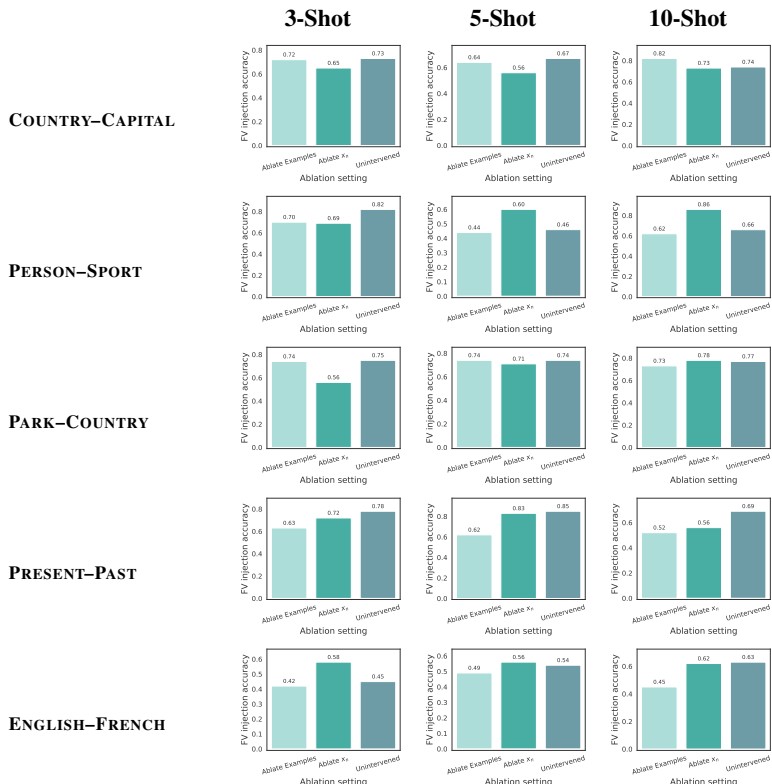

*Figure 62.* Query-patching FV Injection Accuracy ($\text{Acc}_Q$) for Normal Tasks on `Llama-3.2-3B` across few-shot settings. This is an extension of Main Paper Fig. 9.

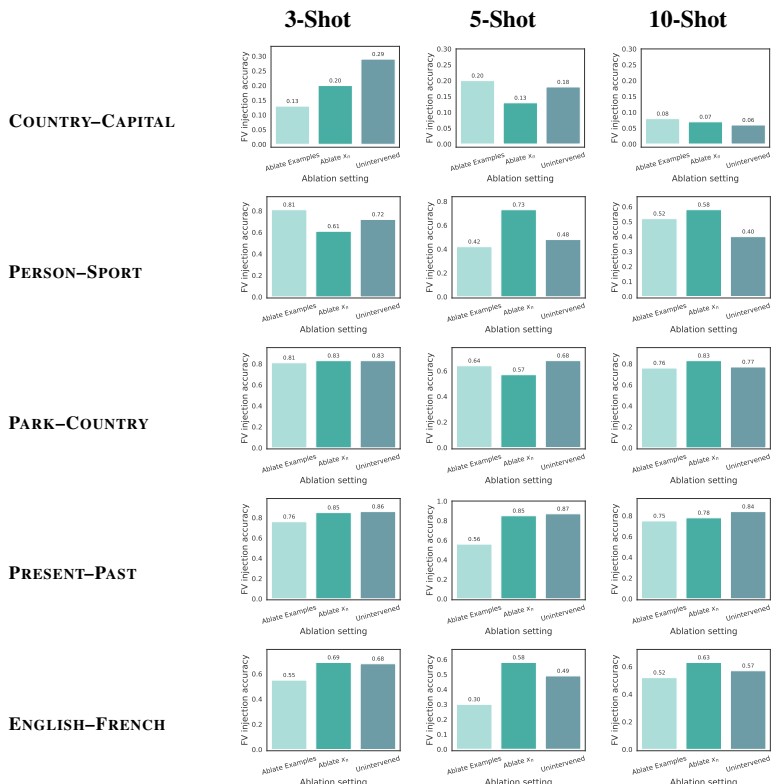

*Figure 63.* Query-patching FV Injection Accuracy ($\text{Acc}_Q$) for Normal Tasks on `Llama-3.1-8B-Instruct` across few-shot settings. This is an extension of Main Paper Fig. 9.

# K. Beyond task identity: Semantic subspace-consistent generalization of Query–Key alignment

The results of Sec.6 showed that, on ambiguous datasets, FV quality and injection accuracy are highly correlated to how attention is apportioned between ambiguous and unambiguous examples, and that this sensitivity is largely mediated by Query–Key alignment. In contrast, many normal datasets either show strong redundancy between the query token $x_{n+1}$ and the examples or are almost insensitive to example corruption. Motivated by this asymmetry, our initial explorations here attempt to probe a more fine-grained question: *what aspects of the example **content** are potentially necessary to induce Query* states that align with the FV circuit? Specifically, we investigate whether examples must instantiate the exact input–output mapping of the nominal task, or if it might suffice for them to position the Query within a broader, appropriate semantic subspace.

In this preliminary study, we fix the original ICL query $x_n$ and systematically vary only the mapping implemented by the examples. For each dataset and $n$-shot setting, we construct several synthetic example sets by remapping the input–output pairs in controlled ways, including:

- preserving the original mapping ($x \rightarrow y$),

- input-only or output-only mappings ($x \rightarrow x$, $y \rightarrow y$),

- corrupted-mappings ($y$ is still in the correct output space but does not correspond to $x$, i.e., corrupted $x \rightarrow y$),

- cross-mappings ($y \rightarrow x$, $x \rightarrow$ other task $y$, other task $x \rightarrow y$),

- examples drawn from completely different tasks ("other task")

All variants keep the same prompt format, while only the semantic relation between example inputs and outputs is altered to observe potential shifts in behavior. We extracted the Queries from these synthetic prompts and applied **Q patching** on the original prompt.

Across the majority of datasets and shot counts, we observe that several synthetic mappings that do *not* implement the original task appear to still induce Query–Key alignment and FV injection accuracy comparable to the original task. In many cases, mappings such as $x \rightarrow x$, $y \rightarrow y$, or $y \rightarrow x$ example sets yield attention allocations over examples and FV accuracy that are within a few points of the original $x \rightarrow y$ prompt, and occasionally even slightly higher. One possible interpretation of these observations is that the model may not strictly require a unique, task-identifying Query state. Instead, it is plausible that as long as examples steer the Query into a relevant region of the representation space, the FV heads can potentially recover a high-quality FV. Functionally, the model seems to exhibit signs of pattern matching in a shared semantic manifold rather than necessarily identifying a discrete task label.

The exceptions may also provide further nuance. For instance, in the CAPITALIZATION AMBIGUOUS task, even some cross-task or cross-mapping constructions perform well, which could hint that the underlying manifold is structured and anisotropic. These asymmetries, while appearing sporadically, point toward a tentative picture where the model exploits a continuous semantic space with preferred directions, rather than a single discrete task representation.

Collectively, these exploratory results suggest the possibility that Query–Key alignment in ICL might not strictly depend on a unique, task-specific Query representation. Instead, a potentially wide family of example-induced Query states–even those only loosely related to the nominal task–could be sufficient to trigger the formation of FV and support injection accuracy. We leave a more rigorous characterization of this manifold for future work.

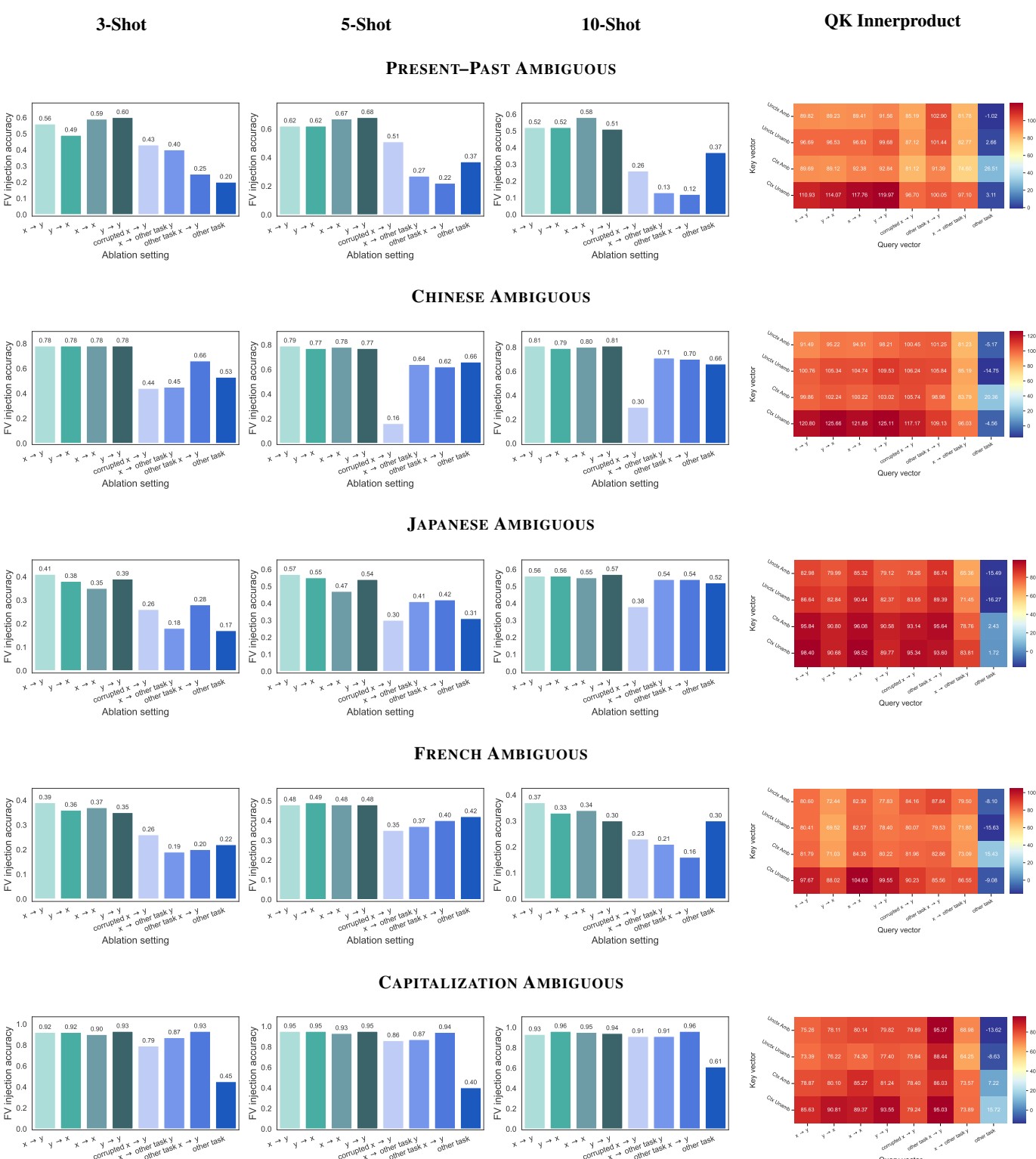

*Figure 64.* **Functional robustness and geometric stability of Query–Key alignment under semantic intervention.** Rows correspond to the five ambiguous task families detailed in Appendix A. **Columns 1–3** report downstream $FV$ injection accuracy after $Q$ patching across 3, 5, and 10-shot contexts. **Column 4** presents the mean inner product between $Q$ and $K$ vectors on the Top-1 $FV$-head, averaged over 100 trials. The matrix visualizes the directional alignment of Queries derived from prompts with diverse semantics against Keys from randomly sampled ambiguous and unambiguous examples under uncontextualized(**unc**)/contextualized(**ctx**) ablation. These results are acquired from `gemma-2-2b`.

## L. How does contextualization reshape Value representations?

In the main paper and Appendix I, we noted that contextualization primarily enhances FV quality through the Query–Key pathway, while the effects of the Value pathway appear more task-dependent and could be a double-edged sword. To tentatively probe this complexity, this section offers a preliminary look at the Value pathway from a purely geometric perspective: given a fixed, contextualized attention routing, how does contextualization shift the function vector in the activation space?

We adopt a framework at the dataset level: for each task, we aggregate per-prompt FVs into two dataset-level summaries

$$\text{FV}_{\text{unc}} \;=\; \frac{1}{|P|} \sum_{p \in P} \text{FV}\big(p; QK_{\text{ctx}}, V_{\text{unc}}\big)$$

$$\text{FV}_{\text{ctx}} \;=\; \frac{1}{|P|} \sum_{p \in P} \text{FV}\big(p; QK_{\text{ctx}}, V_{\text{ctx}}\big),$$

where $P$ is the set of prompts for the task, and $\text{FV}_{\text{unc}}$ and $\text{FV}_{\text{ctx}}$ denote FVs obtained under uncontextualized and contextualized conditions, respectively. Working with these dataset-level averages smooths over per-prompt idiosyncrasies and lets us characterize how Value rewriting changes the *typical* task direction systematically.

Crucially, we keep the Query–Key pathway fixed to its fully contextualized state, $QK_{\text{ctx}}$. Earlier results showed that contextualized QK routing systematically attends more to informative, unambiguous examples and mitigates positional bias, yielding higher FV injection accuracy overall. Fixing $QK_{\text{ctx}}$ therefore serves two purposes: it uses the empirically better attention pattern as a stable baseline, and it cleanly separates the effect of Value variations on the content of the FV from changes in *which* examples are attended to. To interpret these changes, we compare $\text{FV}_{\text{unc}}$ with $\text{FV}_{\text{ctx}}$ directly. Additionally, for ambiguous tasks, we exploit the natural split into all-ambiguous and all-unambiguous prompts and define $\text{FV}_{Amb}$ and $\text{FV}_{Unamb}$ as the average FVs over these two subsets. We intentionally avoid defining a per-prompt "golden" direction (e.g., by picking whichever variant attains higher accuracy), as that would entangle geometric analysis with the very causal effects we aim to keep separate.

Across all models, tasks, and shot counts, the contextualized and uncontextualized FVs remain strongly aligned: the cosine similarity $\cos(\text{FV}_{\text{ctx}}, \text{FV}_{\text{unc}})$ lies between roughly $0.75$ and even $0.98$. Thus contextualization produces a *noticeable but not drastic* variation of the FV rather than flipping the FV into an orthogonal or opposite direction. In other words, Value contextualization refines the task vector within its existing subspace instead of rewriting it into a qualitatively different representation.

On normal datasets, we observe a clear dependence on the number of shots. For 3-shot prompts, contextualization moves the FV only mildly: the cosine similarity between $\text{FV}_{\text{ctx}}$ and $\text{FV}_{\text{unc}}$ is typically above $0.9$. As we increase to 5-shot and 10-shot prompts, the cosine similarity drops systematically, indicating that Value activations undergo progressively larger adjustments when more demonstrations are available. Intuitively, longer prompts provide more opportunities for cross-example contextualization, and the Value pathway uses this extra context to reshape the aggregated task representation more strongly. For ambiguous datasets, we gain a more structured picture by comparing FVs to $\text{FV}_{Amb}$ and $\text{FV}_{Unamb}$. First, $\text{FV}_{Amb}$ and $\text{FV}_{Unamb}$ themselves are highly aligned: their cosine similarity typically lies well above $0.6$, and often above $0.8$, indicating that ambiguous and unambiguous prompts largely inhabit a shared task subspace, differing only by a modest offset that reflects the presence or absence of example ambiguity. Within this shared subspace, we find that contextualization tends to move the contextualized FV closer to *both* prototypes ($\text{FV}_{Amb}$ and $\text{FV}_{Unamb}$) in terms of cosine similarity, but with a greater and more consistent improvement toward the all-unambiguous prototype. Concretely, for most datasets and shot counts we observe that

$$\cos(\text{FV}_{Unamb}, \text{FV}_{\text{ctx}}) \gtrsim \cos(\text{FV}_{Amb}, \text{FV}_{\text{ctx}})$$

We stress that these geometric trends are consistent with, but not by itself sufficient to explain, the causal shifts in FV injection performance. A trend observed in dataset-averaged FVs may not uniformly translate to individual prompts, where an "ambiguous offset" might still play a functional role. In summary, these early-stage geometric analyses hint at a coherent but nuanced picture: Value contextualization appears to perform a moderate rotation of the task vector–further investigation is needed to determine how these geometric shifts directly translate into the "help vs harm" dual effects noted in our main analysis.

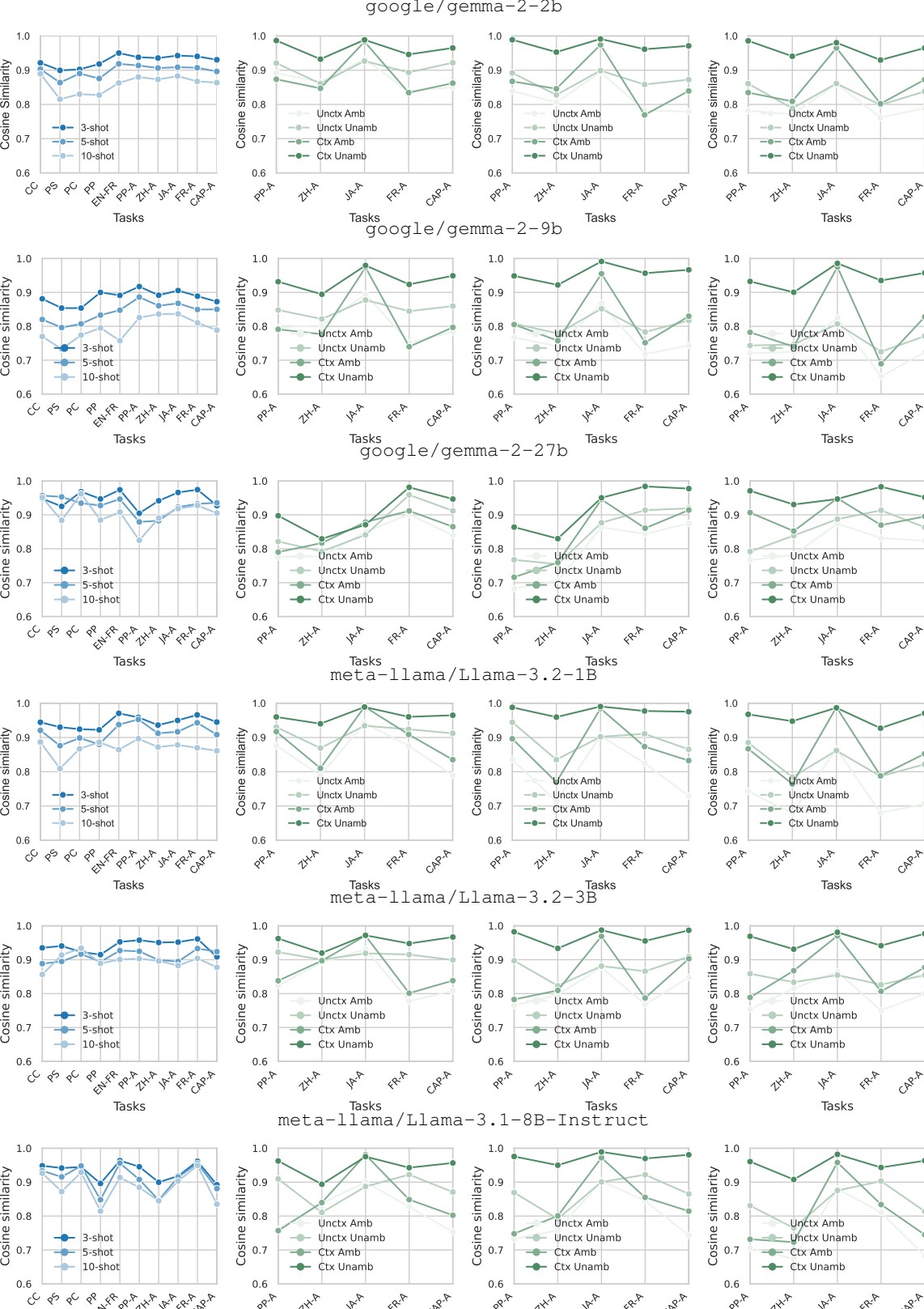

*Figure 65.* Each row represents a model, with columns showing: **(1)** The cosine similarity between QK fixed uncontextualized and contextualized FVs ($\cos(FV_{unc}, FV_{ctx})$), demonstrating that Value contextualization refines the task vector within its existing subspace; **(2–4)** present $2 \times 2$ cosine similarity matrices for ambiguous tasks (at 3, 5, and 10 shots), illustrating the cross-alignment between the $\{FV_{unc}, FV_{ctx}\}$ pair and the prototype FVs $\{FV_{Amb}, FV_{Unamb}\}$. $FV_{ctx}$ and $FV_{Unamb}$ have the highest alignment.

# M. Implications for Theoretical Analysis of ICL

**Data and Input Format** We assume two tasks $F_A, F_B : \mathcal{X} \to \mathcal{Y}$, where $\mathcal{X}$ is a finite discrete space and $\mathcal{Y} := \{-1, 1\}$. We assume each few-shot example is provided as $(x, y) \in \mathcal{X} \times \mathcal{Y}$ provided to the transformer in a single position. An example $(x, y)$ is *ambiguous* if and only if $F_A(x) = F_B(x) = y$. An input prompt $p$ consists of a sequence $(x_1, y_1) \ldots (x_n, y_n)x_{n+1}$, where $F_t(x_i) = y_i$ where $t$ is either $A$ or $B$ (jointly across all $i$). In-context learning is modeled as identifying the task (either $F_A$ or $F_B$) from examples, and applying it to a query $x_{n+1}$ to output $F_t(x_{n+1})$.

We note that the choice of a discrete set of tasks (here, two tasks) on a discrete input space is deliberately different from what is most commonly studied in the theoretical literature on ICL, which typically considers *continuous* task families such as linear regression. We believe that a discrete model is especially relevant to the tasks in our experiments, and to real-world ICL more broadly.

**Model** We consider a one-layer one-head softmax model. There is one MLP before attention (applying to each individual example), and one MLP after attention (applying to the residual connection from $x_{n+1}$ and the output of the attention head):

$$\phi\left(x_{n+1}, \sum_{i=1}^{n} a_i \cdot \psi(x_i, y_i)\right) \tag{2}$$

where $\psi : \mathcal{X} \times \mathcal{Y} \to \mathbb{R}^d$, $\phi : \mathcal{X} \times \mathbb{R}^d \to \mathbb{R}$ are MLPs, which we model as nonparametric $C^1$-continuous functions for tractability; and

$$a_i = \frac{\exp(\alpha(x_{n+1}, x_i, y_i))}{\sum_{j=1}^{n} \exp(\alpha(x_{n+1}, x_j, y_j))} \tag{3}$$

are the attention weights, where $\alpha : \mathcal{X} \times \mathcal{X} \times \mathcal{Y} \to \mathbb{R} \cup \{-\infty, \infty\}$ describes the attention logits.[2] MLPs and attention weights are modeled as general functions $\phi, \psi, \alpha$, which will be theoretically more tractable than neural parameterizations.

Importantly, unlike much theoretical work on ICL that relies on linear transformers (e.g. Von Oswald et al., 2023; Vladymyrov et al., 2024; Mahankali et al., 2024), our model maintains the nonlinearity of transformers, with both nonlinear MLPs and general softmax attention, not linear attention.

**Objective Function** The loss function is mean-squared error between (2) and the task output, with regularizers for the norm of $\psi$ and the smoothness of $\phi$:

$$\mathcal{L} := \underbrace{\mathbb{E}[(\phi - F(x_{n+1}))^2]}_{Prediction\ Error} + \tau \underbrace{\max_{x, y \in \mathcal{X} \times \mathcal{Y}} \|\psi(x, y)\|_2^2}_{Norm\ of\ \psi} + \eta \underbrace{\max_{x_{n+1} \in \mathcal{X}} \sup_{x \in \mathbb{R}^d} \|\nabla_2 \phi(x_{n+1}, x)\|_2}_{Smoothness\ of\ \phi} \tag{4}$$

where $\tau, \eta > 0$ can be arbitrarily chosen, and the expectation is over the data distribution. We do not regularize smoothness in arguments from $\mathcal{X}, \mathcal{Y}$ as we take these to be discrete. Hence, there is no smoothness regularization for $\psi$.

**Result** For this model, we will argue the following structural properties for solutions:

**Theorem M.1.** *Assume some nonzero number of $x \in \mathcal{X}$ is unambiguous. Take $n > 1$, and any $\tau, \eta > 0$. Consider the uniform distribution over $k$-shot prompts for all $k = 1, \ldots, n$. Any global optimum $(\psi, \phi, \alpha)$ of the loss $\mathcal{L}$ has, assuming that it achieves better-than-chance performance, the following structure:*

1. *For unambiguous examples, $\psi(x, y)$ only depends on the task. The attention output $\sum_{i=1}^{n} a_i \cdot \psi(x_i, y_i)$ can be viewed as a FV, in that $\phi(x_{n+1}, \cdot)$ evaluates the task for other queries $x_{n+1}$ as well.*

2. *Whenever a prompt has both ambiguous and unambiguous examples, attention is lower on the ambiguous examples than the unambiguous examples.*

Here, by "better than chance performance" we rule out solutions where regularization is so strong that the optimal solution provides a constant answer $\phi(\cdot)$.

---

[2] Allowing infinite attention logits is an idealizing assumption also made in some other work (Zhou et al., 2024); it ensures the existence of an actual global optimum. An alternative would be to restrict to finite values and consider sequences approximating the optimum.

Note that we are not arguing that the setting, assumptions, and proof strategy suggested here are necessarily the ideal theoretical model of ICL. Rather, the purpose of this discussion is to outline potential directions for future theoretical work on ICL inspired by our mechanistic findings.

*Proof.* We now argue Theorem M.1. While $\mathcal{L}$ is not convex due to the unconstrained nature of $\phi$, we will use a convexity-based argument. The basic idea is that balancing the norm of $\psi$ and the smoothness of $\phi$ is best achieved if $\psi$ provides opposing vectors for the two tasks and $\phi$ interpolates linearly between them. This forces unambiguous examples to map to task-specific vectors; ambiguous examples are then shown not to realize those task vectors, and prompt-invariance forces zero attention on them whenever unambiguous examples are also present.

We now outline this reasoning more formally. We use the variables $p_A$ for prompts for task $A$ (with at least one unambiguous example); $p_B$ for prompts for task $B$ (with at least one unambiguous example); $p_U$ for fully ambiguous prompts.

For a prompt $p = x_1 y_1 \ldots x_n y_n x_{n+1}$,

$$\psi_{p_t} := \sum_{i=1}^{n} a_i \cdot \psi(x_i, y_i) \tag{5}$$

For each $t \in \{A, B\}$, we define $\phi_t(x_{n+1})$ to be the average of all outputs of $\phi$:

$$\phi_t(x_{n+1}) := \mathbb{E}_{p_t}\left[\phi(x_{n+1}, \psi_{p_t})\right]$$

where the expectation runs over prompts for the task $t$ with at least one unambiguous example. We similarly define $\phi_{AB}$ in terms of all-ambiguous prompts.

We now define a lower bound on the loss which will turn out to be tight if and only the model satisfies the desired structural properties.

First, we have for each $x_{n+1}$:

$$\mathbb{E}_{p_t}\left[(\phi(x_{n+1}, \psi_{p_t}) - F_t(x_{n+1}))^2\right] \geq \left(\mathbb{E}_{p_t}\left[\phi(x_{n+1}, \psi_{p_t})\right] - F_t(x_{n+1})\right)^2 = (\phi_t(x_{n+1}) - F_t(x_{n+1}))^2 \tag{6}$$

due to Jensen's inequality. This bound is tight if and only if $\phi(x_{n+1}, \psi_{p_t})$ is independent of the prompt $p_t$ when fixing $x_{n+1}$ and the task $t$.

Take

$$L := \max_{x_{n+1} \in \mathcal{X}} \sup_{x \in \mathbb{R}^d} \underbrace{\|\nabla_2 \phi(x_{n+1}, x)\|_2}_{Smoothness\ of\ \phi} \tag{7}$$

Furthermore, by averaging over all pairs of unambiguous prompts with matching ICL queries,

$$
\begin{aligned}
|\phi_A(x_{n+1}) - \phi_B(x_{n+1})| =& |\mathbb{E}_{p_A, p_B}(\phi(x_{n+1}, \psi_{p_A}) - \phi(x_{n+1}, \psi_{p_B}))| \\
\leq& L \cdot \mathbb{E}_{p_A, p_B}\|\psi_{p_A} - \psi_{p_B}\| \\
\leq& L \cdot \mathbb{E}_{p_A, p_B}\left(\|\psi_{p_A}\|_2 + \|\psi_{p_B}\|_2\right) \\
=& L \cdot \left(\mathbb{E}_{p_A}\|\psi_{p_A}\|_2 + \mathbb{E}_{p_B}\|\psi_{p_B}\|_2\right) \\
\leq& 2 \cdot L \cdot \left(\max_{x,y}\|\psi(x,y)\|_2\right)
\end{aligned}
$$

As a consequence,

$$\frac{|\phi_A(x_{n+1}) - \phi_B(x_{n+1})|}{2} \leq \max_{x,y}\|\psi(x,y)\|_2 \cdot L \tag{8}$$

for each $x_{n+1}$.

Define $S := \max_{x,y}\|\psi(x,y)\|_2^2$. For any unambiguous ICL query $x_{n+1}$, we have $\{F_A(x_{n+1}), F_B(x_{n+1})\} = \{-1, 1\}$. Write

$$u := \phi_A(x_{n+1}), \quad v := \phi_B(x_{n+1}),$$

and, without loss of generality, assume $F_A(x_{n+1}) = 1$ and $F_B(x_{n+1}) = -1$. Define

$$m := \frac{u+v}{2}, \quad s := \frac{u-v}{2}.$$

Then

$$\frac{(u-1)^2 + (v+1)^2}{2} = m^2 + (s-1)^2.$$

By (8), we have $|s| \leq L\sqrt{S}$. Therefore,

$$\frac{(u-1)^2 + (v+1)^2}{2} \geq \inf_{|s| \leq L\sqrt{S}} \left(m^2 + (s-1)^2\right) = (1 - L\sqrt{S})_+^2.$$

Indeed, the infimum is attained at $m = 0$ and at the feasible value of $s$ closest to $1$. Hence, for every unambiguous ICL query,

$$\frac{(\phi_A(x_{n+1}) - F_A(x_{n+1}))^2 + (\phi_B(x_{n+1}) - F_B(x_{n+1}))^2}{2} \geq (1 - L\sqrt{S})_+^2.$$

Averaging over queries over the event $F_A(x_{n+1}) \neq F_B(x_{n+1})$, whose probability is $\delta > 0$, gives the following lower bound when $L \neq 0$:

$$\begin{aligned}
\mathcal{L} &= \mathbb{E}_{p_t}[(\phi(x_{n+1}, \psi_{p_t}) - F_t(x_{n+1}))^2] + \tau S + \eta L \\
&\geq \mathbb{E}[(\phi_t(x_{n+1}) - F_t(x_{n+1}))^2] + \tau S + \eta L \\
&\geq \delta(1 - L\sqrt{S})_+^2 + \tau S + \eta L
\end{aligned} \tag{9}$$

where $\delta > 0$ is the probability, under the data distribution, that $F_A(x_{n+1}) \neq F_B(x_{n+1})$.

We thus have a lower-bound on the loss function in the $L \neq 0$ case, which depends only on $L, S$. We note that, if $L = 0$, then the model output is independent of the second argument of $\phi$ and therefore cannot use the few-shot examples. This case is excluded by the assumption that we are considering a global optimum with above-chance performance.

Assuming $L \neq 0$, we are looking for the following structural features in the model:

1. for unambiguous examples, $\psi(x_i, y_i)$ depends only on the task

2. $\psi_{p_A} = -\psi_{p_B}$

3. $\phi$ is linear on the line between $\psi_{p_A}$ and $\psi_{p_B}$

4. in prompts containing both ambiguous and unambiguous examples, attention is zero on the ambiguous examples

We want to show that any global optimum that has above-chance performance must satisfy these features.

First, for any choice of $S := \max_{x,y} \|\psi(x,y)\|_2^2$ and $L$, there is a model satisfying 1–4 that makes (6), (8), and (9) tight. Take $\psi_A$ to be any vector of magnitude $\sqrt{S}$; $\psi_B := -\psi_A$; $\psi_U = 0$. Let $\phi(x_{n+1}, \psi)$ be linear in $\psi$ with slope $\pm L$ when $F_A(x_{n+1}) \neq F_B(x_{n+1})$, and constant equal to $F_A(x_{n+1})$ when it equals $F_B(x_{n+1})$. Make attention uniformly nonzero on the unambiguous examples, and zero on ambiguous examples. This construction turns Ineqs. 6, 8, 9 into equalities. As a consequence, any model for which the bound (9) is not an equality cannot be a global optimizer of the loss, because there would be a model with the same $S, L$ and a lower loss.

Second, consider any global optimum. By the previous considerations, this configuration must make the bound tight, i.e., turn both inequalities (6, 8) into equalities. First, to make (6) tight, $\phi(x_{n+1}, \psi_{p_t})$ must be independent of the examples beyond the task $t$ and the ICL query $x_{n+1}$. Second, we need to make the chain of inequalities leading to (8) tight for each $x_{n+1}$. One possibility is that $L = 0$, that is, $\phi(x_{n+1}, \psi_{p_t})$ is independent of the task for any $x_{n+1}$. The other possibility is that $L \neq 0$. We now spell out why tightness forces $\psi_{p_A}$ and $\psi_{p_B}$ to be prompt-independent. Write

$$Z_{p_A, p_B} := \phi(x_{n+1}, \psi_{p_A}) - \phi(x_{n+1}, \psi_{p_B})$$

and $M := \max_{x,y} \|\psi(x,y)\|_2$. Since prompts are sampled uniformly from a finite set, equality in the chain of inequalities leading to (8) implies

$$|\mathbb{E}Z_{p_A, p_B}| = \mathbb{E}|Z_{p_A, p_B}| = L \cdot \mathbb{E}_{p_A, p_B} \|\psi_{p_A} - \psi_{p_B}\| = L \cdot \mathbb{E}_{p_A, p_B} (\|\psi_{p_A}\|_2 + \|\psi_{p_B}\|_2) = 2LM.$$

Equality in the first step implies that $Z_{p_A, p_B}$ has a constant sign for every pair $(p_A, p_B)$ in the support. For the remaining steps, the slacks are pointwise nonnegative, so equality of expectations implies pointwise equality for every pair in the support. In particular,

$$\|\psi_{p_A} - \psi_{p_B}\|_2 = \|\psi_{p_A}\|_2 + \|\psi_{p_B}\|_2$$

for every such pair, so $\psi_{p_A}$ and $\psi_{p_B}$ are negatively collinear. Also, because $\|\psi_{p_A}\|_2 \leq M$ and $\|\psi_{p_B}\|_2 \leq M$, equality in

$$\mathbb{E}_{p_A} \|\psi_{p_A}\|_2 + \mathbb{E}_{p_B} \|\psi_{p_B}\|_2 \leq 2M$$

implies $\|\psi_{p_A}\|_2 = M$ for every $p_A$ and $\|\psi_{p_B}\|_2 = M$ for every $p_B$. Now fix one prompt $p_B^\star$. For any two prompts $p_A, p_A'$, both $\psi_{p_A}$ and $\psi_{p_A'}$ are negatively collinear with $\psi_{p_B^\star}$, hence positively collinear with each other; since both have norm $M$, they must be equal. Thus $\psi_{p_A}$ is the same vector for all $p_A$. The same argument, fixing one prompt $p_A^\star$, shows that $\psi_{p_B}$ is the same vector for all $p_B$. Therefore, for each fixed ICL query $x_{n+1}$, the aggregate vectors $\psi_{p_A}$ and $\psi_{p_B}$ depend only on the task and not otherwise on the prompt. Furthermore, to make the Lipschitz bound tight,

$$|\phi(x_{n+1}, \psi_{p_A}) - \phi(x_{n+1}, \psi_{p_B})| = L \cdot \|\psi_{p_A} - \psi_{p_B}\| \tag{10}$$

which means that $\phi(x_{n+1}, \cdot)$ must be linear between $\psi_{p_A}$ and $\psi_{p_B}$.

So far, we have found that $\psi_{p_A}$ and $\psi_{p_B}$ depend on the task but not the prompt beyond $x_{n+1}$. By considering one-shot prompts, this immediately entails that all unambiguous examples $(x_i, y_i)$ have $\psi(x_i, y_i)$ that only depends on the task. Thus, when all examples are unambiguous, $\psi_{p_t}$ depends only on the task. Let $u_A$ and $u_B$ be the corresponding aggregate vectors for tasks $A$ and $B$ for a fixed ICL query $x_{n+1}$. By the argument above, these are opposite nonzero vectors, and $\phi(x_{n+1}, \cdot)$ is linear on the segment between them.

Now consider a fully ambiguous prompt. Conditioned on such a prompt, the underlying task is equally likely to be $A$ or $B$, so the Bayes-optimal prediction is the midpoint

$$\frac{F_A(x_{n+1}) + F_B(x_{n+1})}{2}.$$

By the linearity of $\phi(x_{n+1}, \cdot)$ on the segment between $u_A$ and $u_B$, this midpoint output is attained at the midpoint of that representation segment, namely at

$$\frac{u_A + u_B}{2} = 0.$$

Because the prompt distribution includes 1-shot prompts, we may take a prompt consisting of a single ambiguous example $(x, y)$ together with the ICL query. The sole example then receives attention weight 1, so the aggregate is exactly $\psi(x, y)$. Therefore, for an unambiguous ICL query $x_{n+1}$,

$$\phi(x_{n+1}, \psi(x, y)) = \frac{F_A(x_{n+1}) + F_B(x_{n+1})}{2}.$$

In particular, when $F_A(x_{n+1}) \neq F_B(x_{n+1})$, this value is 0. Hence an ambiguous example cannot satisfy $\psi(x, y) = u_A$ or $\psi(x, y) = u_B$, because those vectors map to the task-specific outputs rather than their midpoint.

For theorem item 2, start from the prompt obtained by deleting all ambiguous examples while keeping the same task and ICL query $x_{n+1}$. By prompt-independence, this reduced prompt has the same aggregate as the original mixed prompt. Since every remaining unambiguous example has vector $u_t$, the reduced-prompt aggregate is exactly $u_t$. Now add the ambiguous examples back one by one. After each addition the aggregate must still equal $u_t$, but the new ambiguous example has $\psi(x, y) \neq u_t$. Under a softmax convex combination, this is only possible if the new example receives attention weight zero. Repeating this for each ambiguous example shows that every ambiguous example in a mixed prompt receives zero attention. Hence attention is strictly lower on ambiguous examples than on unambiguous examples in any mixed prompt.

Overall, we have shown the structural features listed above must hold for any global optimum that has above-chance performance.

We further note that, for any choice of $\tau, \eta$ there must be a minimum at finite $\psi$ (and no minimum at infinite $\psi$) because the loss increases when taking $\psi$ to infinity.

Taken together, we conclude Theorem M.1. $\qquad\square$

