# OpenReview forum: "How Few-Shot Examples Add Up: A Causal Decomposition of Function Vectors in In-Context Learning"
_ICML.cc/2026/Conference — ICML 2026 regular_

### Official Review · Reviewer_zjgS · 2026-03-10

**Soundness:** 2
**Presentation:** 3
**Significance:** 3
**Originality:** 3
**Overall Recommendation:** 4
**Confidence:** 3

**Summary:**

This paper investigates how few-shot examples in a prompt combine to drive in-context learning (ICL) in large language models (LLMs). Building on prior work showing that ICL induces function vectors (FVs)—causal activation directions that encode the task and can be extracted from specific attention heads—the authors ask how an FV is constructed from individual examples within a prompt. They find that an $n$-shot FV can be approximated through a linear combination of per-example sub-FVs, implying that demonstrations contribute additively to the full-prompt FV. They then examine how the FV-relevant attention heads weight different demonstrations. Beyond recency and positional biases, they find that LLMs tend to assign more attention to unambiguous examples that more clearly specify the underlying task, and that contextualization (i.e., cross-example attention that lets examples influence one another) amplifies this pattern. Finally, using causal interventions that separately manipulate attention pathways, the authors show that contextualization improves FV "quality" (measured by FV injection accuracy) primarily through Query–Key alignment, while Value-mediated effects are more heterogeneous across tasks.

**Compliance With Llm Reviewing Policy:**

Affirmed.

**Final Justification:**

I think the rebuttal and discussion meaningfully strengthened the paper. I appreciate the added causal reconstruction results and the authors' clarification of the scope of their claims. In my view, the current evidence supports the conclusion that, within the extracted FV representation, linear combinations of sub-FVs approximate much of the causal effect of the full FV, and that attention routing is related to this process. This is a solid empirical finding, though I do not think the paper establishes that few-shot example integration is mechanistically linear in a more general sense. On balance, I remain supportive of acceptance, provided the final version keeps the claims carefully scoped and avoids overgeneralizing beyond what the evidence directly supports.

**Key Questions For Authors:**

1. The paper fits OLS weights $w_i$ to reconstruct full-prompt FVs from per-example sub-FVs. Is there any relationship between the fitted weights and the FV-head attention mass allocated to each example? More broadly, the paper would benefit from a clearer mechanistic account of how example contributions are integrated (beyond the reconstruction result).

2. The paper suggests a linear-additive integration rule and states in the Discussion that "FVs from different components are integrated into the residual stream via a linear combination rather than through competition"(lines 427–431). Could the authors clarify the intended scope/meaning of this claim and provide stronger supporting evidence (or explicit caveats)? In particular, it is unclear to me to what extent the current results rule out interaction/competition effects rather than showing a strong linear approximation.

3. Did the authors check how FV-head localization varies with the selection criteria and (possibly) across contexts? If so, how sensitive are the key conclusions to the identified FV-head set?

**Limitations:**

While FVs have been shown causally important for ICL, it remains unclear whether they capture the full set of causal mechanisms underlying ICL. It would be helpful for the authors to emphasize that their conclusions rely heavily on the FV framework and to discuss the potential limits of this scope.

**Strengths And Weaknesses:**

### Strengths

- The topic of how individual demonstrations are integrated to support ICL is an important one, as there are ongoing debates and inquiries into the mechanisms underlying ICL.

- This paper runs a large number of experiments investigating the topic, including linear approximations, attention-pattern analyses, and causal interventions. The authors also include LLMs of different sizes and from different model families.

- This paper involves thoughtful methodological design. The "uncontextualized" condition implemented via attention-edge ablation is a reasonable attempt to isolate cross-example interactions while preserving other computation, enabling analyses of effects of contextualization. The use of "ambiguous" prompts is also a good choice to analyze content-dependent routing that can be hard to observe under standard prompts.


### Weaknesses

- Heavy reliance on the FV framework and its operationalization. The paper's conclusions depend heavily on the correctness and completeness of FVs and FV-head localization, and FV injection accuracy is treated as the primary proxy for FV "quality." If FVs capture only part of the causal mechanisms underlying ICL, or if the identified FV head set is incomplete, then the scope of the conclusions may be narrower than implied.

- Limited causal support for the linear-additivity claim. The paper provides causal evidence that (i) FVs can elicit task behavior and (ii) contextualization improves FV quality largely through QK alignment. However, the central "additivity" result is supported primarily by representational reconstruction (high cosine similarity / $R^2$ under OLS fits) rather than direct causal evidence. Consequently, it remains uncertain whether linear additivity reflects the underlying computation or just an approximation in the examined settings. This concern is amplified by the focus on relatively simple toy tasks, making it unclear how well the claimed linear-additive structure generalizes to broader ICL settings.

- Interpretability of globally fit mixture weights $w_i$​. The paper fits OLS weights globally across prompts, which helps reveal stable position-level effects but may obscure prompt-specific composition (e.g., dependence on the query or demonstration content). Moreover, the positional nature of the fitted weights—and their shift between contextualized and uncontextualized settings—is not straightforward to interpret mechanistically. Contextualization plausibly changes both (i) the per-example sub-FVs themselves and (ii) the attention routing that aggregates information across examples. The differences in $w_i$​ may reflect changes in the sub-FVs and/or approximation artifacts, rather than indicating "how much each example contributes."

---

> ### Author Rebuttal · Authors · 2026-03-29
>
> We thank Reviewer zjgS for the rigorous and constructive feedback. In particular, the concerns about the scope of the FV framework and the interpretability of OLS weights motivated additional quantitative analyses. We address the main points below.
>
> > Q1: Heavy reliance on the FV framework and its operationalization..
>
> A1: We agree that our conclusions are bounded by the FV framework. We will make this explicit in the limitations: FVs may not capture the full set of causal mechanisms underlying ICL. Accordingly, our findings on linear additivity and attention reweighting should be interpreted within the FV framework and its associated localization procedures.
>
> > Q2: Limited causal support for the linear-additivity claim.
>
> A2: To strengthen this point beyond representational reconstruction, we additionally conducted a causal reconstruction analysis on Gemma-2-9B across all 10 tasks. We injected linearly reconstructed FVs (weighted sums of sub-FVs) into 0-shot prompts and measured the resulting injection accuracy.
>
> To summarize the result compactly, we report the ratio between the accuracy induced by the reconstructed FV and that of the true full-prompt FV (OLS / Full-FV Baseline). A value close to 1 indicates that the linear reconstruction recovers most of the causal task effect of the full FV.
>
> | Setting | Mean OLS/Baseline | Median OLS/Baseline | # Tasks >= 0.90 | # Tasks >= 0.80 |
> |---|---:|---:|---:|---:|
> | 3-shot ctx | 0.984 | 0.974 | 10/10 | 10/10 |
> | 3-shot unc | 0.890 | 0.883 | 5/10 | 8/10 |
> | 5-shot ctx | 0.946 | 0.973 | 8/10 | 9/10 |
> | 5-shot unc | 0.933 | 1.000 | 7/10 | 8/10 |
>
> In the main contextualized setting, this ratio averages 0.984 for 3-shot and 0.946 for 5-shot prompts. For 3-shot contextualized prompts, 10/10 tasks reach at least 0.90 of the full-FV baseline; for 5-shot contextualized prompts, 8/10 reach at least 0.90, and 9/10 reach at least 0.80. We also observe similar, though noisier, recovery in the uncontextualized setting. Since the clean 0-shot baseline is near zero across tasks, these ratios reflect substantial causal recovery rather than preservation of an already-strong baseline.
>
> This result does provide causal behavioral support that linear combinations of sub-FVs capture a substantial part of the effective task direction in the tested settings.
>
> > Q3: Interpretability of globally fit mixture weights ($w_i$) vs. attention mass.
>
> A3: We agree that globally fitted OLS weights can absorb both sub-FV changes and approximation artifacts. To study their relationship to attention routing, we computed the Pearson correlation ($r$) and Spearman rank correlation ($\rho$) between normalized $w_i$ vectors and attention-mass distributions under 5-shot setting:
>
> | Task Type | Task | Pearson $r$ (unc / ctx) | Spearman $\rho$ (unc / ctx) |
> | :--- | :--- | :--- | :--- |
> | **Ambiguous** | PP-A | 0.80 / 0.85 | 0.70 / 0.50 |
> | | ZH-A | 0.69 / 0.76 | 0.70 / 0.67 |
> | | JA-A | 0.97 / 0.70 | 0.97 / 0.82 |
> | | FR-A | 0.97 / 0.87 | 1.00 / 0.60 |
> | | CAP-A | 0.82 / 0.83 | 0.82 / 0.67 |
> | **Normal** | CC | 0.97 / 0.70 | 0.97 / 0.70 |
> | | PS | 0.78 / 0.82 | 0.90 / 1.00 |
> | | PC | -0.51 / 0.07 | -0.47 / 0.10 |
> | | PP | 1.00 / 0.84 | 1.00 / 0.56 |
> | | EN-FR | 0.98 / 0.73 | 1.00 / 0.21 |
>
> We broadly observe strong correlations across both Normal & Ambiguous settings (unc/ctx). The slight drop under contextualization in some settings is consistent with the reviewer's intuition that contextualization can shift sub-FVs geometry. Overall, these analyses suggest that attention mass and OLS weights are highly related. In Appendix L, we give a first step attempt at discussing how contextualization reshapes the FVs themselves in geometry under controlled attention routing conditions.
>
> > Q4: Linear-additive integration rule vs. competition.
>
> A4: We agree that the original wording is not rigorous. Our intended claim is: within the identified FV heads, sub-FV aggregation is a linear combination at the formation of the overall FV. This is not a totally competitive process, since irrelevant demonstrations corresponding to other tasks do not compete to alter the designated task direction of an FV head. We do not intend this to rule out interaction or competition effects elsewhere in the computation, including downstream execution of different task FVs. We will revise the discussion accordingly.
>
> > Q5: FV-head localization sensitivity and selection criteria.
>
> A5: We followed the Average Indirect Effect criterion used in the original FV paper. As a robustness check, we compared localized FV headsets across different $n$-shot settings and also localized FV heads for the ambiguous datasets using all-unambiguous prompts. The resulting headsets exhibited no significant difference: the top FV heads remained largely consistent across these conditions and only minor differences in relative ranking.

---

> > ### Author Rebuttal · Reviewer_zjgS · 2026-04-03
> >
> > Thank you for the detailed rebuttal, the clarifications, and the new analyses. I think the response improves the paper, especially by clarifying the scope of the claims and by providing additional causal reconstruction results. I found the new evidence helpful, but I still have some reservations about how strongly the paper supports its broader mechanistic conclusions.
> >
> > In particular, **the mechanistic account remains incomplete.**
> >
> > The new causal reconstruction result is valuable because it goes beyond representational fit and tests whether linearly reconstructed FVs preserve the causal task effect of the full extracted FV. This is stronger evidence than cosine similarity or $R^2$ alone, and the reported contextualized results appear strong. However, I think this is still evidence for a strong linear approximation within the FV framework, rather than a convincing demonstration that few-shot example integration is mechanistically linear. It also does not rule out important interaction or competition effects outside this approximation.
> >
> > Similarly, the correlations between OLS weights and attention mass are suggestive, but they do not fully establish a mechanistic interpretation of the fitted weights, especially given the occasionally weak correlations. The added experiments reduce some uncertainty here, but they do not fully resolve the ambiguity about what these fitted weights are actually capturing mechanistically.
> >
> > My main concern is therefore about the scope of the conclusions. The rebuttal makes the paper stronger, but I still think the results support a narrower conclusion than the paper sometimes suggests: namely, that *within the extracted FV representation, linear combinations of sub-FVs approximate much of the causal effect of the full FV, and that attention routing is related to this process.* What remains unclear is the precise mechanism by which LLMs integrate few-shot examples, and whether important nonlinear or competitive effects exist.
> >
> > Overall, I find that the rebuttal meaningfully strengthens the paper, but some of the core mechanistic questions remain open.

---

> > > ### Author Response · Authors · 2026-04-03
> > >
> > > We thank Reviewer zjgS for the response and the constructive discussion. We are glad that you found the new causal reconstruction results valuable and strong, and that the additional analyses meaningfully strengthen the paper.
> > >
> > > Regarding the broader mechanistic scope, we completely agree with your precise summary of our core conclusion: specifically within the extracted FV representation, linear combinations of sub-FVs robustly approximate the causal effect of the full prompt. Rather than claiming to account for the entirety of ICL mechanism, which may indeed involve downstream nonlinear or competitive effects. By demonstrating that integration is fundamentally linear and additive at the FV formation stage, we provide a concrete, causally verified mechanism for how few-shot examples compose within this framework.
> > >
> > > To prevent any ambiguity or overclaiming, we will revise to strictly define this boundary, ensuring our mechanistic claims are bounded within the FV operationalization, exactly as you have noted. We appreciate the rigorous feedback, which has helped ensure both the empirical depth and the precision of our claims.

---

### Official Review · Reviewer_ppAX · 2026-03-12

**Soundness:** 3
**Presentation:** 2
**Significance:** 3
**Originality:** 3
**Overall Recommendation:** 4
**Confidence:** 2

**Summary:**

This paper investigates the formation of Function Vectors (FV) within the In-Context Learning (ICL) framework, specifically focusing on how individual demonstrations contribute to the final model output. The authors discover that models contextualize example representations based on preceding demonstrations, leading to an adaptive reweighting of which examples dominate the FV. A key finding is that the consistency of FV quality is largely driven by Query–Key alignment, particularly in scenarios involving high ambiguity.

**Compliance With Llm Reviewing Policy:**

Affirmed.

**Final Justification:**

Thank you for the authors' response. I decide to keep my score.

**Key Questions For Authors:**

See Weakness

**Limitations:**

See Weakness

**Strengths And Weaknesses:**

## Strengths

*  The research addresses a critical question regarding the inner workings of In-Context Learning, a paradigm that has become central to current LLM deployment.

* The paper introduces the concept of **additive superposition with context-dependent attention reweighting** as a potential explanation for how models synthesize information from multiple examples.



## Weaknesses

1. The writing lacks the necessary clarity to bridge the gap between theory and evidence. While the authors propose "additive superposition with context-dependent attention reweighting," the paper only presents fragmented evidence of this mechanism. It remains unclear how these individual components integrate into a unified, functional theory.

 2. While the paper explores the influence of demonstration positioning—a phenomenon generally recognized in the field—it fails to articulate how these positional results specifically validate or construct the proposed overarching mechanism.

3. The transition from individual experiments to the global mechanism is underdeveloped, leaving the reader to guess how the "fragments" of data support the paper's primary claims.

---

> ### Author Rebuttal · Authors · 2026-03-29
>
> We thank Reviewer ppAX for the constructive feedback and the opportunity to clarify the cohesion of our work. Below, we address your concerns regarding the synthesis of our findings and the role of positional analysis.
>
> > Q1: The writing lacks the necessary clarity to bridge the gap between theory and evidence... The transition from individual experiments to the global mechanism is underdeveloped, leaving the reader to guess how the "fragments" of data support the paper's primary claims.
>
> A1: We appreciate this feedback and agree that the global mechanism should be stated more explicitly. Our intended account has three linked parts:
> (1) Section 3 shows that full-prompt FVs are well approximated by linear combinations of example-level sub-FVs;
> (2) Section 4 shows that contextualization changes the effective contribution of these components by reweighting attention toward more unambiguous examples;
> (3) Section 5 provides causal evidence that this reweighting improves FV quality primarily through QK-side routing effects.
>
> We will revise the presentation to make this 3-step mechanism explicit and to connect each experiment more clearly to the specific part it supports.
>
> > Q2: While the paper explores the influence of demonstration positioning—a phenomenon generally recognized in the field—it fails to articulate how these positional results specifically validate or construct the proposed overarching mechanism.
>
> A2: We acknowledge that positional influences (e.g., recency bias) are important factors. Our aim is not to treat position influence as an independent mechanism, but rather as an observable factor that interacts with the broader attention reweighting account. In the ambiguous prompt settings, by varying the position of the unambiguous examples and showing that attention still preferentially concentrates on them, we use this analysis to confirm that this effect depends heavily on the semantic nature of the demonstration itself, rather than being attributable to position effects.

---

> > ### Author Rebuttal · Reviewer_ppAX · 2026-04-03
> >
> > Thank you for the authors' response. I decide to keep the score.

---

### Official Review · Reviewer_nibc · 2026-03-13

**Soundness:** 3
**Presentation:** 3
**Significance:** 3
**Originality:** 3
**Overall Recommendation:** 5
**Confidence:** 2

**Summary:**

This paper provides a mechanistic explanation of the ability of LLMs to perform in-context learning (ICL), in terms of the function vector (FV). Specifically, it shows that the overall FV is roughly the same as the linear combination of per-example FVs. Moreover, it shows that more attention is paid to unambiguous example, with contextualization acting as a re-weighting filter. Lastly, it shows that when ambiguous examples are present, query-key alignment consistently improves FV quality.

**Compliance With Llm Reviewing Policy:**

Affirmed.

**Final Justification:**

The authors has clarified my concerns.

**Key Questions For Authors:**

1. Could your study answer the following question: what proportions of ambiguous vs. unambiguous examples in the prompts can enable successful ICL?

2. (minor) the last sentence of the first column of Page 3 ("To extract a per-example ...") seems to be missing some words after "we"

**Limitations:**

Yes

**Strengths And Weaknesses:**

**Soundness:** This paper is technically sound. The contribution is clearly stated and experiments are rigorously performed to support their hypotheses.

**Presentation:** This paper is well-structured and clearly written, albeit a bit dense at some parts.

**Significance:** This paper addresses an important problem on the mechanism by which few-shot ICL occurs, focusing on function vectors.

**Originality:** This paper deepens our understanding on how few-shots ICL prompts contribute to the corresponding function vector, and how ICL occurs in the presence of ambiguous examples.

---

> ### Author Rebuttal · Authors · 2026-03-29
>
> We thank Reviewer nibc for the helpful review and address the specific questions below.
>
> > Q1: Could your study answer the following question: what proportions of ambiguous vs. unambiguous examples in the prompts can enable successful ICL?
>
> A1: While our current study does not aim to identify a precise proportion threshold for successful ICL, our mechanistic results suggest that success depends less on a hard proportion and more on the effective attention mass captured by the unambiguous examples. Adding more ambiguous examples is expected to dilute the effective attention allocated to the unambiguous ones, which may eventually prevent successful task induction. However, in our tested 3/5/10-shot settings, a small proportion of unambiguous examples is sufficient for success, provided that QK alignment concentrates sufficient attention on them.
>
> > Q2: the last sentence of the first column of Page 3 ("To extract a per-example ...") seems to be missing some words after "we"
>
> A2: We thank the reviewer for catching this mistake. The sentence should read: “To extract a per-example sub-FV, we mask attention ...”.

---

> > ### Author Rebuttal · Reviewer_nibc · 2026-04-03
> >
> > Thank you for the response! I am raising my score to 5.

---

### Official Review · Reviewer_Rp4U · 2026-03-13

**Soundness:** 3
**Presentation:** 3
**Significance:** 3
**Originality:** 3
**Overall Recommendation:** 5
**Confidence:** 2

**Summary:**

This paper investigates the internal mechanism of few-shot in-context learning and explains how multiple demonstrations jointly form a function vector. It shows that an n-shot function vector can be approximated as a linear superposition of example-level sub-FVs, while contextualization further reweights attention toward demonstrations that more clearly identify the task.

**Compliance With Llm Reviewing Policy:**

Affirmed.

**Key Questions For Authors:**

1. The paper argues that, in most ambiguous settings, Query–Key alignment is driven primarily by the demonstrations rather than the query, but to what extent does this conclusion depend on the artificially constructed ambiguous prompts, and can the authors provide stronger evidence for its generality?

2. Authors point out that an n-shot FV can be approximated as a linear superposition of example-level sub-FVs, but does this conclusion hold only in the controlled 3-shot, 5-shot, and 10-shot settings; when the context becomes much longer and the number of demonstrations increases substantially, could this linear approximation break down?

3. The paper provides a mechanistic account of how demonstrations form and reweight function vectors, but can these findings be translated beyond controlled conditions into improvements in in-context learning performance on more realistic tasks?

**Limitations:**

yes

**Strengths And Weaknesses:**

Strengths:
1.The paper offers a prompt-level, causal account and a mechanistic, testable account of how few-shot examples form and refine function vectors.
2.Authors combines near-linear superposition with a causal decomposition of Query–Key and Value pathways, and shows that Query–Key alignment consistently improves FV quality.

Weaknesses:
1.This paper focuses on discrete mapping toy tasks, so extending the framework to long-horizon or compositional reasoning remains a key challenge.
2.Value effects are still unclear, and ambiguity is operationalized through controlled task constructions.

---

> ### Author Rebuttal · Authors · 2026-03-29
>
> We thank Reviewer Rp4U for the positive evaluation and constructive feedback. Below we address the specific questions raised.
> > Q1: To what extent does this conclusion(in most ambiguous settings, Query–Key alignment is driven primarily by the demonstrations rather than the query) depend on the artificially constructed ambiguous prompts?
>
> A1: We agree that our ambiguous prompts are a controlled diagnostic setup, but the ambiguity is not introduced by artificially fabricated, contradictory, or factually invalid demonstrations. The ambiguous demonstrations are still semantically valid examples drawn from the same task distribution; they are “ambiguous” only in the sense that, taken in isolation, they can support more than one task hypothesis until additional context is provided. In this respect, the ambiguous prompts do not differ sharply from the normal prompts at the semantic level. What we control is primarily the composition of the prompt—i.e., the ratio of ambiguous to unambiguous demonstrations—rather than constructing synthetic or inconsistent examples to force a particular effect. We therefore view the setup as a controlled way to expose context-dependent routing under genuine ambiguity, rather than an artifact of inserting unnatural demonstrations. Moreover, under the same ambiguous prompt structure, different tasks need not show the same dominant disambiguating source: for example, in CAP-A the query plays the stronger role. This suggests that prompt structure alone is not sufficient to explain the broad demonstration-driven alignment observed in the other tasks.
>
> > Q2: Robustness of the linear approximation in much longer contexts?
>
> A2: We agree that this is an important boundary condition. Our main claims are based on the tested regimes, where the linear approximation is robust. To probe whether the linear component persists in longer contexts, we additionally ran a preliminary **20-shot** validation on two representative tasks (COUNTRY-CAPITAL and PRESENT-PAST AMBIGUOUS) on Gemma-2-2B and 9B. For these tasks, we examined both representation-level reconstruction and causal recovery(see our response to Reviewer zjgS, Q2).
>
> | Model | Task | Cosine Similarity | R² | Accuracy OLS / Full-FV |
> |---|---|---:|---:|---:|
> |Gemma-2-9B | COUNTRY-CAPITAL (ctx) | 0.846 | 0.694 | 1.116 |
> || COUNTRY-CAPITAL (unc) | 0.923 | 0.830 | 1.000 |
> || PRESENT-PAST-AMBIGUOUS (ctx) | 0.910 | 0.812 | 0.818 |
> || PRESENT-PAST-AMBIGUOUS (unc) | 0.936 | 0.826 | 1.000 |
> | Gemma-2-2B | COUNTRY-CAPITAL (ctx) | 0.918 | 0.838 | 0.936 |
> || COUNTRY-CAPITAL (unc) | 0.866 | 0.733 | 0.975 |
> || PRESENT-PAST AMBIGUOUS (ctx) | 0.874 | 0.756 | 0.884 |
> || PRESENT-PAST AMBIGUOUS (unc) | 0.965 | 0.899 | 0.923 |
>
> While this is only a limited sample rather than a full many-shot sweep across all tasks, it suggests that the linear component does not collapse in longer contexts: even at 20-shot, linear reconstruction can still explain a substantial fraction of the full-prompt representation and recover much of its causal effect. We will clarify that our strongest evidence comes from the main tested regimes, and treat longer-context robustness as promising but still preliminary.
>
> > Q3: The paper provides a mechanistic account of how demonstrations form and reweight function vectors, but can these findings be translated beyond controlled conditions into improvements in in-context learning performance on more realistic tasks?
>
> A3: While improving realistic benchmarks is beyond the scope of this paper which focuses on foundational insights, our findings provide actionable paths toward such improvements. For instance, understanding that QK alignment naturally concentrates on unambiguous examples can directly guide prompt engineering; one can leverage this effect to distinguish demonstration quality and selectively sample high-quality examples when constructing prompts. Furthermore, routing-focused interventions may be a promising direction for improving robustness under noisy contexts.

---

### Decision · Program_Chairs · 2026-04-30

**Decision:**

Accept (regular)

**Comment:**

This paper studies few-shot ICL through the lens of FVs. It concludes that the function vector induced by few-shot demonstrations can be approximated as a linear combination of example-level sub-FVs, and highlights the role of Query-Key alignment in reweighting the contributions of different in-context examples.

The paper is well organized and provides sufficient empirical evidence to support its main claims. The topic is timely and of strong interest to the ICML community. During the rebuttal and discussion phases, the authors addressed most of the concerns raised by reviewers, and the overall feedback is positive.